# Boundary-layer height and surface stability at Hyytiälä, Finland in ERA5 and observations

Victoria A. Sinclair[1], Jenna Ritvanen[1,2], Gabin Urbancic[2], Irina Statnaia[1,2], Yurii Batrak[3], Dmitri Moisseev[1,2], and Mona Kurppa[1]

[1]Institute for Atmospheric and Earth System Research / Physics, Faculty of Science, University of Helsinki, PO BOX 64, FI-00014
[2]Finnish Meteorological Institute, Helsinki, Finland
[3]Development Centre for Weather Forecasting, Norwegian Meteorological Institute, Oslo, Norway

**Correspondence:** Victoria A. Sinclair (victoria.sinclair@helsinki.fi)

**Abstract.** We investigate the boundary layer (BL) height at Hyytiälä in southern Finland diagnosed from radiosonde observations, a microwave radiometer (MWR) and ERA5 reanalysis. Four different, pre-existing algorithms are used to diagnose the BL height from the radiosondes. The diagnosed BL height is sensitive to the method used. The level of agreement, and the sign of systematic bias between the 4 different methods, depends on the surface-layer stability. For very unstable situations, the median BL height diagnosed from the radiosondes varies from 600 m to 1500 m depending on which method is applied. Good agreement between the BL height in ERA5 and diagnosed from the radiosondes using Richardson number-based methods is found for almost all stability classes, suggesting that ERA5 has adequate vertical resolution near the surface to resolve the BL structure. However, ERA5 overestimates the BL height in very stable conditions highlighting the on-going challenge for numerical models to correctly resolve the stable BL. Furthermore, ERA5 BL height differs most from the radiosondes at 18 UTC suggesting ERA5 does not resolve the evening transition correctly. BL height estimates from the MWR are also found to be reliable in unstable situations but often are inaccurate under stable conditions when, in comparison to ERA5 BL heights, they are much deeper. The errors in the MWR BL height estimates originate from the limitations of the manufacturers algorithm for stable conditions and also the mis-identification of the type of BL. A climatology of the annual and diurnal cycle of BL height, based on ERA5 data, and surface-layer stability, based on eddy covariance observations, was created. The shallowest (353 m) monthly median BL height occurs in February and the deepest (576 m) in June. In winter there is no diurnal cycle in BL height, unstable BLs are rare yet so are very stable BLs. The shallowest BLs occur at night in spring and summer and very stable conditions are most common at night in the warm season. Finally, using ERA5 gridded data we determined that the BL height observed at Hyytiälä is representative of most land areas in southern and central Finland. However, the spatial variability of the BL height is largest during daytime in summer reducing the area over which BL height observations from Hyytiälä would be representative of.

# 1 Introduction

The boundary layer (BL) is the lowest part of the troposphere and is in direct contact with the Earth's surface. All exchange of heat, momentum, moisture, and trace gases between the surface and the free troposphere takes place through the BL. Two key characteristics of the BL are its height and stability. These characteristics are of critical importance for determining the magnitude of fluxes between the surface and BL and for air quality and pollution dispersion. Typically, higher concentrations of pollutants occur when the BL is shallow and stably stratified. The BL height and stability are also crucial variables to include in the analysis of surface-based aerosol and trace gas concentration observations as often measurements performed at ground level are generalised to represent conditions throughout the BL and across larger horizontal scales. The accuracy of such an assumption depends on spatial and temporal variations in the BL height.

Unfortunately, given its critical importance, it is not straightforward to estimate the BL height (Seibert et al., 2000) as there are many different definitions of the BL height, the structure of the BL is very complex, and different types of BL occur, for example stable, neutral and convective BLs. In this study, we use a very broad definition of the BL: it is the lowest part of the atmosphere and its behaviour is directly influenced by the surface. We use the generic term of BL height for all types of BLs rather than attempt to use more specific terms, for example mixing height.

The BL height cannot be measured directly but must instead be estimated from observations made above the surface. Therefore, estimates of the BL height are challenging as vertical profiles of temperature and wind speed are required at high spatial resolution. These vertical profiles can be obtained from radiosonde soundings, tall towers or masts, or ground-based remote sensing instruments. Radiosondes provide direct observations of temperature, humidity and wind speed at high vertical resolution but such soundings typically only occur twice, or at most, four times a day and globally there are very few sounding stations resulting in limited spatial coverage. However, some radiosonde stations have long temporal records often extending back for at least 20 years. Observations from tall towers or masts provide high temporal resolution vertical profiles but these measurements typically extend only a few hundred meters above the surface and thus are only capable of resolving shallow BLs. Ground-based remote sensing observations provide vertical profiles of attenuated backscatter or turbulence at high temporal resolution which can be used to derive the BL height. However, a limitation of remote-sensing instruments, in comparison to radiosondes or tall-towers and masts, is that often more than one instrument is needed to measure both the profile of temperature and wind. For example, Doppler Lidars and wind profilers can only provide wind profiles whereas microwave radiometers only provide temperature profiles. An additional, general limitation of remote-sensing based instruments is the limited spatial coverage due to a small, albeit constantly growing, number of instruments worldwide and the absence of very long time series of observations. Even once vertical profiles of temperature and wind are obtained, quantitatively estimating the BL height objectively is still challenging as many different methods exist and it is well documented that different methods give different estimates of the BL height (Seibert et al., 2000; Lotteraner and Piringer, 2016)

Climatologies of the BL height have been created using observations from radiosondes, remote-sensing instruments and gridded reanalysis data sets. Seidel et al. (2010) analysed radiosondes from 505 stations worldwide over 10 years and used 7 different methods to estimate the BL height. They found substantial sensitivity of BL heights to the estimation method,

with parcel based methods producing lower estimates and methods that find the maximum in potential temperature gradient or minimum in relative humidity gradient producing the largest values of BL height. Seidel et al. (2010) also note that radiosonde soundings, with 2-4 profiles per day, are insufficient to capture the diurnal cycle. Beyrich and Leps (2012) created a climatology of BL height at Lindenberg, Germany based on more than 10 years of radiosondes and also estimated the uncertainty of the diagnosed BL heights. Their most notable results were that night time BLs are shallower in summer than in winter, and that the highest uncertainty in BL height estimates was from the 18 UTC soundings. Baars et al. (2008) and Granados-Muñoz et al. (2012) both used remote-sensing instruments to create one-year climatologies of BL height for specific European stations. Similarly, Collaud Coen et al. (2014) developed an operational BL height detection method using remote-sensing instruments and different algorithms to determine the BL height and compared the results to radiosonde based estimates and values from a high resolution numerical weather prediction model. For the two Swiss stations they considered, the microwave radiometer provided convective boundary-layer heights in good agreement with the radiosonde sounding while the model, in general, overestimated the depth of the BL.

In contrast to observations, gridded reanalysis data sets have uniform spatial coverage across the globe, long and consistent time series with no gaps, and the most recent reanalysis datasets have 3 or even 1 hour temporal resolution allowing the diurnal cycle to be resolved and, therefore, are well suited to created climatologies from. von Engeln and Teixeira (2013) created a 20-year, world-wide climatology of BL height based on ERA-Interim reanalysis and in a comparison to both sea- and land-based radiosondes found ERA-Interim values of BL height to be in good agreement with the observations. Seidel et al. (2012) also analysed ERA-Interim reanalysis and radiosondes for a 24 year period over most of Europe and the continental United States (regions north of 60°N in Europe were excluded). They conclude that a bulk Richardson number based method was optimal to diagnose BL height from radiosonde, reanalysis, and climate model data sets and that BL height estimates from ERA-Interim were deeper than those from the radiosondes.

Despite these numerous climatological studies on BL height, there are few studies which consider climatologies of boundary-layer or surface-layer stability. Liu and Liang (2010) analysed radiosondes from 14 major field campaigns around the world and, in addition to estimating the BL height, classified the BL type as either stable, neutral, or unstable, finding that the occurrence frequencies follow a narrow, intermediate, and wide Gamma distribution, respectively, over both land and oceans. Zhang et al. (2018) considered similar stability classes over China during summertime based on radiosondes released at 14:00 Beijing time and found that 70% of the radiosondes they considered sampled convective BLs, 26 % were neutrally stratified and only 4% were stably stratified.

In this study, we primarily focus on the BL height and stability at the SMEAR (Station for Measuring Ecosystem–Atmosphere Relations, Hari et al., 2013) station Hyytiälä, also referred to as SMEAR II, which is located in southern Finland (the station is described in detail in section 2). SMEAR II has extensive, long-term surface-based measurements of atmospheric, forest, and soil variables, and in particular, there are world-renown, long-term measurements of aerosol and trace gases starting from 1996 (Hari and Kulmala, 2005). Some previous studies have investigated the BL height at SMEAR II, however, many studies were case studies or short-term studies (e.g. Lauros et al., 2007; Eerdekens et al., 2009; Ouwersloot et al., 2012). A recent, more in-depth study of boundary-layer turbulence was presented by Manninen et al. (2018) who analysed one year of Doppler

lidar data from Hyytiälä to identify the main source of turbulent mixing. They show that there is almost no diurnal cycle in the occurrence or source of turbulence in winter whereas in summer surface-based turbulence has a strong diurnal cycle. Notably they also find a considerable amount of nocturnal mixing near the surface, especially during the winter.

The aim of the current study is to provide an in-depth, long-term analysis of boundary-layer height and stability at SMEAR II by combining a range of observations from different instruments with the most modern global reanalysis data set from the European Centre for Medium Range Weather Forecasts (ECMWF), ERA5. BL height estimates are taken from pre-existing methods and datasets; no new methods to estimate BL height are developed in this study. The measurement station and data used is described in section 2. While we focus on one specific station, it should be noted that the methods described here are applicable, in principle, to many stations worldwide, for example, all European Aerosols, Clouds, and Trace gases Research InfraStructure (ACTRIS) stations and Atmospheric Radiation Measurement (ARM) program locations. Given the different observation types, and the difference in data between observations and reanalysis, the BL height estimates we analyse have been produced using different methods. Therefore, we begin this investigation with an overview of these methods and discussion of how they theoretically compare to each other and what their known limitations are (section 3). In section 4, we show some examples of how these different methods, applied to observational and reanalysis data sets, estimate the boundary-layer height in both stable and unstable conditions. A systematic, statistical comparison between the different methods of diagnosing the BL height is given in Section 5. A long-term climatology of BL height (based on reanalysis) and surface-layer stability (based on observations) is presented in section 6 and an estimate of the spatial representativity of the measurements of BL height made at SMEAR II is made based on ERA5 reanalysis data in section 7. Lastly, in section 8 the results are discussed and conclusions presented.

## 2  Measurement station and Data

### 2.1  SMEAR II

The SMEAR II station (Hari and Kulmala, 2005; Petäjä et al., 2016) is located in southern Finland (61°51'N, 24°17'E, 181 m a.s.l.), 220 km northwest of Helsinki (Fig. 1). The station is situated in a Boreal Scots pine forest stand which is 59 year old (in 2021) and has a mean canopy height of around 18 m (Bäck et al., 2012). A small lake, Kuivajärvi, (surface area of 8.4 km$^2$) is located directly beside the station (Fig. 1). The station is located on slightly sloping land, which combined with the different land uses (forest, lake, some open clearings), means that the location is some-what inhomogeneous. Measurements began in 1996 and consist of extensive aerosol, trace gas and standard meteorological measurements. The bulk of the in-situ measurements are made either at ground level or on an instrumented tower at the main station location indicated by "M" in Figure 1. Ground-based remote sensing instruments have gradually been added to the site since 2014 and are mainly located on a small field 200 m from the main station. This means that not all observations used in this study are exactly co-located, and therefore caution is needed when comparing them to each other. The atmosphere at SMEAR II is generally very clean which makes it challenging to use ceilometers and Doppler lidars to derive BL height at this station due to the small signal to noise

ratio (Manninen et al., 2018). SMEAR II is part of the European Aerosols, Clouds, and Trace gases Research InfraStructure (ACTRIS) and is a certified Integrated Carbon Observation System (ICOS) atmospheric station.

## 2.2 Radiosondes

Regular, continuous radiosondes are not released from SMEAR II, however, radiosondes have been released during intensive campaigns. An intensive measurement campaign called The Biogenic Aerosols - Effects on Clouds and Climate (BAECC) took place in Hyytiälä from 1 February - 13 September 2014 (Petäjä et al., 2016, and also see the Data availability statement at the end of this paper) during which time the ARM Mobile facility was deployed to Hyytiälä. During the campaign, alongside with many other measurements, Vaisala RS92 radiosonde soundings were released four times per day. The radiosonde observations

correspond to the four standard synoptic times of 00, 06, 12 and 18 UTC (local solar time is UTC + approximately 1hr 40 minutes; local legal time is UTC+2 hr in winter, UTC+3 hr in summer). However, the actual release times of the radiosondes varied slightly from day-to-day but were always before these times; the earliest sounding was released 45 minutes before the corresponding synoptic time and the latest sounding 9 minutes before the synoptic time. The soundings were released from a small field approximately 200 m southwest from the main SMEAR II station (location marked by R in Fig. 1). The radiosondes

measure temperature, relative humidity, pressure and position via a GPS sensor every 2 seconds which corresponds to approximately every 10 - 15 metres or 1.1 - 1.3 hPa. Typically, the radiosonde observations extend from the surface to at least 20 km above the surface. From these observed variables, post-processing allows vertical profiles of dry and wet bulb temperature, relative humidity, pressure, altitude above sea level, horizontal wind speed and direction and geographic location to be provided. For all permanent sites and mobile facility deployments, ARM computes and provides many Value-Added-Products (VAPs).

One of these is the "Planetary Boundary Layer Height" which consists of four estimates of the BL height made from applying different methods to the radiosonde profiles. These different methods are described in section 3. Although the radiosonde data from the BAECC campaign was transmitted into the Global Telecommunication System (GTS), by inspecting the ERA5 analysis increment files, we conclude that these soundings have not been assimilated into ERA5.

## 2.3 Microwave Radiometer

In this study a humidity and temperature profiling microwave radiometer (MWR; Rose et al., 2005; RPG Radiometer Physics GmbH, 2014a) was used to provide an additional estimate of BL height. The radiometer has been a part of the ACTRIS cloud profiling station since June 2018. The MWR is located 200 m southeast from the main SMEAR II station (R in Fig. 1). The MWR measures emissions in two frequency bands, 22-31 GHz and 51-58 GHz, which correspond to water vapour and oxygen absorption lines. To further improve the retrieval of temperature profiles within the BL, measurements at 7 elevation angles,

starting at 5° from horizontal, were performed. The profiles are available approximately every 10 minutes, with vertical resolution varying from 30 m below 1200 m to 200 m above that. The temperature profiles are retrieved from the measurements with an artificial neural network algorithm (Solheim et al., 1998) that was trained for the climate of Southern Finland using several years of radiosonde observations from the Finnish Meteorological Institute (FMI) station in Jokioinen. These measurements were used to simulate MWR observations by applying radiative transfer computations. The simulated MWR and radiosonde

observations were used to train the neural network. A previous study on a similar MWR found a random error between the MWR temperature profiles and radiosonde measurements increasing from 0.5 K in the lower BL to 1.7 K at 4 km altitude (Löhnert and Maier, 2012), while the accuracy (as root mean square error) of the temperature profiles given by the manufacturer ranges from $\pm 0.7$ K in the BL to $\pm 1.0$ K above 2000 m. Retrieval of the BL height from the temperature profiles is described in Section 3.2.

Since the MWR measures rather weak emission signals, such things as a wet antenna radome due to rain can severely affect the quality of the measurements. To address this potential issue, the MWR assembly includes a heating element and air blower that continuously dries the radome, and the actual radome has a hydrophobic coating. However, this coating will wear off gradually over time and we would expect the quality of the measurements to decrease as the coating decays. To detect rain events, and remove observations that were potentially affected by a wet radome, the MWR setup includes a Vaisala WXT 530 weather station, which uses an impact sensor for the precipitation detection. Because this sensor may miss light precipitation, OTT Parsivel[2] (Tokay et al., 2014) and OTT Pluvio[2] measurements were also used to flag the data that could be potentially affected by rain. OTT Parsivel[2] is a present weather sensor capable of measuring precipitation intensity and raindrop size distribution, and OTT Pluvio[2] is a weighing precipitation gauge. Both instruments are located within 30 m of the MWR. To avoid using MWR measurements affected by rain, measurements at times when rain intensity exceeded 0.1 mm h$^{-1}$, corresponding to accumulation of 0.008 mm, during the previous 5 minutes, detected by either the OTT Parsivel[2] disdrometer or OTT Pluvio[2], were excluded from the analysis.

## 2.4 Eddy Covariance and surface stability

Eddy covariance (EC) is the method applied to compute vertical turbulent fluxes within the atmospheric BL. The vertical turbulent fluxes of scalar quantities are obtained by calculating the covariance of the scalar and the vertical component of the wind speed. At SMEAR II, two Gill Solent 1012R ultrasonic anemometers are installed on the instrumented tower (marked as M in Figure 1) which enables the use of the EC method. Note that the instrumented tower is in a slightly different location to the MWR and radiosonde release site. We use data from the lower of the two sonics which is at 23 m and therefore above the canopy. The sonic anemometer measures the three wind speed components and sonic temperature at 10 Hz. The turbulent fluxes analysed in this study were computed using the EC package EddyUH, a public software package developed at the University of Helsinki for post-field processing of eddy covariance data (Mammarella et al., 2016).

We retrieved the processed data of the sensible heat flux ($H$), the friction velocity ($u_*$), air temperature and pressure as 30-minute averages covering the time period 1 January 1997 to 31 December 2019 from the SmartSMEAR public online database (https://smear.avaa.csc.fi). This data set contains missing data which is due to the assumptions made during the computation of the fluxes (e.g. stationarity, $u_*$ exceeding a certain threshold) and technical instrumentation limitations (e.g. icing in winter). This topic, and the general quality control of EC data, is discussed in detail by Mammarella et al. (2016) and Vickers and Mahrt (1997).

Using this data we computed the Obukhov parameter, $L$ using

$$L = -\frac{T_0 u_*^3}{\kappa g \overline{w'T'}}. \tag{1}$$

where $\kappa$ is the von Kármán constant, g is the gravitational constant, $T_0$ is a reference temperature where we used the 30-minute mean temperature at 16.8 m, $\overline{w'T'}$ is the vertical velocity and temperature covariance (also called the kinematic heat flux) given by

$$\overline{w'T'} = \frac{H}{\rho c_p} \tag{2}$$

where $\rho$ is air density and $c_p$ is the heat capacity of air at constant pressure. Using the Obukhov length we can compute the stability parameter, $\zeta$, defined as,

$$\zeta = \frac{z}{L} \tag{3}$$

where $z$ is the height above the surface. Negative values of the stability parameter imply unstable stratification whereas positive values indicate stable conditions. When attempting to classify the stability near the surface, $\zeta$ has the advantage compared to the Richardson number in that only one level of observations is required. In this study, we define 6 stability classes to characterise the state of the surface layer. The stability classes are defined and subsequently referred to as: very unstable ($\zeta < -0.1$), weakly unstable ($-0.1 < \zeta < -0.01$), near-neutral unstable ($-0.01 < \zeta < 0$), near-neutral stable ($0 < \zeta < 0.01$), weakly stable ($0.01 < \zeta < 0.1$), very stable ($\zeta > 0.1$).

## 2.5 ERA5 reanalysis

To complement the observations, we also use ERA5 reanalysis data (Hersbach et al., 2020). ERA5 is the fifth generation of ECMWF atmospheric reanalysis, and as is the case for all reanalysis, was produced by combining a vast number of observations and a numerical weather prediction model. Due to the involvement of a modelling system, reanalysis cannot be considered as truth in the same manner as an observation can, but it should be considered to be a best-guess of the state of the atmosphere. The numerical model forming the basis of ERA5 is the Integrated Forecast System (IFS) Cycle 41r2. Reanalysis data sets differ notably from operational archives produced by numerical weather prediction models. Although operational models have much higher resolution (typically from 1 to 4 km in contemporary regional numerical weather prediction systems) than global reanalysis data sets, due to frequent model updates, they can not provide consistent long term data sets, required for climatological studies.

ERA5 data covers the time period from 1950 to present with a time resolution of 1 hour and has the nominal spatial resolution of 31 km (or 0.28125 degree longitude and latitude near the equator). Spectral fields in ERA5 are stored with T639 truncation and grid-space fields are stored on N320 reduced Gaussian grid. Here we used data from 1 January 1979 to 31 December 2019, re-gridded to a regular 0.25 degree grid. BL height is provided as a diagnostic (see section 3.3) and we extract the BL height covering most of Finland and Scandinavia. Most of the analysis is based on BL height from the grid cell nearest to the Hyytiälä SMEAR II station which, in ERA5, has an altitude of 139 m a.s.l and has a land fraction of 0.91. In addition, we also extract a

limited number of vertical profiles of potential temperature, relative humidity and the zonal ($u$) and meridional ($v$) wind speed components from which the scalar wind speed is calculated. These profiles are extracted on the native model levels, of which there are 137, with vertical grid spacing ranging from 20 to 100 m in the lowest 1 km.

## 3 Methods to estimate boundary-layer height

In this section, the methods applied to the radiosonde, microwave radiometer and reanalysis data sets to quantify the BL height are described. We only use BL height estimates from pre-existing methods, that are used by others already, and we do not develop new methods to identify the BL height nor attempt to improve pre-existing methods. Although all methods diagnose a BL height, due to their different approaches, they do not all consider exactly the same physical processes. For example, some methods only consider potential temperature and hence buoyancy-driven turbulence whereas others also consider wind profiles and thus shear-induced turbulence. This means that even when applied to the same datasets, we cannot expect the different methods to fully agree. We attempt to highlight the main differences between the methods here and indicate the sensitivities of the different approaches.

### 3.1 BL height from radiosonde soundings

The BL height estimates from the radiosonde soundings are taken directly from the planetary boundary layer height value added product (VAP) developed by the Atmospheric Radiation Measurement (ARM) user facility (Sivaraman et al., 2013). This VAP is routinely produced by ARM for multiple stations. The BL height estimates for the BAECC campaign were downloaded from the ARM user facility portal (https://adc.arm.gov/discover). The vertical structure of the BL is complex and shows notable spatial and temporal variability, which means that a single method, based on a one specific definition of the BL, is not always appropriate for estimating the BL height under different conditions. Therefore, four different methods to estimate the BL height from radiosondes are included in this VAP. The difference between the four methods can be considered as a partial estimate of the uncertainty in the diagnosed BL heights. The four methods are those proposed by Heffter (1980) (referred to as H80), Liu and Liang (2010) (referred to as LL10), and a bulk Richardson number method (Seibert et al. (2000)) using two different critical Richardson number values (0.25 and 0.5) as thresholds to define the top of the BL (referred to as $R_{0.25}$ and $R_{0.50}$). Before the BL height was calculated by ARM, the radiosonde data was sub-sampled onto a uniform 5 hPa vertical grid.

The H80 method determines the BL height using vertical gradients of potential temperature. First the radiosonde input data is further smoothed using a three-point moving average to a vertical resolution of 15 hPa to reduce localised noise and then the vertical gradient (lapse rate) of potential temperature is calculated. Inversion layers (up to 5) are then identified as contiguous heights where the potential temperature lapse rate is greater than 0.005 K m$^{-1}$. The BL height is identified as the lowest inversion layer in which the potential temperature difference between the base and top of the inversion is greater than 2 K. If no such layer is found below 4 km, the BL height is set to the height of the inversion layer with the largest maximum potential temperature gradient.

The H80 method does not account for shear-driven turbulence and therefore is not well suited to situations, such as stable BLs, where wind shear is the main source of turbulence. This limitation is highlighted by the fact that the H80 method was originally only applied during daytime by Heffter (1980). Therefore, we can expect the H80 method to be best suited to convective conditions. The $0.005$ K m$^{-1}$ threshold for the inversion strength is somewhat subjective and means that a strong capping inversion must be present. Delle Monache et al. (2004) noted that the $0.005$ K m$^{-1}$ threshold resulted in unrealistically high BL heights at the ARM Southern Great Plains site in Oklahoma and therefore, they used a threshold of $0.001$ K m$^{-1}$ instead. Similarly, Hayden et al. (1997) also used a lower threshold of $0.002$ K m$^{-1}$ when analysing data from British Columbia. However, as we use the pre-computed ARM VAP, the threshold applied here is $0.005$ K m$^{-1}$ which means the H80 method may estimate deeper BLs, in convective situations, than other methods.

The LL10 method requires the type of the BL to be determined before the BL height is estimated. The BL type (convective, stable or neutral residual layers) is identified using the potential temperature difference between the 5th and 2nd measurement above the surface (i.e. over an 110-145 m deep near-surface layer). This potential temperature difference is compared to a stability threshold, $\delta_s$, which Liu and Liang (2010) identified the value of based on "*visual validation with trial and error*". According to Liu and Liang (2010), $\delta_s$ has a value of 1 K over land (used here) and 0.2 K over ocean and ice. If $\theta_5 - \theta_2 < -\delta_s$, a convective BL is present, if $\theta_5 - \theta_2 > \delta_s$, a stable BL is present, and if $-\delta_d \leq \theta_5 - \theta_2 \leq \delta_s$, a neutral residual layer is declared.

Once the BL type is identified by the LL10 scheme, the BL height is diagnosed. For convective BLs and neutral residual layers, the BL height is defined to be the height at which an air parcel rising adiabatically from 150 m above the surface becomes neutrally buoyant. To determine the level of neutral buoyancy, for each vertical level, the potential temperature difference between this level and the level closest to 150 m above the surface is calculated. An upward scan is performed until the first level when this difference exceeds a instability threshold ($\delta_u = 0.5$) is identified. Starting from this level, a second upward scan and second threshold is then applied to account for over-shooting parcels. The final BL height is therefore the first level above the level of neutral buoyancy where the vertical potential temperature gradient exceeds $0.004$ K m$^{-1}$. The BL height will depend on the values of these thresholds, for example, decreasing $\delta_u$, would result in lower BL heights, and increasing the potential temperature gradient threshold will lead to deeper BLs (Jozef et al., 2022).

If the BL is stable, first the level at which the vertical potential temperature gradient reaches a minimum is identified. If this level is a local peak, defined as the difference in the potential temperature gradient between this level and the level below is less than -40 K km$^{-1}$, or if there is an inversion layer in the two layers above, then this level is defined to be the BL height. In stable conditions, the Liu and Liang (2010) method also checks for the presence of low-level jets as shear-driven turbulence associated with these features can influence the BL height. A low-level jet is identified if the wind speed has a localised maximum which is at least 2 ms$^{-1}$ stronger than the wind speed in the levels above and below. If a low-level jet is identified, the BL height is defined as the height of the wind maximum. If both the criteria based on potential temperature gradients and wind-shear are met, then the BL height is taken as the lower of the two heights. If neither the condition based on stability nor on wind shear are met, the BL height is specified to be a missing value which occurs in less than 2% of the soundings from the BAECC campaign.

The LL10 scheme contains subjective thresholds (e.g. $\delta_s$, $\delta_u$) which will affect the diagnosed BL height and may not work well for all environments (e.g. Jozef et al., 2022). In particular, the classification of the BL type can have a large impact on the diganosed BL height. Zhang et al. (2021) analysed the climatology of the BL type diagnosed from the LL10 scheme at nine ARM observatories. They found that convective BLs are much less common than neutral BLs at all land stations which suggests that $\delta_s$=0.5 may be too large and that some weakly convective BLs are identified as neutral by the LL10 scheme. The BL type identified by the LL10 method is compared to the EC stability class (Fig. S1). Almost all cases identified by the EC data as very or weakly unstable are identified as neutral BLs by the LL10 method. The majority (74%) of very stable cases in the EC stability are identified as stable BLs by the LL10 method but the remaining 26% are identified as neutral BLs. However, it must be noted that the sounding and EC system are not co-located and that the EC system uses data at one level (23 m), whereas the LL10 uses the difference in potential temperature across an $\sim 125$ m layer starting at $\sim 20$ m above the surface, to determine the stability of the surface layer.

The BL height is also diagnosed using the bulk Richardson number, $Ri_b$, which is the ratio of the buoyant and mechanical production terms of the turbulent kinetic energy (TKE) budget equation. At a critical value of the Richardson number, the mechanical production of turbulence is balanced by buoyant consumption and at Richardson numbers equal or larger to the critical value the flow can not sustain turbulence (Stull, 1988). Thus, the BL height is diagnosed to be the height at which $Ri_b$ reaches a critical value $Ri_c$. In the ARM VAP, two critical values, 0.25 and 0.5 are considered.

Different formulations of $Ri_b$ exist. In the ARM VAP, $Ri_b$ is calculated using

$$Ri_b = \left( \frac{gz}{\theta_{v0}} \right) \left( \frac{\theta_{vz} - \theta_{v0}}{u_z^2 + v_z^2} \right) \tag{4}$$

where $g$ is the acceleration due to gravity, $\theta_{v0}$ and $\theta_{vz}$ are the virtual potential temperature at the surface and height $z$, respectively and $u_z$ and $v_z$ are the horizontal wind speed components at height $z$. Unlike other formulations, no friction velocity term (e.g. Vogelezang and Holtslag, 1996) nor excess temperature term (Troen and Mahrt, 1986) is included. The horizontal wind speeds at the surface are set to be zero, which Seidel et al. (2012) show leads to differences in the diagnosed BL height of between 50 - 200 m compared to when wind speeds measured at 2 m are used. Richardson number methods are also known to be sensitive to the choice of surface temperature (Beyrich and Leps, 2012). From the radiosonde observations, $\theta_{v0}$ is calculated using the lowest measurement which, according to the ARM documentation, is usually at a height of 2 m. The BL heights identified from the Richardson number method are also sensitive to the choice of critical Richardson number. Theoretically, it is clear that a larger critical value will result in deeper BLs. Both Beyrich and Leps (2012) and Seidel et al. (2012) quantified the sensitivity of the BL height to the choice of $Ri_c$. Beyrich and Leps (2012) showed that the BL height increased by 1% for every 0.05 increase in $Ri_c$ and Seidel et al. (2012) concluded that the sensitivity to $Ri_c$ was the smallest of the 4 sources of uncertainity that they considered.

## 3.2 BL height from microwave radiometer measurements

In this study we use the BL height estimates provided directly from the MWR manufacturer's software (RPG Radiometer Physics GmbH, 2014b). The MWR provides an estimate of the BL height based solely on the vertical profile of potential

temperature as no vertical profiles of wind speed are available from the MWR. The potential temperature ($\theta$) is calculated from the retrieved temperature profile at each altitude, $z$, using

$$\theta(z) = T(z) \left( \frac{p_0}{p(z)} \right)^{R/c_p} \tag{5}$$

where $T(z)$ is the temperature at altitude $z$, $p(z)$ is the pressure at altitude $z$ estimated from the barometric formula for the standard atmosphere, $p_0$ is a reference pressure, $R$ is the ideal gas constant, and $c_p$ is the specific heat capacity at constant pressure. The potential temperature profile is then used to determine whether the BL is unstable or stable. The BL is classified as unstable if $\theta(z) < \theta(z_1)$ for any $z > z_1$ i.e. if potential temperature decreases with height. In this case, the BL height $z_{\mathrm{BLH}}$ is defined through the parcel method (Holzworth, 1964) as the height at which the potential temperature is equal to the "surface"

potential temperature, i.e. $\theta(z_{\mathrm{BLH}}) = \theta(z_1)$ where $\theta(z1)$ is the temperature measured at a height of 1.5 m a.g.l. by the Vaisala WXT 530 weather station incorporated into the MWR setup. Thus, for unstable cases the MWR defines the BL height as the level of neutral buoyancy and, unlike the H80 and LL10 methods, does not attempt to account for over-shooting thermals. Otherwise, the BL is classified as stable, and the BL height is defined to be the height at which the vertical derivative of the potential temperature, $\theta'$, has a localised minimum. If $\theta'(z)$ has no minimum, then $z_{\mathrm{BLH}} = 0$. Very similar methods have been

applied to MWR temperature profiles by Collaud Coen et al. (2014) and Moreira et al. (2020) to diagnose the BL height.

### 3.3   BL height from ERA5

The BL height diagnostic in ERA5 is required to be valid for all types and depths of boundary layer given the global nature of ERA5. Within ERA5, the BL height is diagnosed online using a Richardson number based method and the BL height is defined as the lowest model level at which the bulk-Richardson number reaches the critical value of 0.25. As was the case

with the Richardson number method applied to the radiosondes, no friction velocity nor excess surface temperature term is included and the surface wind speeds are assumed to be zero. However, in ERA5, the Richardson number is computed using the virtual dry static energy, $Sv$, rather that the virtual potential temperature as was the case for the radiosondes. The virtual dry static energy is given by $Sv = c_p T_v + gz$, where $T_v$ is the virtual temperature. This is the first of three reasons why a-priori a perfect agreement should not be expected between the BL heights diagnosed with the Richardson number method applied

to the radiosondes and ERA5. The second reason is that the vertical resolution of ERA5 is lower than the radiosondes, and therefore, sharp features such as strong inversions or low-level jets will likely be weaker and smoother in ERA5 than in the radiosonde observations. The third reason is what level the "surface" values of virtual potential temperature / virtual dry static energy are taken from. In both the radiosondes and ERA5, this is the lowest level. However, in ERA5 this is typically around 10 m a.g.l. whereas in the radiosonde soundings this is much lower - typically 2 m a.g.l. Additional details of how the BL

height is computed in ERA5 are in the IFS documentation (ECMWF, 2015).

## 4  Case studies

In this section we show examples of the radiosonde and MWR data and how the diagnosed BL height relates to the observed vertical profiles of meteorological variables. In addition, for each presented case study, we compare the observations to ERA5 data. A systematic, statistical comparison is presented in section 5.

### 4.1  BL height diagnosed from Radiosondes

An example of vertical profiles of potential temperature and wind speed measured by radiosondes is illustrated in Fig. 2. In addition, the BL height diagnosed by the 4 methods described in section 3 and by ERA5 are also indicated in Fig. 2. This specific case study, covering the 7-8 May 2014, includes 5 radiosonde ascents and highlights the typical diurnal evolution of the BL height and the sensitivity of the diagnosed BL height to the method.

At 00 UTC on 7 May, the potential temperature at the surface is below $0°$C and increases rapidly with height within the lowest 50 meters indicating a stably stratified BL. The EC surface-layer stability class is very stable at this time ($\zeta > 0.1$) and the LL10 method also indicates a stable BL. The BL height estimates from the 4 methods applied to the radiosonde profiles range from 5 m ($R_{0.25}$ method) to 128 m (LL10 method). ERA5 diagnoses a deeper BL than the Richardson number methods applied to the radiosondes most likely because ERA5 does not resolve the strong, shallow surface-based inversion and because, unlike the soundings, ERA5 has a low-level jet present which will generate shear-driven turbulence.

At 06 UTC, the surface has warmed and the strong surface-based stable layer has started to mix out. The EC stability class is now weakly unstable ($-0.1 < \zeta < -0.01$) and the LL10 method diagnoses a neutral BL. Notable differences in the diagnosed BL height are evident at this time; the H80 method identifies the BL top to be co-located with the inversion at the top of the residual layer and thus indicates a BL height of 2289 m whereas the remaining three methods have heights from 69 m to 324 m. The large difference between the H80 and the other methods occurs as the H80 method requires a very strong vertical potential temperature gradient and therefore does not identify the weak inversion at 300 m as the BL top. This over-estimation is in agreement with the results of Hayden et al. (1997) and Delle Monache et al. (2004) (see section 3). ERA5 diagnoses a deeper BL than all methods except H80. This is because the inversion in ERA5 is higher and also smoother than in the observations.

By 12 UTC the BL is well mixed with potential temperature almost constant with height until 2.6 km where a temperature inversion is present. This is consistent with the very unstable EC stability class ($\zeta < -0.1$) present at this time, yet the LL10 method diagnose a neutral BL. The diagnosed BL heights again show a large variation ranging from 894 m ($R_{0.25}$) to 2614 m (H80) and, based on visual analysis, it appears that both Richardson number-based methods underestimate the height of the convective, well-mixed layer. The ERA5 BL height is much higher than diagnosed from the radiosondes using the $R_{0.25}$ and $R_{0.5}$ methods. This is likely because ERA5 has a warmer surface temperature than observed as well as a much smoother potential temperature profile.

At 18 UTC, the surface has started to cool and a weak stable layer has begun to develop between the surface and 500 m. The EC measurements – located 200 m away – identify the surface layer to be weakly unstable ($-0.1 < \zeta < -0.01$) and then near-neutral unstable ($-0.01 < \zeta < 0$) during the 1 hr period from 18 - 19 UTC. In addition, a low-level jet is beginning to develop.

This complex situation results in divergent estimates of the BL height. The H80 and LL10 methods maintain a deep BL whereas both Richardson number methods diagnose shallower BLs, the top of which is just above the low-level jet. The H80 method produces deeper BLs than the $Ri$ methods as the surface-based inversion is too weak compared to the potential temperature gradient threshold (0.005 K m$^{-1}$) required by the H80 method to be identified as the BL top. The Richardson number methods diagnose shallower BLs than the LL10 method due to the differences in the "surface" temperature. The weakly stable layer between the surface and 500 m means that the surface temperature used to estimate the BL height by the $Ri$ method is cooler than that used in the LL10 method which are taken from the first level above 150 m. This results in less buoyancy in the $Ri$ methods and shallower BLs. In this case, the LL10 method identifies the BL type to be neutral as although the potential temperature does increase with height near the surface, the increase is less than 1 K (the required minimum inversion strength above the convective BL top or below the stable top (Liu and Liang, 2010)). Therefore, the LL10 scheme does not consider the wind speed profile and does not detect the low-level jet. If the LL10 scheme had classified this sounding as stable, the BL height would have been identified at the height of the low-level jet maximum (437 m) and thus much lower than it actually was (2513 m). ERA5 diagnoses a shallower BL than any of the methods applied to the radiosonde because a much more stable layer has developed at the surface resulting in cooler surface temperatures. This, along with the slightly stronger low-level jet in ERA5, suggests that the nocturnal transition may occur quicker in ERA5 than in reality.

By 00 UTC on 8 May, the surface has continued cooled resulting in a strong stable surface layer (the stability class is very stable at this time and also the LL10 method determines the BL to be stable now) and a strong low-level jet is present. All methods diagnose the BL height to be shallow with heights ranging from 56 m (H80) to 160 m ($R_{0.5}$). In contrast to 18 UTC, ERA5 now diagnoses deeper BLs than any of the radiosonde methods as the very shallow surface inversion is now weaker than observed. This is likely due to the limited number of vertical levels near the surface in ERA5 (only 4 levels below 100 m).

## 4.2 BL height diagnosed from Microwave radiometer measurements

Next, we show an example of the temperature profiles and BL height derived from the microwave radiometer measurements and compare these to ERA5. Figure 3 shows the MWR and ERA5 temperature profiles along with a cloud profiling product generated from cloud radar and lidar measurements at the SMEAR II site (CLU, 2019), and the EC stability class timeseries on 16 June 2019. This day had no precipitation measured by the ground instruments that would affect the quality of the MWR measurements.

The temperature profile and diurnal evolution in ERA5 is in good agreement with the temperature measured from the MWR (Fig 3a-b). Some differences are present, for example, ERA5 shows a temperature inversion between 1.5 - 2 km a.g.l. at 12 - 16 UTC that is not present in the MWR temperature profile.

The BL heights diagnosed by the MWR and ERA5 show a strong diurnal cycle. A shallow BL is identified from 00 - 04 UTC (01:40 - 5:40 local solar time; 3am - 7am local legal time), after which the BL grows quickly in depth to 1 - 1.5 km a.g.l. A similar evolution of the BL height can be seen in the cloud classification product (Fig. 3c), where the growth of the BL can be seen by the increasing height of the "Aerosol & insects" and "Insects" categories. After 10 UTC (11:40 local solar time, 1pm local legal time), the MWR BL height mostly follows the 10 °C isotherm at an altitude of 1 km a.g.l, while the BL height

from ERA5 is higher, typically at 1.5 km a.g.l.. The BL height in the cloud classification product is between 0.8 - 1.2 km a.g.l., indicated approximately by the top of the "Insects" category (Chandra et al., 2010; Franck et al., 2021), and resembles the

415 MWR BL height evolution even though the MWR estimates the BL height slightly higher. This suggests that between 10 - 17 UTC, ERA5 overestimates the BL height. This is likely caused by the different methods applied to the two different datasets; ERA5 uses a Richardson number method with a critical value of 0.25 which by definition will give deeper BL heights than the parcel method applied to the MWR. At 17 UTC (18:40 local solar time, 8pm local legal time), the BL height determined from both the MWR and ERA5 show the rapid collapse of the convective BL into a stable BL, and between 17 - 22 UTC the BL

height estimated in ERA5 is generally in good agreement with the measurements from the MWR. However, after 22 UTC, the BL identified by the MWR is more than 1 km deeper than the BL diagnosed by ERA5 and furthermore, the MWR estimates of BL height vary significantly during the night.

The MWR also identifies whether the BL is stable or unstable (indicated by filled/unfilled markers in Fig 3a-b). During daytime, the BL is identified by the MWR as unstable, which is in good agreement with the stability class identified from the

425 EC observations (Fig. 3d) which shows mainly weakly unstable conditions. During nighttime, the MWR determines the BL to be mostly stable, although a few points are identified as unstable. Again, this agrees well with the EC stability class which shows weakly stable conditions at night (Fig. 3d).

The MWR BL height values identified as unstable agree well with the ERA5 BL height values. However, for the stable values that occur at night, the agreement is poor. The lack of agreement, and that the MWR estimates BL heights greater than

430 1 km when both the EC and MWR show stable conditions are present, suggests that the MWR does not provide accurate BL height estimates in stable conditions. This point is further investigated in Section 5.3.

## 5 Systematic comparison of BL height diagnosed from different methods

In this section, a systematic, statistical comparison between the different estimates of BL height is presented. First, the four BL height estimates from the radiosonde data are compared to each other and then to ERA5 BL height estimates. Second, the

435 diagnosed BL height from the MWR is compared to ERA5. In particular, we aim to determine how well the BL height in ERA5 represents observations and identify whether the level of agreement between observations and ERA5 depends on the observed surface-layer stability. To quantify the level of agreement between the observations and ERA5, we first calculate the Pearson's correlation coefficients ($r$, Chang and Hanna (2004)). In addition, to quantify any systematic bias between the two estimates of BL height the fractional bias (FB) is calculated using

$$FB = \frac{\overline{h_{obs}} - \overline{h_{era5}}}{0.5 \left( \overline{h_{obs}} + \overline{h_{era5}} \right)} \qquad (6)$$

where $h_{obs}$ is the BL height estimate from observations (either the MWR or radiosondes) and $h_{era5}$ is the BL height from ERA5. FB varies between -2 and +2 and has a perfect value of zero. If FB is positive, the observations have deeper BLs than ERA5 and vice-versa, if FB is negative, observations have shallower BLs than ERA5. As a reference, $|FB| = 0.67$ and $|FB| = 0.4$ indicates that $\overline{h_{obs}}$ and $\overline{h_{era5}}$ differ by a factor of two and a factor of 1.5, respectively.

The normalised root mean square deviation (nRMSD) is also calculated following Equation 3 in Chang and Hanna (2004):

$$nRMSD = \sqrt{\frac{\overline{(\overline{h_{obs}} - \overline{h_{era5}})^2}}{\overline{h_{obs}} \cdot \overline{h_{era5}}}}. \tag{7}$$

nRMSD is always positive and a value of zero would indicate a perfect fit between the observations and ERA5. In contrast to FB, the nRMSD quantifies all sources of error (random and systematic) and the normalisation makes the values non-dimensional, and thus, this is a measure of relative error, not absolute error. The term RMSD is used, rather than the more common root mean square error, as the term "error" implies one estimate is correct and the other is not. This is not the case here and we solely aim to quantify the difference between the different methods.

## 5.1 Comparison of the 4 Radiosondes methods

Figure 4 compares the diagnosed BL height from the 4 radiosonde methods for all times in the BAECC campaign period (1st February 2014 - 13th September 2014) to each other. The corresponding statistical values ($r$, FB and nRMSD) are provided in Table 1. The Pearson's correlation coefficient for each comparison is positive and statistically significant at the 95% level (Table 1). Given that all methods use the same input data, positive correlations were expected, but given the differences between the methods, it was not clear a-priori if the correlations would be statistically significant. However, all panels in Figure 4 show considerable scatter indicating that the diagnosed BL height can depend strongly on which method is applied.

Most notable in Figure 4 is that the BL heights diagnosed by the H80 method differ the most from the other three methods. This is quantitatively demonstrated by the $r$ values which are in the range 0.542 to 0.624 and are lower than the correlation coefficients between the LL10 method and the two Richardson number methods (0.70 and 0.72). The FB values calculated between the H80 method and all other methods are positive and exceed 0.5. This means that the BL heights diagnosed by the H80 method are consistently larger than those from the other three methods. This is consistent with the case study shown in Figure 2 and is in agreement with Seidel et al. (2010) who found that methods which use the vertical gradient of potential temperature to identify the BL height (as H80 does), produce the larger values of BL height than other methods. Closer analysis of our results reveals that in the case of deep (> 1km) BLs, the H80 method almost always over-estimates the height of the BL compared to the LL10 method and both Richardson number methods. This is very likely because of the strong potential temperature gradient and temperature increase across across the inversion required by the H80 method, and, because the H80 method was designed to incorporate turbulent mixing caused by buoyant thermals overshooting their level of neutral buoyancy. When very shallow (< 100 m) BLs are considered, the H80 method estimates shallower BLs than the LL10 method (Fig. 4a) but deeper BLs than the $R_{0.25}$ method (Fig. 4c). Furthermore, when the H80 method diagnoses BL heights between 500 - 1000m, there is good agreement with the $R_{0.25}$ method. This shows that, although the bias of the H80 method is on average positive, how well the diagnosed BL heights compare to those from other methods does depend on the height and therefore, potentially, the stability of the BL.

The LL10 method agrees relatively well with both Richardson number approaches (Fig. 4b, d) and in both cases $r$ exceeds 0.7. In comparison to the $R_{0.25}$ method, the LL10 method diagnoses slightly deeper BLs on average (FB=0.186) and this positive bias is particularly pronounced for deeper BLs. In contrast, when the LL10 scheme diagnoses BL heights in the range

200 - 1000 m, the BL height estimate from the $R_{0.25}$ method is deeper. The FB between the LL10 and $R_{0.5}$ method is small and negative indicating that the LL10 method diagnoses slightly shallower BLs than the $R_{0.5}$ method. The colours in Figures 4a, b and d show the BL type identified by the LL10 scheme (Fig. S2 is the same as Fig. 4 except that the colours show the stability class identified from the EC measurements). Neutral and convective BLs (red) are systematically deeper than the stable BLs (blue). In this study, only 2 out of 805 soundings were defined to be convective, whereas 597 (74%) where defined as neutral and 206 (26%) as stable by the LL10 scheme. This differs from the surface stability class estimated from eddy covariance data which, if only February to October is included to ensure a fair comparison with the radiosondes, shows that 38% of times are classed as very unstable or weakly unstable, 19% of times are near-neutral stable or near-neutral unstable and 43% are weakly stable or strongly stable.

Of all the comparisons, in terms of the FB and nRMSD, the best agreement is between the LL10 and $R_{0.5}$ methods whereas the best correlation (r=0.96) is between the $R_{0.25}$ method and the $R_{0.5}$ method (Fig. 4f). The $R_{0.25}$ method gives systematically lower estimates of the BL height than the $R_{0.5}$ method (FB of -0.261) which is to be expected given that the Richardson number generally increases in height as the amount of turbulence decreases.

To further understand the differences between the 4 different radiosonde methods, we now consider the distributions of diagnosed BL height for each synoptic observation time (Fig. 5) and for different stability classes (Fig. 6). The shallowest BLs in all four methods are diagnosed at 00 UTC and the deepest at 12 UTC. At 00 UTC, the median value diagnosed by the H80 method is very similar to the $R_{0.25}$ method and smaller than the median values diagnosed by the LL10 and the $R_{0.5}$ methods. This is not the case at 06, 12 and 18 UTC where, similar to shown in Fig. 4, the H80 method has consistently deeper BLs. Figure 5 also indicates that the LL10 method has a narrower distribution of diagnosed BL heights at 00 UTC and especially at 06 UTC compared to the other 3 methods.

Analysing the relation between the BL heights diagnosed from the 4 different methods based on observed surface-layer stability allows a more physically-based understanding than by analysing this in terms of time of day. For very unstable conditions ($\zeta \leq$ -0.1, Fig. 6a), the median BL height values vary considerably, from $\sim$600 m to over 1500 m, between the 4 different methods. The shallowest BLs, in terms of the mean, median and 3rd quartile are diagnosed by the $R_{0.25}$ method closely followed by the $R_{0.5}$ method which diagnoses slightly deeper BLs. The LL10 scheme and especially the H80 method produce the deepest BLs when the surface layer is very unstable. In general, similar behaviour between the 4 schemes is observed for weakly unstable and near-neutral unstable conditions. The biggest difference is that the LL10 scheme is in better agreement with the Richardson number methods as the surface layer approaches neutral stratification.

For very stable conditions ($\zeta \geq$ 0.1, Fig. 6b), all methods diagnose very shallow BLs and the absolute differences are small; the largest median value (357.9 m) is from the LL10 scheme and the smallest median value (236.0 m) is from the $R_{0.25}$ method. The H80 method (median value 254.3 m) has the largest number of outliers, and given this method occasionally diagnosed BL heights greater than 1 km when the eddy covariance observations show the surface layer is very stable, it suggests that what is identified as the BL top is likely the top of a residual layer. When weakly stable and near-neutral stable conditions are considered, the H80 method and the $R_{0.5}$ method have the deepest BLs in terms of the median value. Marsik et al. (1995) state that the H80 method can overestimate the BL height when there is a surface-based inversion as the BL height is defined to be

2 K above the top of the inversion. The LL10 scheme has the shallowest BLs and the narrowest distributions. The majority of the cases identified as weakly stable or near-neutral stable by the EC measurements are identified as neutral by the LL10

scheme (Fig. S1). Therefore, in these cases, the BL height is defined as a level of neutral buoyancy, which if the near surface layer is stable, will be very close to the surface. Hence, the mis-classification of the BL type by the LL10 scheme may explain the shallow BLs identified in weakly stable and near-neutral stable conditions. Thus, in conclusion the H80 method produced the deepest BLs for all stability classes except for the very stable case. How the LL10 scheme compares to the two Richardson number methods depends strongly on stability class with the LL10 scheme diagnosing deeper BLs for unstable cases and

shallower BLs for stable conditions.

## 5.2 Comparison of the radiosondes methods to ERA5

Figure 5 also compares each of the 4 radiosonde BL height estimates to ERA5 for 00, 06, 12 and 18 UTC and a quantitative comparison is presented in Table 2. Given that ERA5 uses a Richardson number approach to diagnose BL height, we would expect the best agreement between ERA5 and the Richardson number methods applied to radiosonde data.

At 00 and 06 UTC the best agreement is obtained between ERA5 and the $R_{0.5}$ method. This is confirmed by high correlation coefficients and very small nRMSD and FB values (Table 2). Good agreement is also found between the $R_{0.25}$ method and ERA5 at 00 and 06 UTC although the FB is negative demonstrating that, on average, ERA5 diagnoses systematically deeper BLs than the $R_{0.25}$ method at night. Interesting, the distribution of BL heights at 00 UTC is broader in both Richardson number methods applied to the radiosondes than ERA5 suggesting that ERA5 lacks variability at 00 UTC. Poor agreement

exists between both the LL10 and H80 methods and ERA5 at 00 and 06 UTC (Table 2). A positive FB is found between ERA5 and the H80 method at 00 and 06 UTC indicating that, on average, the H80 method diagnoses deeper BLs than ERA5, and in particular much deeper BLs at 06 UTC than ERA5 (FB value of 0.581). In contrast, a negative FB occurs between the LL10 method and ERA5 at 00 and 06 UTC.

At 12 UTC, the ERA5 BL height distribution agrees well with the two Richardson number methods and, now also with the

535 LL10 method (Fig. 5); statistically the best agreement at 12 UTC is between the LL10 method and ERA5. Poor agreement remains between ERA5 and the H80 method, although a larger correlation coefficient now exists than at 00 and 06 UTC (Table 2). At 12 UTC, both the $R_{0.5}$ and LL10 methods have small but positive FB meaning that ERA5 estimates slightly shallower BLs than these two methods at 12 UTC. In contrast, ERA5 estimates deeper BLs than the $R_{0.25}$ method. At 18 UTC, good agreement occurs between ERA5 and both Richardson number methods with the best agreement between ERA5 and the $R_{0.25}$

method. However, at 18 UTC, the BL heights diagnosed from both the H80 and LL10 scheme agree very poorly with ERA5 estimates (correlation coefficients less than 0.01) and the FB values are very large and positive; the H80 method has a FB of 1.087 which corresponds to an over-estimation of BL height by more than a factor of 3 compared to ERA5.

The poor agreement at all synoptic times between ERA5 and H80, and at 00, 06 and 18 UTC between ERA5 and LL10, is almost certainly due to the differences in the methods and essentially an unfair comparison. The H80 method does not consider

shear-driven turbulence and is only well suited to unstable, well-mixed BLs. This explains the particularly poor agreement at night and also that the best agreement between ERA5 and H80 does occur at 12 UTC when well-mixed BLs are more common.

At 00, 06 and 12 UTC, ERA5 overestimates the BL height in comparison to the $R_{0.25}$ radiosonde method and agrees better with the $R_{0.5}$ method applied to radiosonde data than the $R_{0.25}$ method even though ERA5 has a critical Richardson number of 0.25. One potential reason that ERA5 estimates deeper BL than the $R_{0.25}$ method is the difference in vertical resolution which is coarser in ERA5. A second potential reason is due to differences in the surface temperature, to which the Richardson number method is known to be sensitive to. At 18 UTC, the time of the evening transition, ERA5 agrees better with $R_{0.25}$ than $R_{0.5}$ which overestimates the BL height. This may be because the collapse of the well-mixed BL occurs too quickly in ERA5.

Figure 6 compares the different radiosonde methods to ERA5 but for different stability classes, identified using the EC observations (section 2.4). For the most unstable BLs, the largest correlation coefficient and smallest FB exists between the $R_{0.5}$ method and ERA5. As ERA5 uses a critical Richardson number of 0.25, theoretically, ERA5 should have shallower BLs than the $R_{0.5}$ method applied to the radiosondes. However, the lower vertical resolution of ERA5 compared to the soundings means that in very unstable conditions the capping inversion at the top of the BL may be weaker (more smoothed out) in ERA5 than in the sounding (which is the case at 12 and 18 UTC in Fig. 2). This would cause a deeper BL in ERA5 than in the sounding if the same critical Richardson number was used. This may also explain why ERA5 has deeper BLs compared to the $R_{0.25}$ method applied to radiosondes (negative FB, Table 3). For very unstable conditions, where buoyancy-driven turbulence dominates, and therefore, when the comparison between the H80 method and ERA5 is fairer, ERA5 diagnoses much shallower (almost by a factor of 2) BLs than the H80 method and has a much smaller IQRs and thus much less variability in the diagnosed depth of the BL than H80. ERA5 also diagnoses shallower BLs than the LL10 method (by 23%) for very unstable conditions. For weakly unstable and near-neutral unstable BLs, (Fig. 6a) the best, and very good, agreement exists between the $R_{0.25}$ method and ERA5 (Table 3). ERA5 predicts shallower BLs than all 4 radiosonde methods (the FB is positive for all methods - Table 3): the bias is very small for the $R_{0.25}$ method, moderate for the $R_{0.5}$ and LL10 methods and very large for H80.

For near-neutral stable and weakly stable BLs (Fig. 6b), again ERA5 agrees best with the $R_{0.25}$ method. The correlation coefficient is high (0.793) and the FB is very close to zero. ERA5 diagnoses shallower BLs than the $R_{0.5}$ method, and much shallower BLs than the H80 method, for near-neutral stable and weakly stable BLs – similar to what was found for near-neutral unstable and weakly unstable BLs. In contrast, ERA5 predicts deeper BLs (negative FB) compared to those from the LL10 scheme for near-neutral stable and weakly stable BLs which is opposite of what was found for the three unstable classes. Furthermore, the LL10 scheme and ERA5 have very small correlation coefficients that are not statistically significant which means that the LL10 scheme and ERA5 are not in good agreement for near-neutral stable and weakly stable BLs. The much shallower near-neutral stable and weakly stable BLs in LL10 than ERA5 is likely caused by how shear-driven turbulence is considered. Only 7 out of 89 cases that the EC stability class defined as near-neutral stable are identified as stable by the LL10 scheme (Fig. S1a) and thus in 81 cases, the LL10 scheme does not consider the wind profile and hence estimates very shallow BLs.

For the most stable BLs, all schemes and ERA5 diagnose the shallowest BLs of all stability classes. ERA5 has a larger IQR and thus more variability than the 4 radiosonde methods (Fig. 6b), however the number of outliers for all radiosonde methods is greater than in all other stability classes. For these very stable cases, there is a large negative FB between ERA5 and both Richardson number methods indicating that ERA5 significantly over-estimates the BL height. In particular, the FB between the

$R_{0.25}$ method and ERA5 is -0.971 meaning that, on average, BLs in ERA5 are more than twice as deep as detected from the $R_{0.25}$ method applied to the radiosonde observations. However, despite the systematic bias and the known challenges models have representing stable BLs correctly, there is still a reasonable agreement between ERA5 and the two Richardson number methods — both have correlation coefficients of ~0.6. The mean BL height predicted by ERA5 is smaller than that from the H80 method, and the FB also indicates that ERA5 has shallower BLs than the H80 method for the very stable class but this result is strongly influenced by the number of outliers. A small, although still statistically significant, correlation coefficient exists between H80 and ERA5 indicating a poor level of agreement overall. Similarly, a very small correlation coefficient exists between the LL10 scheme and ERA5 (as was the case for the other stable classes) indicating that in very stable conditions the LL10 scheme diagnoses the BL height very differently to ERA5.

In summary, when ERA5 is compared to the Richardson number methods, we conclude that ERA5 accurately predicts the BL height in the majority of - but not all - situations. The most notable exceptions are that ERA5 significantly overestimates the height of stable BLs and underestimates the BL height at 18 UTC. This suggests that ERA5 lacks the vertical resolution to fully resolve very shallow stable BLs and that ERA5 likely does not capture the evening transition well. However, as long as these two limitations are considered, ERA5 can be used as a basis for a long-term climatology of BL height at Hyytiälä.

### 5.3 Microwave Radiometer and ERA5

To compare the MWR BL heights, reported approximately every 10 minutes, with the ERA5 values output as hourly values, a 1-hour median of the MWR BL height values was used. Additionally, as was the case for the radiosonde comparison to ERA5, the surface stability classes derived from the eddy covariance observations are used to bin the MWR data and to determine the effect of the surface-layer stability on the agreement between MWR and ERA5 BL height values.

Figure 7a shows the distributions of both the MWR and ERA5 BL height values, categorised according to the surface stability class. For the unstable surface stability classes, the distributions agree well. Interestingly, the correlation coefficient in the very unstable class is lower than for the weakly unstable and near-neutral unstable surface stability classes (Table 4). The highest correlation (0.65) and smallest nRMSD (0.016) was achieved in the weakly unstable class (Table 4). The FB is small and positive for the most unstable class, almost zero for the weakly unstable class, and small and negative for the near-neutral unstable class (Table 4). This means that for very unstable BLs, the MWR diagnoses deeper BLs than ERA5, but for less unstable BLs, the MWR has shallower BLs than ERA5. In unstable conditions the MWR uses the parcel method to diagnose BL height whereas ERA5 uses a Richardson number method. Theoretically, parcel methods should give shallower BLs than the Richardson number method and this has been confirmed in previous studies (e.g. Lotteraner and Piringer, 2016). This difference in the methodologies could explain why ERA5 has deeper BLs than the MWR in near-neutral unstable conditions but not why the ERA5 has shallower BLs than the MWR for very unstable conditions. The difference in very unstable conditions may be related to the strength of any surface-based super-adiabatic layers and the surface temperature. If the surface temperature is warmer in the MWR observations than ERA5, this would explain why the MWR diagnoses deeper BLs.

When moving towards stable conditions, the level of agreement between the MWR BL heights and ERA5 decreases and the number of outliers increases, especially in the MWR distribution. However, as ERA5 incorporates wind profiles, and hence

shear-driven turbulence when determining the BL height, this disagreement is not too surprising as shear-driven turbulence is more dominant in stable BLs than in unstable BLs. For the near-neutral stable case, the correlation coefficient is only 0.19 and negative correlations exist for the weakly stable and very stable bins (Table 4). The FB is large and negative for the near-neutral stable case indicating that the MWR identified shallower BLs than ERA5 in these situations. The BLs in ERA5 are very likely deeper due to the inclusion of shear-driven turbulence. In contrast, in the very stable stability class, the MWR BL height values are notably larger than the ERA5 BL height values; the third quartile value of the BL heights in ERA5 is less than the first quartile in the MWR BL heights (Fig. 7a) and the FB is large and positive (Table 4). Furthermore, the MWR distribution is very broad with a large IQR and is much broader than the corresponding distribution from ERA5 (Fig. 7a). This difference between ERA5 and the MWR is the opposite of what would be expected based on the methodological differences.

These statistical results from the very stable class support our findings from the case study (Section 4.2). Firstly, the height of the BL diagnosed by the MWR under stable conditions can be significantly overestimated compared to ERA5. Secondly, the BL heights diagnosed by the MWR under stable conditions are deeper (median value of ∼700 m) than typically expected; Garratt (1992) states that the depth of stable BLs is *"no more than a few hundred metres at most"*. This suggests that the manufacturer algorithm supplied with the MWR may not be well suited to diagnosing the BL height under stable conditions.

To further investigate the limitations in the MWR BL height algorithm, we filter the data so that all BL height values estimated when the MWR algorithm assesses the BL to be stable are removed (Fig. 7b). This filtering removes outliers from the distributions in the stable surface stability classes and improves the level of agreement between ERA5 and the MWR in almost all surface stability classes (Table 4). Notably, now the median value of the BLH in the very stable surface stability class is much more in-line with expectations. In addition, the distribution of the MWR BLH values in the very stable surface stability class is narrow and the distributions of MWR and ERA5 BLH values span approximately the same altitude interval. However, the correlation coefficient in this class, while positive and larger than for the unfiltered dataset, is no longer statistically significant at the 95 % confidence level due to small sample size. The correlation increases also in the unstable surface stability classes, with the highest increase (0.164) occurring in the very unstable class. This suggests that, in addition to over-estimating the BL height in stable conditions, the MWR has also misclassified the stability of some measurements. Additionally, after removing the stable MWR BLH values, the fractional bias is negative in all surface stability classes. That is, the unstable BL height from the MWR observations is consistently lower than in ERA5 which agrees with theoretical expectations and previous studies. However, the high correlation, and the low FB and nRMSD values, for the unstable surface stability classes suggest that although the Richardson number method applied to ERA5 data, and the parcel method applied to MWR data differ, the diagnosed BL heights are consistent.

Figure 8 shows the timeseries of the MWR BL height hourly median values (calculated only from the unstable MWR BL height values) along with the ERA5 BL height values. The surface stability class (determined from EC observations) is indicated by the colour of the MWR circle marker. For clarity, the surface stability classes have been divided only into unstable and stable compound classes, with very, weakly and near-neutral classes combined. The seasonal cycle of the BL height is seen clearly in Figure 8. For both datasets, the BL heights from April until September show high peaks and clear diurnal cycles. Note that the stable BL height values from the MWR have been filtered out, so most of the MWR BL heights during summer nights

are not seen in Figure 8. In the summer (June to August), ERA5 tends to estimate visibly deeper BLs during daytime than the MWR. This is confirmed if we calculate the FB for each month individually (not shown); a negative FB is found in June, July and August between ERA5 and the MWR BL heights. Furthermore, the case study of June 16 (Section 4.2) also showed that ERA5 estimated deeper daytime BLs than the MWR. In winter, the BL height identified by both ERA5 and the MWR is shallow and the variation during the day is much smaller than in summer. Good agreement, especially in December–January, is seen between the MWR and ERA5 BL heights in the winter months

The MWR and ERA5 BL heights display similar diurnal cycles throughout the timeseries, except for during October 2018 where noticeable differences exist both in BL height as well in the diurnal cycle. This disagreement can mostly be accounted for by two factors. Firstly, the temperature profile retrieval can fail. Reasons for this include a wet radome or light drizzle that is not detected by the precipitation sensors used to filter rain cases. Additionally, the retrieval can be affected by inconsistent cloud base temperature retrievals from the infrared radiometer due to scattered cloud cover. This effect can be seen on 1 October 2018 (Fig. S3) and can be identified by the inconsistent (noisy) temperature profile. Note that in this case, even though the precipitation sensors used to remove rain cases did not capture the drizzle, the weather station included in the MWR was able to flag some of the temperature profiles during the case as affected by rain (not shown), but the flag did not appear in the BL height values that were consequently accepted. In this case, the ERA5 BL height follows the insect layer and boundary-layer cloud development seen in the cloud classification product closely and thus performs better.

Secondly, even if the temperature profiles are visually sound, differences in the BL height can occur between the MWR and ERA5 due to differences in the definition of the BL height. An example of this is seen on 28 October (Fig. S4), where the MWR BL height has a clear diurnal cycle with very low values from 02-07 UTC and very large values during the afternoon. In contrast, ERA5 does not have a diurnal cycle and the ERA5 BL height follows the maximum altitude of aerosol echo in the cloud classification product and appears to provides a more realistic estimate of the BL height than the MWR. The MWR may diagnose lower BL heights than ERA5 at night in this case as the BL is weakly stable and therefore may be dominated by shear-driven turbulence. As the MWR only considers the potential temperature profile, its definition of the BL height is not appropriate for these conditions. ERA5, which uses a Richardson number method, does account for shear-driven turbulence and this could explain the deeper BLs in ERA5 in this case. The daytime BL height values from the MWR ($> 1500$m) are much deeper than those from ERA5 and are also very deep for late October (greater than the 95th percentile in the ERA5 climatology presented in section 6). Even though the surface stability indicates an unstable surface layer during the day, it appears that the MWR algorithm is not capable of determining the BL height correctly in this case.

Notably, the surface stability class does not seem to determine whether there is good agreement or not between the MWR and ERA5 during the autumn and winter months (Fig. 8). In October 2018, several of the days where the MWR overestimates the BL height are actually classified as unstable by the EC measurements, and as such, as we have shown previously, the MWR should provide good BL height estimates. And to the contrary, several days with mostly stable surface stability, for example 22-23 November 2018, show good agreement and similar diurnal cycles between the ERA5 and MWR BL height values. However, when comparing the MWR BL height estimates to ERA5, it is important to note that ERA5 is not an observation and cannot be assumed to be perfect and that the MWR and ERA5 definitions of the BL height are different. Therefore, it is

difficult to accurately determine how trustworthy the MWR BL height estimates are under different conditions. We recommend that, in addition to the stability of the boundary layer, local meteorological conditions should be considered carefully and the MWR BL height values should be compared to other data sources when deciding whether they are trustworthy or not.

## 6   Boundary-layer height and surface-layer stability climatology at Hyytiälä

Unfortunately, no long-term measurements of BL height from either radiosondes nor remote sensing instruments are available from Hyytiälä, as is the case for many long-term surface stations. However, from the results presented in section 5, we conclude that ERA5 represents the depth of the BL at Hyytiälä very well except under very stable conditions and thus can be used as the basis for a long-term climatology. To better understand the meteorological causes of the BL height annual and diurnal climatology presented in this section, we also present a climatology of surface-layer stability class as determined from the

eddy covariance observations.

The annual cycle of the BL height in ERA5 is shown in Figure 9a. The shallowest boundary layers occur in winter (December, January, February - DJF), with the lowest monthly median value of 353 m occurring in February, which is the statistically the coldest month in southern Finland; the mean February temperature at Hyytiälä averaged over 1979 to 2019 is -7.4°C. The full distribution of BL height at the 4 standard synoptic times (00, 06, 12 and 18 UTC, Fig. S5) shows that the distribution of

BL heights is positively skewed and has a similar shape at all times. In DJF, it is rare that the BL height exceeds 1 km and very shallow boundary layers (i.e. in the lowest bin 0 - 125 m in the histogram in Figure S5), are less common than moderately shallow boundary layers (i.e those in the 2nd to 4th bins, 125 m - 500 m). To get very shallow BLs, usually very stable conditions are required which needs calm wind conditions, strong radiative cooling at night. and thus clear skies. Analysis of the long-term wind speed measurements at 67.2 m on the mast at SMEAR II show that winter is windier and has less calm conditions than

all other seasons (not shown). A recent study using ceilometer and pyranometer data at Hyytiälä (Ylivinkka et al., 2020) shows that winter also has more cloud than other seasons. Both the increased cloud cover and stronger wind explain why very shallow BLs are less common in winter than moderately shallow BLs; the presence of cloud prevents strong radiative cooling at night and stronger winds create more shear-driven turbulence.

During spring (March, April, May - MAM), there is a rapid increase in the BL height and the deepest BLs of all months

(largest extremes) occur in May (Fig. 9a). This is despite May not being the warmest month of the year; the monthly mean temperature in May at SMEAR II is 9.1°C in May compared to 16.1°C in July. In addition, the variability in the BL height in May is very large. The long-term temperature observations at SMEAR II show that the standard deviation, and the difference between the 90th and 10th percentiles, is larger in May than in all months except DJF (not shown). This means that the temperature is very variable in May and this likely explains the high variability in the BL height. During April and May,

the mean BL height is larger than the median indicating that the BL height distribution is less Gaussian and more positively skewed. This is confirmed by Figure S5 which shows that, particularly at 00 and 18 UTC, the distribution of BL height in MAM is strongly positively skewed and also that the most common BL height is very shallow - less than 125 m.

The highest median value of the BL height occurs in June (576 m) and similarly to May, there is a high degree of variability in the BL height (Fig. 9a). The variability is likely related to the diurnal cycle — all hours of the day are included in Figure 9a. There is also a notable decrease in the median BL height across the three summer months (June-July-August, JJA) which is potentially caused by the pronounced decrease in the length of day and thus the incoming solar radiation between June and August: there is 19 hr 30 min of daylight on 21 June and 15 hr 19 minutes on 21 August. The full distributions of BL height in JJA (Fig. S5) also indicate that the distribution at 12 UTC in JJA is less skewed than in MAM and the peak is shifted to higher values.

In autumn (September, October and November, SON), the median BL height is similar across all of these three months (Fig. 9a), however, the 75th percentile, the extremes (whiskers) and the variability are all smaller in November than in September. Particularly, in October and November, the mean and median BL heights are similar suggesting that the BL height distributions are approximately Gaussian. Figure S5 shows that the BL height distributions in the autumn months are more Gaussian than in spring or summer but that even in autumn the BL height distribution has a small positive skew. Figure S5 also indicates that very shallow BLs are rare in autumn. This is very likely due to the frequent cloudy and windy conditions in autumn.

The annual cycle in the observed surface-layer stability class (Fig. 9b) can also be considered and related to the annual cycle in BL height. To a first order approximation, we would expected deeper BLs to be more unstable than shallow BLs. The surface layer is rarely unstable in winter, due to limited incoming solar radiation and thus weak surface heat fluxes. The occurrence of all three unstable classes (all red colours) is almost constant from April until July whereas August, usually regarded as a summer month, has fewer times that are classified as unstable. The increase in unstable classes starting in April is consistent with the ERA5 BL height shown in Figure 9a which shows April is the month where deep BLs start to become evident. Figure 9b also shows that the very unstable class has a strong annual cycle peaking (18.6%) in July and rarely (<2.1%) occurring between November and February. Similarly, the weakly unstable class also has a pronounced annual cycle but is most common in May (26.6%) which is the same month that the deepest BLs develop. The near-neutral unstable class occurs rarely in all months (the largest occurrence of 10.1% occurs in February) and it has a weak annual cycle.

The three different stable stability classes all have different annual cycles. The near-neutral stable class occurs most frequently in the cold season and least frequently in summer and is more common than the near-neutral unstable class, particularly in November and December when 20.6% and 18.2% of all times are classified as near-neutral stable. The weakly stable class has an annual cycle with two peaks, the largest peak in October (27.8%) and a smaller secondary peak in March (21.1%) and a minimum in May (14.2%). These peaks are caused by the much more frequent occurrence of weakly stable cases at night in March, April, September, and October compared to other months (not shown). Weakly stable conditions likely prevail at night during these months as statistically these months are quite windy so even in the case of radiative cooling, weakly stable BLs likely form instead of very stable BLS. Interestingly, the very stable stability class is most common in the warm season and least common in the cold season. The lack of strongly stable layers in winter is likely due to the prevalence of cloud which would limit surface nocturnal cooling. This hypothesis is supported by the results of Manninen et al. (2018) who showed that most of the nighttime turbulence during winter in Hyytiälä is associated with cloud. A caveat to the annual cycle in stability

class is that the number of missing data points also has a strong annual cycle with many more missing observations during winter than summer, mostly likely related to icing of the sonic anemometer.

The mean diurnal cycle of the BL height for each month is shown in Figure 10a. In all months, the maximum mean BL height occurs at 12 or 13 UTC (13:40 or 14:40 local solar time; 2 or 3pm local legal time depending on season). This shows that the radiosondes released at SMEAR II at 12 UTC most likely do measure the deepest BLs which is not the case for all locations worldwide when the standard synoptic times of 00 and 12 UTC often do not coincide with the deepest BLs. The largest diurnal cycle in mean BL height occurs in May, closely followed by June, and is caused by both the large maximum values during the day and the small minimum values at night. The variation in the diurnal cycle of BL height in May is shown in Figure 11a and is large. This is mainly due to the large diurnal temperature range at this time of year but also may be influenced by cloud cover. Ylivinkka et al. (2020) show that there is a weak diurnal cycle in cloud cover at Hyytiälä in May with less cloud at night which would promote shallow BLs at night. During daytime, there is a large variation in BL heights with maximum values almost reaching 3 km, however, 50% of BLs at 12 UTC in May have a height between 1100 and 1900 m. This large variability is likely caused by the large variability in temperature. The median values are similar to the mean values during daytime, as the distribution is broad and symmetric (Fig. S5), but the mean values are greater than the median BL height at night, as the distribution is highly non-Gaussian and instead resembles a Gamma distribution.

November, December and January have very small diurnal cycles in BL height (Fig. 10a). Figure 11b further emphasises the lack of any diurnal cycle in December as even the variability in the BL height is almost constant with time of day. This lack of diurnal cycle is because in late autumn and winter, days are short, cloudy conditions are common, and given the very high zenith angle there is a very small diurnal cycle in incoming solar radiation. A secondary reason for the lack of diurnal cycle in BL height is that synoptic-scale weather patterns and thermal advection may have a stronger influence on the BL height compared to other times of year. Strong cold-air advection over a warmer surface can lead to strong upwards surface heat fluxes and deep and well mixed BLs (Sinclair et al., 2010) even at night. This process may be particularly relevant in late autumn as strong extra-tropical cyclones are more common in autumn than spring or summer (Laurila et al., 2021). The absence of a diurnal cycle in BL height is in agreement with Manninen et al. (2018) who note that there is almost no diurnal cycle in the occurrence of turbulence during winter at Hyytiälä.

Figure 10a shows that the shallowest night time BLs do not occur in winter and Figure S5 demonstrates that very shallow, night-time BLs occur more often in spring and summer than in winter. This may contradict expectations where it is often assumed that the shallowest BLs develop during the coldest part of the year, however, Beyrich and Leps (2012) also found shallower BLs at night in summer than in winter. Very shallow BLs tend to develop under stable conditions, which usually occur under calm and clear conditions. Such conditions are more common in spring and summer than in winter at Hyytiälä. Spring likely has more shallow BLs than summer as summer nights are short (maximum 19 h 40 min of daylight in Hyytiälä), and hence there is not much time for radiative cooling to take place and for a very stable and shallow BL to develop. Winter likely has more shallow BLs than Autumn due to the fact that ground is snow covered in winter and there is less solar radiation in winter which gives more time for inversions to develop.

Figure 10b shows the mean diurnal cycle of the stability class for all months together. As expected, the very unstable and weakly unstable classes have a pronounced diurnal cycle peaking between 11 - 13 UTC and rarely occurring at night. The very stable and weakly stable classes have the opposite diurnal cycle and it is rare that weakly or strongly stable conditions occur during the day. Noticeably, there is no diurnal cycle in the amount of missing data which may suggest that missing data is not strongly influenced by stability which does have a strong diurnal cycle. Figure 10b considers all months together which very likely distorts some interesting features as different seasons likely exhibit very different behaviour in terms of the diurnal cycle of stability. Therefore, we also consider the diurnal cycle of stability class for May (Fig. 11c) and December separately (Fig. 11d). In May, almost all times with valid measurements between 6 and 16 UTC (07:70 and 17:40 local solar time; 9am and 7pm local legal time) are classed as unstable with the majority falling in the weakly unstable class. Between 21 UTC and 4 UTC, the majority of times are classed as near-neutral stable. In December, there is a more pronounced diurnal cycle in the stability class than in the BL height. The near-neutral stable class has the largest diurnal cycle peaking at night and is rare during the day whereas the very unstable class is exceptionally rare in December. Of note is that at night shallower BLs occur in May than in December despite that the occurrence of the very stable class is slightly larger at night in December in comparison to May.

Overall, the diurnal cycle of ERA5 diagnosed BL heights is largely consistent with the diurnal cycle in the observed stability (i.e deeper BLs occur more often when the surface layer is unstable and shallower BLS are more prevalent when observations indicate a stable surface layer). This, in addition to the good agreement between the ERA5 BL heights and the radiosondes estimates of BL height using the Ri method presented in section 5, indicates that the ERA5 temperature and wind profiles are similar to the sounding profiles and that if the same method is applied to both ERA5 and the radiosonde soundings to identify the BL height, good agreement can be expected.

## 7 Spatial variability of BL height in ERA5

So far we have only considered the BL height from one single grid point from ERA5. However, the BL height is not uniform and varies spatially, for example, due to differences in the properties of the underlying surface caused by differences in land use or the presence of water bodies. The aim of this section is to determine the variability of the BL height in the region surrounding Hyytiälä depending on both month and time of day. Firstly, this enables us to determine over how large an area the now ongoing BL height measurements made with the MWR at Hyytiälä are representative of. Secondly, this also enables us to determine whether the conclusions drawn in sections 5 and 6 are also valid over the surrounding areas.

To assess the spatial variability of the BL height, we use a subset of the global reanalysis data set ERA5. Spatially, we consider a part of Northern Europe (see Fig. 12) and analyse the time period from January 1979 to December 2019 using ERA5 data with 1-hour temporal resolution. This data set thus consists of time series of BL height from many grid points. For each calendar month, we first calculate the Pearson correlation coefficients (averaged over all 42 years) between the time series of BL height extracted at the Hyytiälä location (and extensively analysed in this study) and the time series of the BL height from each of the individual grid cells within the selected area. When high temporal resolution data is considered, high correlations indicate that the BL height at those grid points has a similar temporal evolution to the BL height at Hyytiälä. Here we assume

that larger areas of high spatial correlations indicate lower spatial variability, and conversely smaller areas of high correlations indicate high variability within the BL height field. As a lower limit of 'high' correlation, we use the correlation coefficient of 0.75. For practical applications it may also be useful to know how often the BL height at a given point is considerably different from the BL height at Hyytiälä. Therefore, for each ERA5 grid point in our northern European domain, we also compute the proportion of times that the BL height is within 150 m of the BL height at the ERA5 grid point closest to Hyytiälä.

When all hours of the day are considered, correlations exceeding 0.75 cover most land areas in southern and central Finland (orange contours in Fig. 12). There is a moderate annual cycle with the largest areas of high correlation occurring between April and July and the smallest areas in the winter months. May has the largest area of high correlations which, in addition to southern and central Finland, also extends to some small regions of Sweden and large parts of Estonia. This means that, when all hours of the day are considered, the temporal variability of the BL height is similar over a slightly smaller area in the cold season compared to spring and early summer. The difference in the diurnal cycle, and hence the temporal evolution, of BL height over sea and lakes compared to over land is also evident in Figure 12 as there are no sea areas where the correlation exceeds 0.75. When the percentage of times that the BL height is within 150 m of that at Hyytiälä is considered (shading in Fig. 12), values exceeding 50% cover most of southern and central Finland and the highest values occur closest to Hyytiälä. The largest area with similar BL heights to Hyytiälä occurs in February and the smallest area in June, which reflects the annual cycle in absolute BL height at Hyytiälä (Fig. 9a). These results indicate that in May and June, when many observation campaigns take place in Hyytiälä (e.g. Laakso et al., 2007; Lampilahti et al., 2021), measurements of BL height have a similar temporal variability as BL heights over a relatively large area. However, the area over which the BL height can be assumed to be approximately the same (i.e. within 150 m) as measured at Hyytiälä is relatively small. In contrast, in winter, the area over which the BL height is approximately the same in an absolute sense is much larger than in summer.

The spatial variability of the BL height also depends on the time of day (Figs. S6 – S9). When only BL height values at 00 UTC are considered (Fig. S6), the area with high correlations covers most of southern and central Finland and has a weak annual cycle. In contrast, the area over which the BL height at 00 UTC is within 150 m of that at Hyytiälä is much larger in summer, when nocturnal BLs are shallower, than in winter. At 12 UTC (Fig. S8), the variability of the BL height has a strong annual cycle with highest variability (smallest area of high correlation and smallest area where the BL height difference is within 150 m) during the period from May to September and the lowest variability in the cold season. This corresponds to a reduced spatial representativeness of daytime observations of BL height at Hyytiälä in the summer compared to in the winter. The high degree of spatial variability at 12 UTC in summer could be attributed to deep convective BLs which can be highly variable and are much more strongly influenced by the surface heat flux and thus the surface type and amount of incoming radiation (and hence cloud cover) than stable BLs which are often also influenced by shear driven turbulence.

Although this analysis of BL heights from ERA5 indicates the spatial variability of the BL height to some extent, the results should be taken with utmost care as ERA5 generally underestimate the true variability. This is due to the spatial resolution of ERA5, which is about 31 km, and thus does not allow for accurate representation of small-scale land surface features. These irregularities, such as small lakes or patches of different vegetation affect the evolution of the BL and increase its spatial

variability. A more accurate estimate of spatial variability would require a data set with much higher spatial resolution, however, this is not within the scope of the current study.

## 8   Discussion and Conclusions

In this study, we examined the BL height and surface stability at the SMEAR II station, Hyytiälä in southern Finland. A systematic, statistical comparison between four different pre-existing methods of diagnosing the BL height from radiosonde data and ERA5 data was presented. BL height estimates from a microwave radiometer(MWR) were also presented and compared to ERA5. Due to no time overlap between the radiosondes and the MWR data, these two observation types could not be compared. A unique aspect of our comparison is that we quantified the effect of surface-layer stability on how well different methods agree with each other.

When the 4 different methods applied to radiosonde data were compared, there were positive statistically significant correlations between all methods. However, considerable scatter was present meaning that the diagnosed BL height can depend strongly on the method used. Furthermore, the level of agreement between the 4 methods is also found to depends strongly on the stratification of the surface layer. The BL height diagnosed from the H80 method was consistently deeper than the BL diagnosed from the other three schemes, especially during daytime and in unstable conditions. This is in agreement with previous studies (Seidel et al., 2010; Lotteraner and Piringer, 2016) and is primarily due to the requirement of a strong inversion (the lapse rate must exceed 0.005 K m$^{-1}$). Previous studies (Delle Monache et al., 2004; Hayden et al., 1997) have noted that this threshold is too large and results in over-estimating the depth of the BL. The H80 methods diagnoses the BL height at the height where the potential temperature is 2 K warmer than the base of the inversion. This means that the H80 method is appropriate for diagnosing the maximum potential depth that mixing can occur over. However, the H80 method is not physically robust for stable BLs, as it does not consider shear-driven turbulence, and, in the case of surface-based inversions, the 2 K threshold means that the BL is diagnosed to be too deep (Marsik et al., 1995). Thus, the H80 method is best suited to convective situations. In contrast, the Richardson number method does include shear-driven turbulence and is therefore more applicable to all BL types.

The LL10 method diagnoses deeper BLs than both Richardson number methods for very unstable and weakly unstable cases but shallower BLs for near-neutral and weakly stable BLs. The shallower BLs in stable conditions may arise due to the method that LL10 uses to determine the type of BL and, in particular, to the value of the stability threshold, $\delta_s$, which is 1 K. When Liu and Liang (2010) developed this method and tested it using radiosondes from 14 field campaigns, they found that typically 60% of soundings were classed as neutral whereas 74% of soundings in this study were classed as neutral. When the LL10 BL type is compared to the stability from the eddy covariance, it is apparent that the LL10 neutral category includes many convective cases and some stable cases. Misclassifying the BL as neutral when it is convective has no impact on the diagnosed BL height, whereas misdiagnosing a stable BL as neutral does, as the presence of low-level jets, and thus their associated shear-driven turbulent mixing, is not considered which leads to an under-estimation of the BL height. We suggest that in the

885 future a systematic analysis of the impact of the inversion strength threshold on the diagnosed BL height should be conducted and that this threshold may vary from location to location.

A good degree of correlation ($r > 0.6$) is found between ERA5 and the two Richardson number methods at all synoptic times and for almost all stability classes. Poorer agreement occurs between ERA5 and the H80 and LL10 methods particularly for stable cases, but this is very likely due to the differences in the methods applied. The good agreement between ERA5 and both

Richardson number methods suggests that the high vertical resolution of ERA5 (24 model levels below 1.5 km and grid spacing of 25 - 120 m in the BL) appears to be sufficient to capture the BL structure in most situations. This is a key advance over previous reanalysis such as ERA-Interim which only had 12 levels below 1.5 km. However, for the very stable class, ERA5 estimates deeper BLs than the $R_{0.25}$ method and the correlation is poor. Thus, ERA5 still cannot capture the depth of very stable BLs accurately, which is likely due to deficiencies in the BL parameterization or lack of resolution. It is also notable

that the smallest correlation coefficients between both Richardson number methods and ERA5 occur at 18 UTC. This indicates that defining the BL height during the collapse of the convective BL and transition to more stable conditions is challenging and potentially also that ERA5 struggles to capture this process accurately. A similar result was found by Beyrich and Leps (2012) who show that the largest uncertainty in BL height estimates made from radiosondes at Lindenberg, Germany, occur during the evening transition. Users of ERA5 BL height should be cautious using BL height estimates during the evening transition

period.

Some caveats to the comparison between ERA5 and the radiosonde based estimates of BL height should be noted. First, although the BAECC radiosondes were not assimilated into ERA5, soundings from the nearby operational stations of Jokioinen ($\sim$ 125 km SSW of Hyytiälä) and Jyväskylä ($\sim$ 100 km NW of Hyytiälä) were. Second, the soundings did not cover the full annual cycle meaning that we were unable to verify ERA5 for October - January. However, the stability analysis suggests that

we were able to verify ERA5 for all BL stability types.

ERA5 BL heights were also compared to those from the MWR. Perfect agreement could not be expected since the MWR only considered potential temperature profiles to diagnose the BL whereas ERA5 and the Richardson number method also incorporate wind speed and thus shear-driven turbulence. BL height estimates from the MWR agreed reasonably well with ERA5 but only in certain conditions. For unstable situations when the BL is well-mixed, the MWR derived BL height agrees

well with ERA5. ERA5 does overestimate the BL height compared to MWR during June-August 2018, however, given that ERA5 is a model based data set, it is not clear based on our data whether the MWR also overestimates the "true" BL height. A key outcome of our analysis is that the MWR does not reliably estimate the BL height under stable conditions, which at Hyytiälä, occur commonly at night between April and September. We hypothesise that this is due to the algorithm used by the MWR software under stable conditions and that potentially users could improve on this. Furthermore, some of the errors in the

MWR diagnosed BL height arise as the classification of the stability type (stable vs unstable) fails; checking the accuracy of this classification using alternative observations is recommended. Finally, we identified that the MWR algorithm can also fail to identify the BL height under unstable conditions, as identified from the EC measurements, if there is fog/drizzle or scattered cloud cover. This is particularly true in autumn and winter.

As ERA5 agrees well with observations of BL height, a climatology of the annual and diurnal cycles in BL height determined from ERA5 was computed and presented alongside a climatology of the observed surface-layer stability. The shallowest monthly median BL height (353 m) occurs in February and it is rare that the BL height exceeds 1 km in December-February. Consistent with this is that very unstable conditions only occur around 2% of the time between December and February. Very stable conditions are also quite rare in winter which is very likely due to the cloudy and windy conditions that commonly prevail. Consistent with this, is that in winter in Hyytiälä very shallow BLs are less common than moderately shallow BLs. Also in winter, due to the very small diurnal cycle in incoming solar radiation, there is almost no diurnal variation in the BL height. During spring, the height of the diagnosed BL rapidly increases with the deepest BLs, in terms of the extremes, occurring in May. The variability in BL height is also the largest in May which can be explained by the large degree of variability in temperature in May. The shallowest BLs of anytime occur at night during April – June which is also when very stable and weakly stable conditions are most common. The BL height has a maximum median value in June (576 m) and then decreases during the remains of the summer despite July being the warmest month. This decrease is likely due to the decrease in the length of the day from the maximum in June. In autumn, very shallow BLs are rare, the dominant stability classes are near-neutral stable and weakly stable, and the diurnal cycle of BL height is weaker than spring or summer.

Finally, an estimate of the spatial representativity of the measurements of BL height made at SMEAR II is made based on ERA5 reanalysis data. The analysis shows that, when all hours of the day are considered, the BL height at Hyytiälä is representatitive of most land areas in southern and central Finland. However, the spatial variability of the BL height depends on the time of day especially between May and September. The area which the Hyytiälä BL height values are representative of is much smaller at 12 UTC than 00 UTC and also smaller in summer than in winter.

The results presented here highlight the difficulty in accurately measuring the BL height and demonstrate that to have reliable and accurate estimates in all conditions, a range of measurements is needed. Thus, if BL height estimates are used to better understand surface based measurements of trace gases or aerosol particles, an appreciation of these challenges and knowledge of likely sources of error, and under what conditions they primarily occur, is necessary. Furthermore, this study has shown the large annual, seasonal and spatial variability in the BL height in a high latitude, yet fairly spatially homogeneous and flat location. Lastly, it is encouraging to report that modern reanalysis products with high temporal and spatial resolution can capture the BL height and its evolution well in most situations and, in the absence of observations, can be used with confidence.

*Data availability.* The measurements from the SMEAR II station are available at the SmartSMEAR portal (https://avaa.tdata.fi/web/smart/smear). The observations from the FMI automatic weather station used in this study are available at the FMI open data portal (https://en.ilmatieteenlaitos.fi/open-data). The data from the BAECC campaign are available at the Atmospheric Radiation Measurement (ARM) User Facility portal (https://www.arm.gov/research/campaigns/amf2014baecc). The ERA5 reanalysis data are available at the Copernicus Climate Data Store portal (https://cds.climate.copernicus.eu/). The ECOCLIMAP-SG land cover map is available from https://opensource.umr-cnrm.fr/projects/ecoclimap-sg. The ground-based remote-sensing data (the cloud profiling products shown in Figures 3c, S3c and S4c) used in this article are generated by the European Research Infrastructure for the observation of Aerosol, Clouds and Trace Gases (ACTRIS) and are available from the ACTRIS Data Centre using the following link: https://hdl.handle.net/21.12132/1.d28c281574434f92.

*Author contributions.* VAS jointly conceived the study, wrote the majority of the manuscript, performed the climatological analysis presented in section 6 and contributed to the interpretation of all results. JR performed the analysis and interpretation of the microwave radiometer data and contributed to the writing of sections 3.2, 4.2 and 5.3. GB analysed the eddy covariance data, computed the stability classes and contributed to the interpretation of these results. IS compared the radiosonde BL height estimates to those from ERA5 data. YB designed, performed and interpreted the spatial representativeness analysis presented in section 7 and contributed to writing section 7. DM jointly conceived the study, provided expert knowledge about the microwave radiometer data, and provided guidance on the interpretation of all results. MK selected and analysed the radiosonde case study and contributed to the interpretation of all results. All authors provided comments on drafts of the manuscript

*Competing interests.* The authors declare that they have no conflict of interest.

*Acknowledgements.* This work was partially funded by Academy of Finland Flagship funding (grant no. 337549). We acknowledge the open data policy of the National Land Survey of Finland as the present study uses the following data from the National Land Survey of Finland: Elevation model 10 m, NLS orthophotos, Cadastral index map. We acknowledge ACTRIS for providing the CLU (2019) dataset in this study, which was produced by the Finnish Meteorological Institute, and is available for download from https://cloudnet.fmi.fi/.

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

**Table 1.** Statistical measures between BL height estimates. r is the Pearson's correlation coefficient and nRMSD is the Normalised Root Mean Square Deviation. All correlation coefficient are statistically significant at the 95% level. The best value in each column is highlighted in bold. The variables listed as the x-variables are taken as the observations in Equation 6 and those listed as the y-variables take the place of the model in Equation 6. Thus, if the x-variable has larger BLH than the y-variable, then the FB > 0. If the x-variable has smaller BLH than the y-variable, then the FB < 0.

| x-variable | y-variable | Pearson's r | fractional Bias (FB) | nRMSD |
|---|---|---|---|---|
| Heffter | Liu Liang | 0.624 | 0.585 | 0.612 |
| Heffter | $R_{0.25}$ | 0.542 | 0.769 | 0.833 |
| Heffter | $R_{0.50}$ | 0.578 | 0.530 | 0.550 |
| Liu Liang | $R_{0.25}$ | 0.701 | 0.186 | 0.187 |
| Liu Liang | $R_{0.50}$ | 0.721 | **-0.076** | **0.076** |
| $R_{0.25}$ | $R_{0.50}$ | **0.960** | -0.261 | 0.263 |

**Table 2.** Statistical measures between BL height estimates from radiosonde measurements and ERA5. r is the Pearson's correlation coefficient, FB is fractional bias and nRMSD is the Normalised Mean Square deviation. Values were calculated for different time of the day. Radiosonde data is taken as the observations in Equation 6. Correlation coefficients which are statistically non-significant at the 95% level are marked in italics. For each measure, the best value between the four different datasets is marked in bold.

| Time of day | Heffter vs ERA5 | | | Liu Liang vs ERA5 | | | $R_{0.25}$ vs ERA5 | | | $R_{0.50}$ vs ERA5 | | |
|---|---|---|---|---|---|---|---|---|---|---|---|---|
| | r | FB | nRMSD | r | FB | nRMSD | r | FB | nRMSD | r | FB | nRMSD |
| 00 UTC | 0.372 | 0.170 | 0.170 | 0.253 | -0.277 | 0.280 | 0.801 | -0.318 | 0.322 | **0.834** | **0.089** | **0.089** |
| 06 UTC | 0.193 | 0.581 | 0.607 | 0.417 | -0.365 | 0.371 | 0.679 | -0.303 | 0.307 | **0.711** | **-0.007** | **0.007** |
| 12 UTC | 0.539 | 0.499 | 0.515 | **0.794** | **0.071** | **0.071** | 0.645 | -0.107 | 0.107 | 0.696 | 0.083 | 0.083 |
| 18 UTC | *0.092* | 1.087 | 1.295 | *0.001* | 0.588 | 0.615 | **0.644** | **0.193** | **0.194** | 0.622 | 0.493 | 0.509 |
| All | 0.478 | 0.637 | 0.671 | 0.605 | **0.079** | **0.079** | 0.754 | -0.111 | 0.111 | **0.770** | 0.153 | 0.153 |

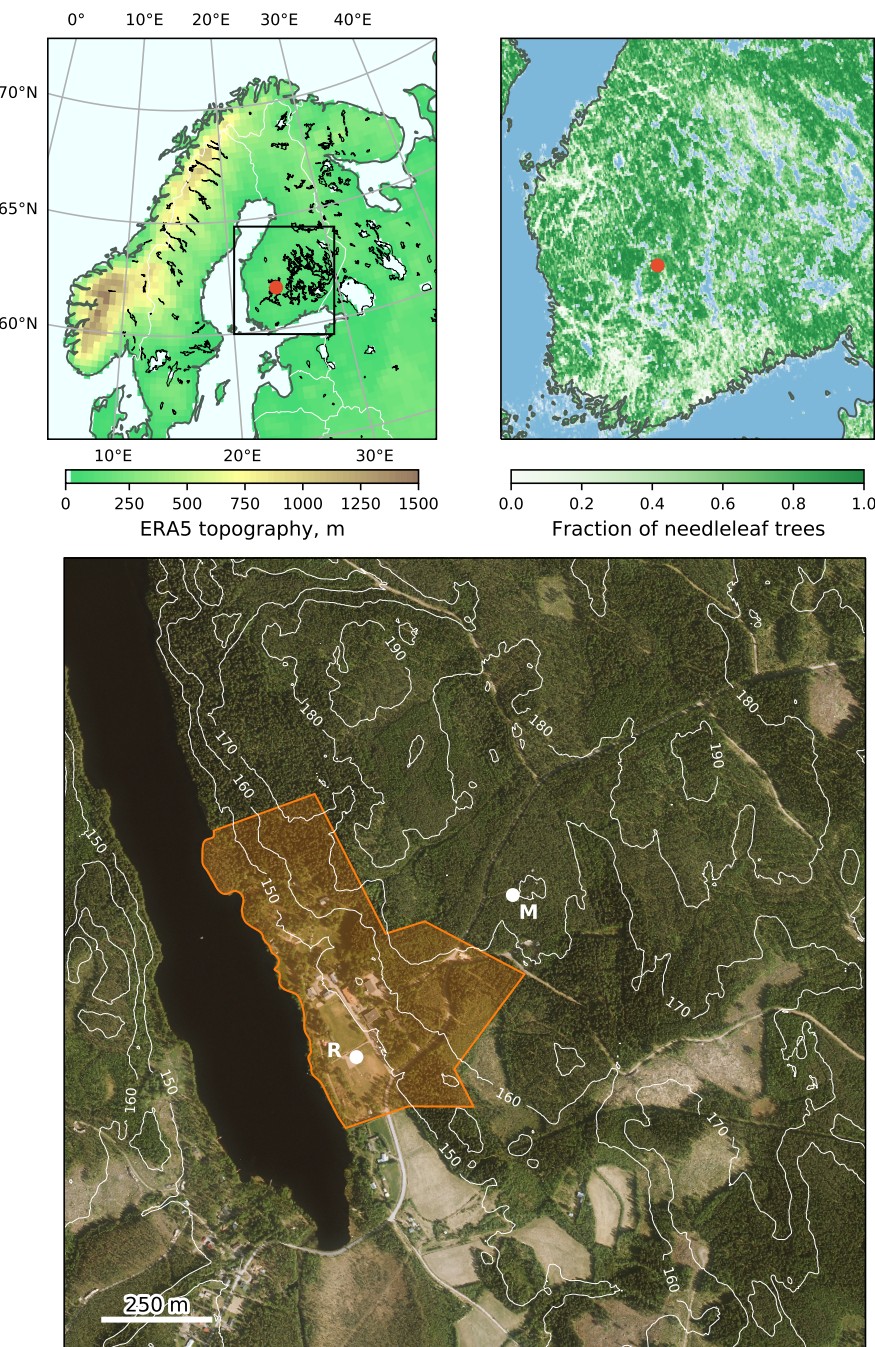

**Figure 1.** Location of Hyytiälä as shown by a red dot in the top two panels which show the topography (m) from ERA5 (left) and the fraction of needle leaf trees obtained from ECOCLIMAP-SG land cover map (right). The bottom panel shows the local area around Hyytiälä. The local topography (obtained from the National Land Survey of Finland) is shown in white contours (contour interval 10 m) and the area of the station is marked in orange. The location of the MWR is marked by an R and the mast where the eddy covariance measurements are made is marked by an M.

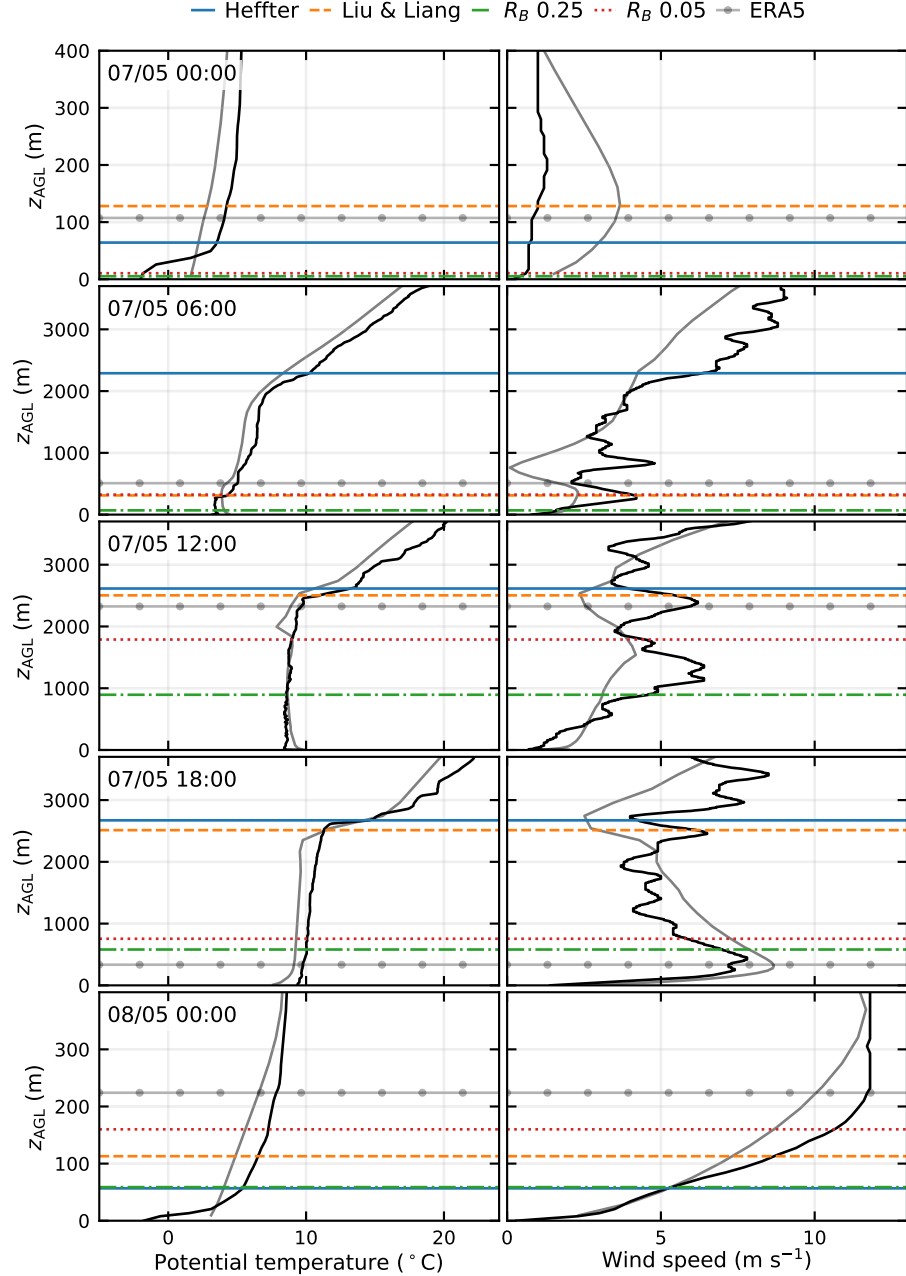

**Figure 2.** Vertical profiles of air temperature (left column) and horizontal wind speed (right column) from radiosonde soundings (black solid line - raw, high resolution data is plotted) and ERA5 (grey solid line) and the estimates of BL height using different methods at SMEAR III on 7-8 May 2014. The BL height is estimated using the methods by Heffter (1980) and Liu and Liang (2010) as well as using a threshold value for the bulk-Richardson number ($R_B$) of 0.05 and 0.25. $z_{AGL}$ stands for height above ground level. Times are given in UTC.

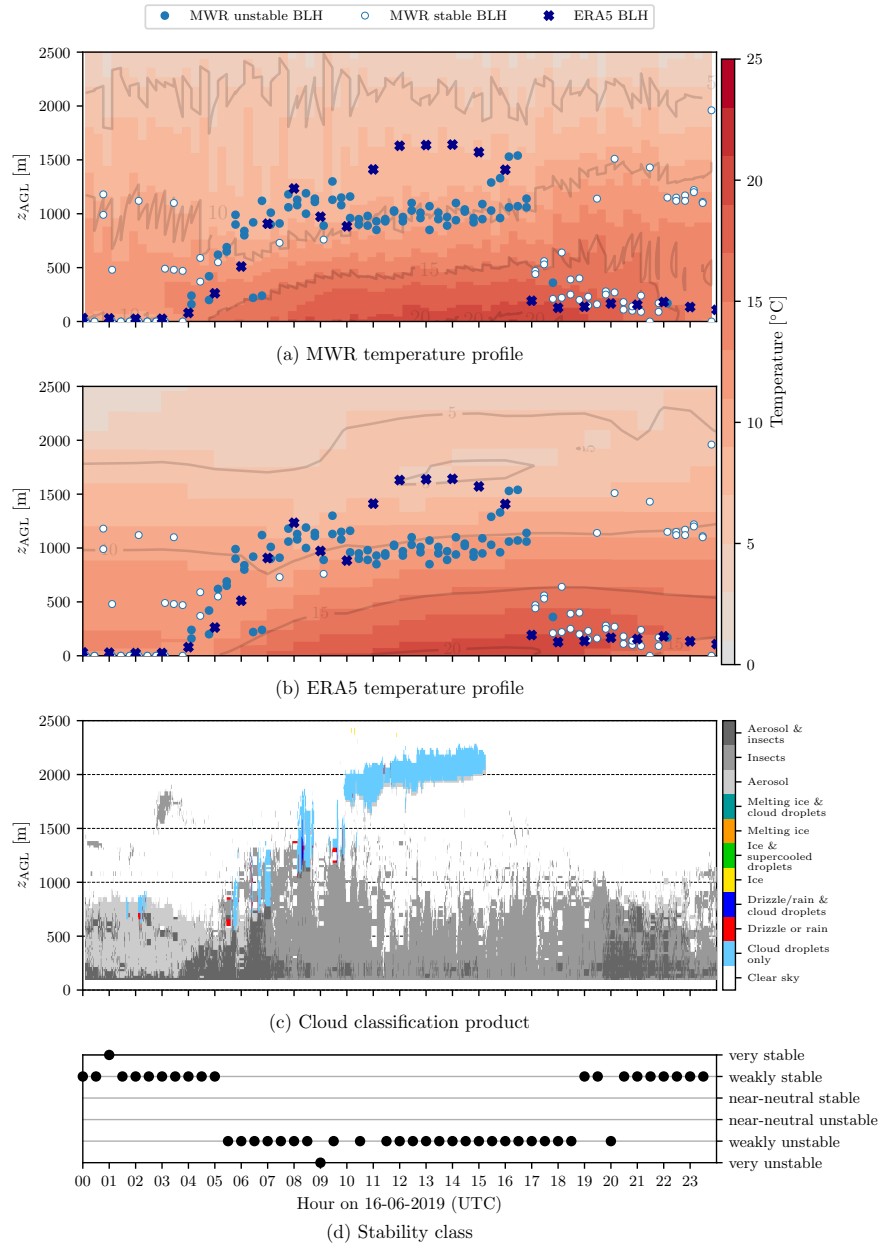

**Figure 3.** (a) Temperature profiles from MWR measurements and (b) ERA5, (c) a cloud profiling product (CLU, 2019) and (d) the stability regimes for 16 June 2019. Panels (a-b) have the MWR unstable BL height (filled circle), MWR stable BL height (empty circle), and ERA5 BL height (cross) plotted. The grey lines in panels (a-b) denote isotherms in 5 °C intervals. $z_{AGL}$ stands for height above ground level.

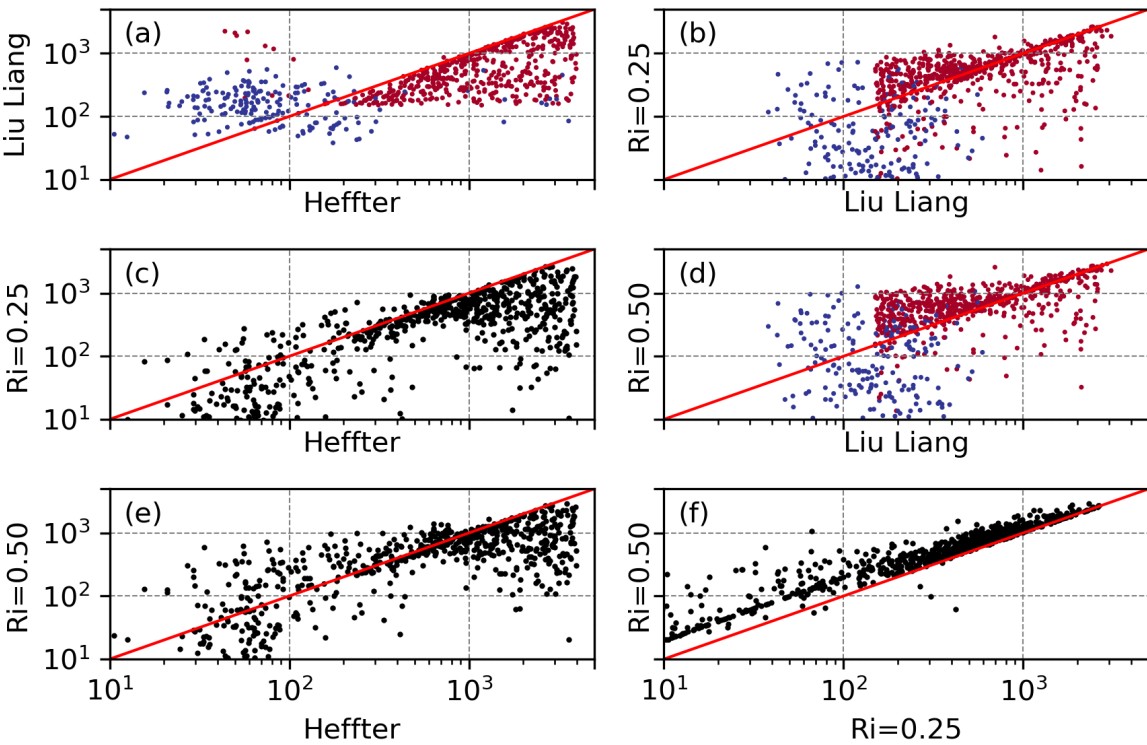

**Figure 4.** Scatter plots showing the relation between the BL height diagnosed from radiosondes taken during BAECC using 4 different methods. Data is from 1st February 2014 - 13th September 2014. In panels a, b and d, red points are when the LL10 scheme has diagnosed either a neutral or unstable BL and blue points are stable BLs. Red solid lines shows the 1-to-1 line. Note the logarithmic scale

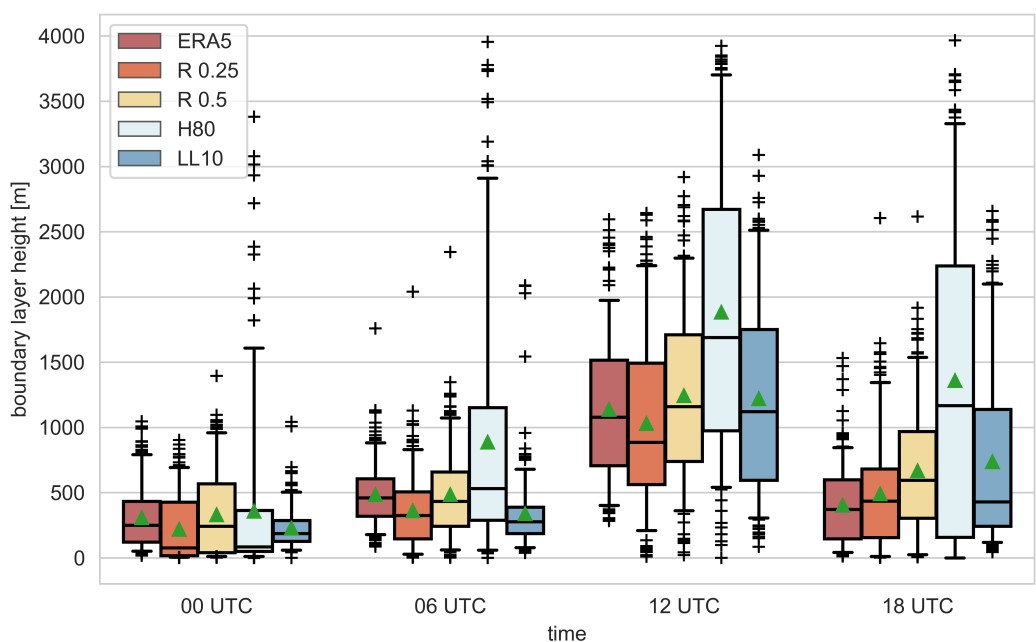

**Figure 5.** Boundary-layer height distributions from ERA5 and from the 4 different methods applied to the radiosondes divided by time of the day. The boxes in each time bin are colour coded according to the legend. The shaded boxes extends from the first quartile (Q1) to the third quartile (Q3) values of the data i.e the interquartile range (IQR). The black solid line shows the median and the green triangles the mean values. The whiskers extend to the 5th and 95th percentiles. Black crosses beyond the whiskers show the outliers.

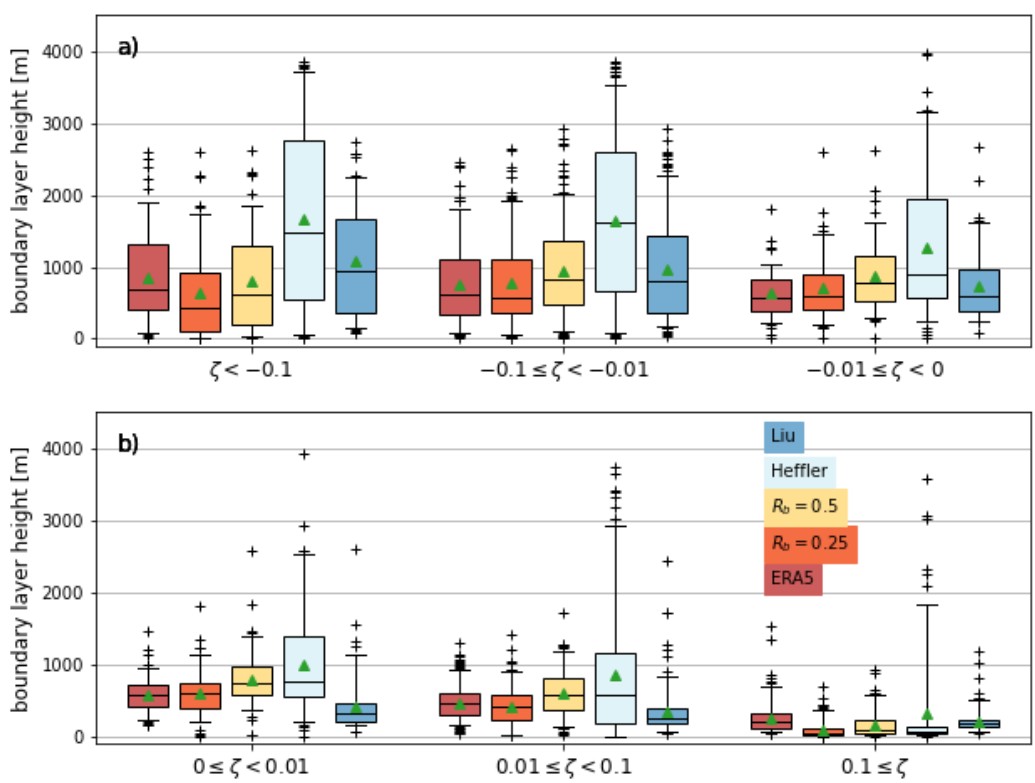

**Figure 6.** Boundary-layer height distributions from ERA5 (red) and from the 4 different methods applied to the radiosondes divided by stability class computed from the EC observations. Unstable classes are shown in (a) and stable classes in (b). The boxes in each stability bin are color coded according to the legend and the boxes and whiskers are defined as in Fig. 5

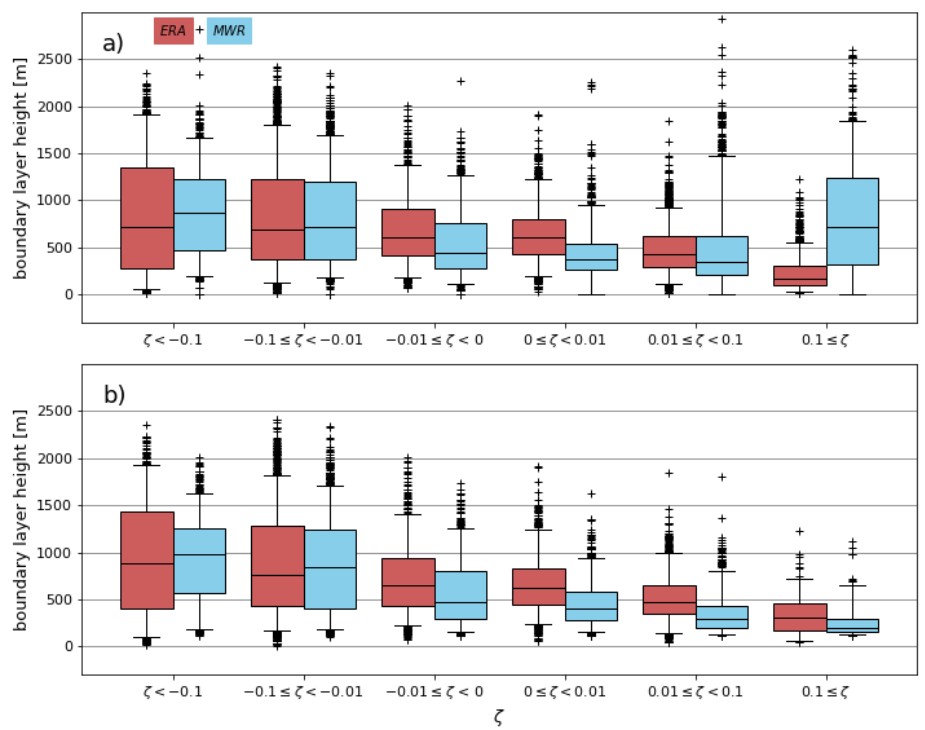

**Figure 7.** Boundary layer height distributions from ERA5 (red) and from the MWR (blue) divided by the stability class computed from the EC observations. The boxes and whiskers are defined as in Fig. 5. Panel (a) includes all MWR data. Panel (b) includes only times when the MWR has determined the BL to be unstable. The data covers the time period from 1st September 2018 – 31st August 2019.

**Table 3.** Statistical measures between BL height estimates from radiosonde measurements and ERA5. r is the Pearson's correlation coefficient and nRMSD is the Normalised Mean Square Deviation. Values were calculated for different stability classes. Radiosonde data is taken as the observations in Equation 6. Correlation coefficients which are statistically non-significant at the 95% level are marked in italics. For each measure, the best value between the four different datasets is marked in bold.

| stability | Heffter vs ERA5 | | | Liu Liang vs ERA5 | | | $R_{0.25}$ vs ERA5 | | | $R_{0.50}$ vs ERA5 | | |
| --- | --- | --- | --- | --- | --- | --- | --- | --- | --- | --- | --- | --- |
| | r | FB | nRMSD | r | FB | nRMSD | r | FB | nRMSD | r | FB | nRMSD |
| $\zeta < -0.1$ | 0.553 | 0.644 | 0.680 | 0.650 | 0.214 | 0.215 | 0.749 | -0.328 | 0.332 | **0.799** | **-0.0983** | **0.0985** |
| $-0.1 \leq \zeta < -0.01$ | 0.312 | 0.766 | 0.829 | 0.644 | 0.269 | 0.272 | 0.814 | **0.0415** | **0.0415** | **0.820** | 0.246 | 0.248 |
| $-0.01 \leq \zeta < 0$ | *0.0882* | 0.690 | 0.736 | 0.247 | 0.158 | 0.159 | 0.630 | **0.104** | **0.104** | **0.671** | 0.323 | 0.327 |
| $0 \leq \zeta < 0.01$ | 0.567 | 0.589 | 0.617 | *0.170* | -0.329 | 0.333 | **0.793** | **0.0306** | **0.0306** | 0.768 | 0.291 | 0.294 |
| $0.01 \leq \zeta < 0.1$ | 0.243 | 0.589 | 0.617 | *0.118* | -0.301 | 0.304 | 0.746 | **-0.103** | **0.104** | **0.754** | 0.257 | 0.259 |
| $0.1 \leq \zeta$ | 0.301 | 0.201 | 0.202 | *0.103* | **-0.190** | **0.191** | 0.593 | -0.971 | 1.11 | **0.604** | -0.457 | -0.469 |
| All | 0.502 | 0.636 | 0.671 | 0.646 | **0.0773** | **0.0773** | 0.789 | -0.112 | 0.113 | **0.803** | 0.151 | 0.151 |

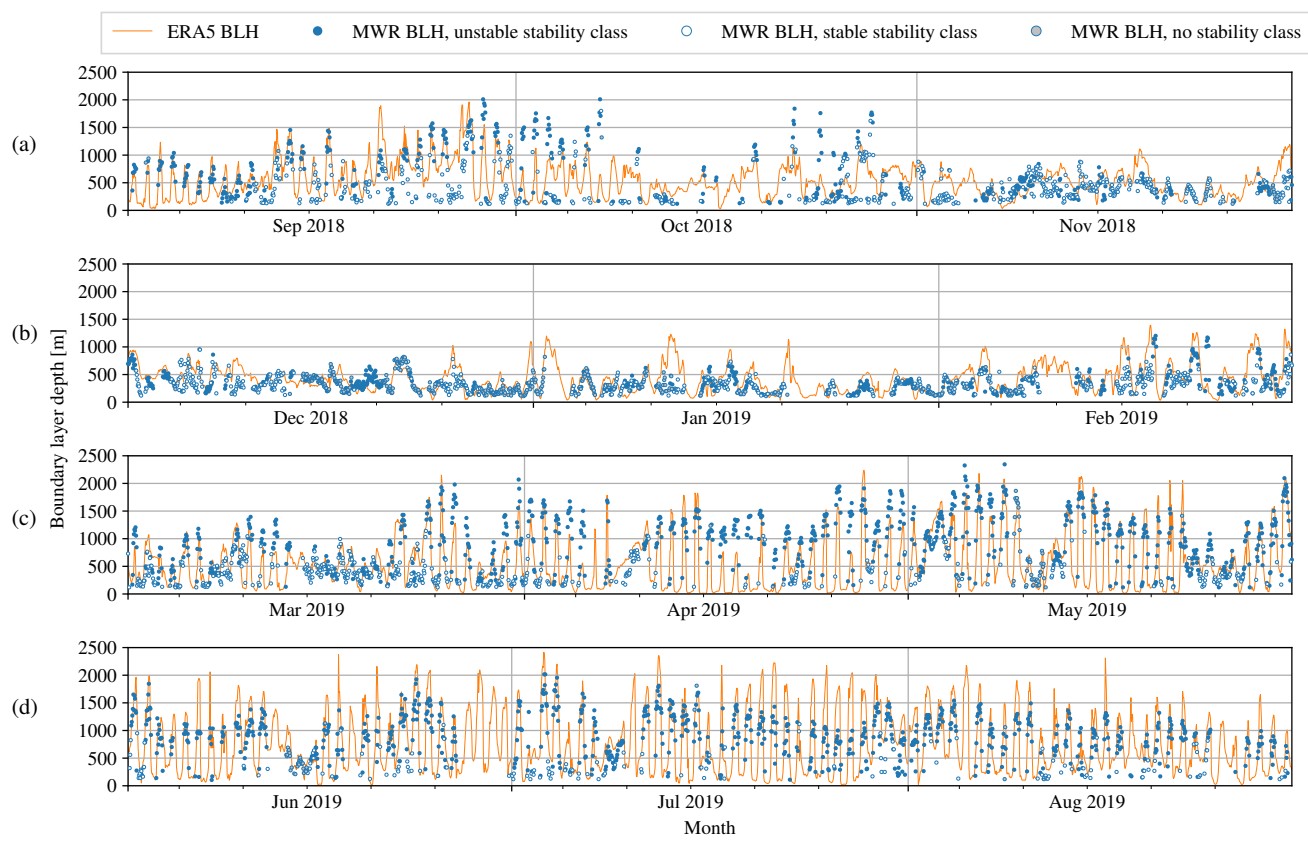

**Figure 8.** Timeseries of ERA5 and MWR boundary layer heights for (a) September 2018 – November 2018, (b) December 2018 – February 2019, (c) March – May 2019, and (d) June – August 2019. The orange line indicates the ERA5 BLH. The circles indicate the hourly median values of MWR unstable BLH; the times stability class indicates unstable BL (blue filled circles), the times the stability class indicated a stable BL (empty circles), and the times when stability class was missing (grey filled circles). Vertical lines denote the change of month, and ticks denote the 5th, 10th, 15th, 20th, and 25th days of the month.

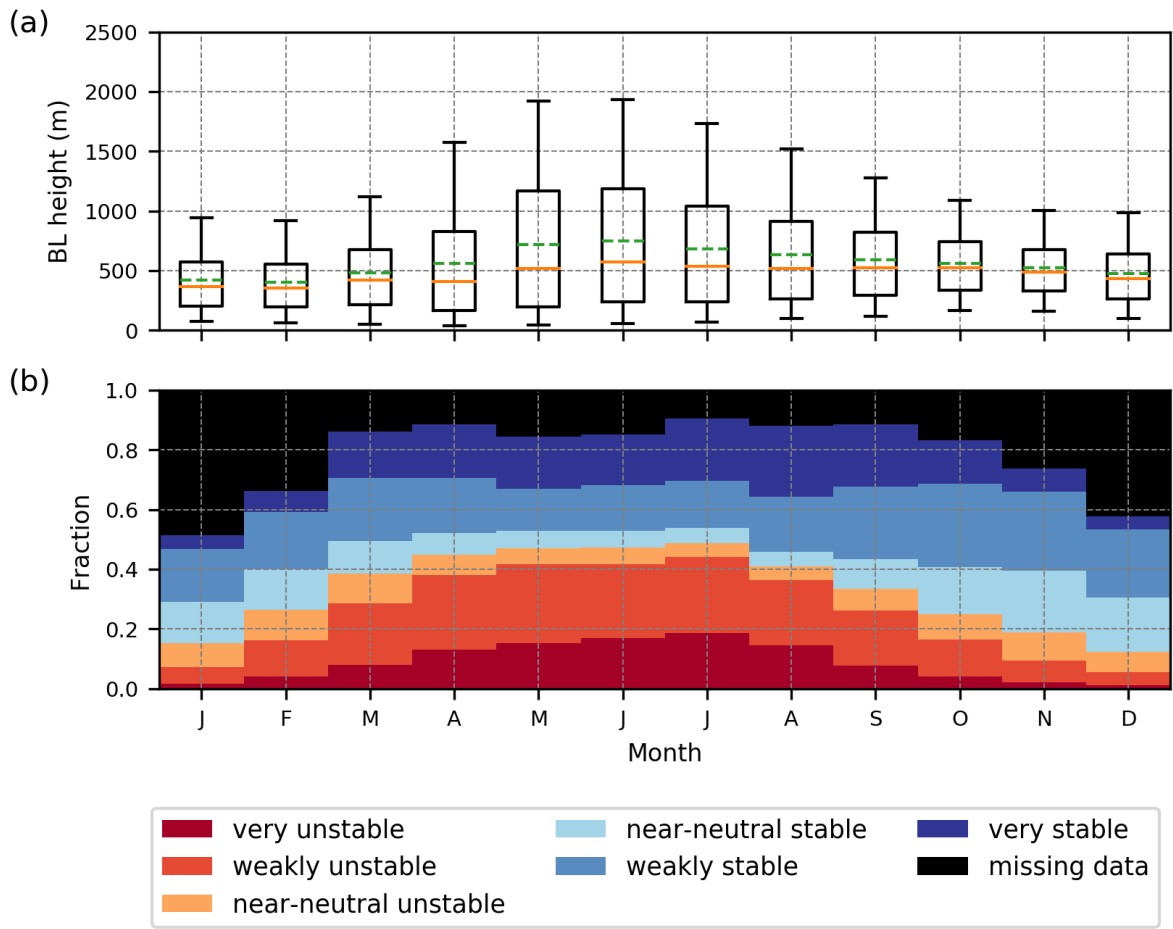

**Figure 9.** Annual mean cycles of (a) the ERA5 diagnosed BL height averaged over 1979 - 2019 and (b) the stability class derived from the EC data averaged over 1997 - 2019. In (a) ERA5 data at the grid point closest to Hyytiälä is plotted. Orange lines show the median values, green lines the mean value. Boxes represent the 25th to 75th percentile (interquartile range, IQR) and whiskers extend to the 5th and 95th percentiles. Outliers are not plotted. In (b) the fraction of each time that each stability class is observed is plotted for each month.

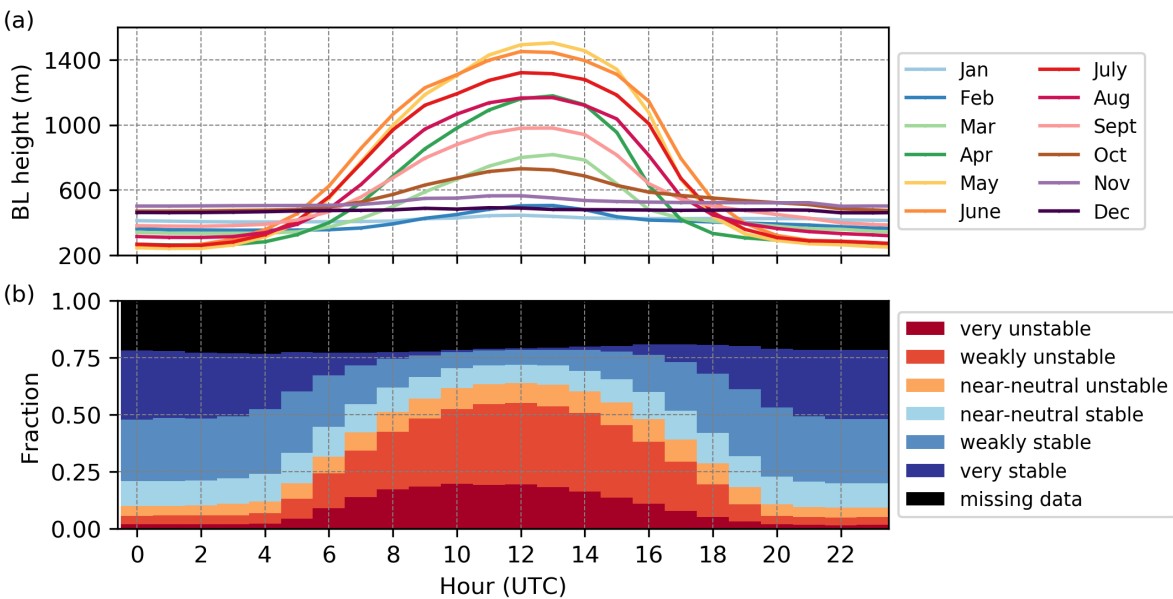

**Figure 10.** (a) Mean diurnal cycle of the BL height from ERA5 for each month averaged over 1979 - 2019. Data from every hour of the day is included. (b) Mean diurnal cycle of stability class for all months averaged over 1997 - 2019. The fraction of all possible observations times that each stability class is observed is shown.

**Table 4.** Statistical measures between BL height estimates from microwave radiometer and ERA5. r is the Pearson's correlation coefficient and nRMSD is the normalised root mean square deviation. Values were calculated for different stability classes using both all MWR BLH values, and only MWR BLH values determined with the algorithm for unstable BL. Correlation coefficients which are statistically non-significant at the 95% level are marked in italics.

| Stability | All MWR BLH values | | | Only unstable MWR BLH values | | |
|---|---|---|---|---|---|---|
| | $r$ | FB | nRMSD | $r$ | FB | nRMSD |
| $\zeta < -0.1$ | 0.376 | 0.049 | 0.049 | 0.540 | $-0.019$ | 0.019 |
| $-0.1 \leq \zeta < -0.01$ | 0.650 | $-0.016$ | 0.016 | 0.690 | $-0.016$ | 0.016 |
| $-0.01 \leq \zeta < 0$ | 0.624 | $-0.234$ | 0.236 | 0.745 | $-0.230$ | 0.231 |
| $0 \leq \zeta < 0.01$ | 0.182 | $-0.366$ | 0.373 | 0.562 | $-0.373$ | 0.379 |
| $0.01 \leq \zeta < 0.1$ | $-0.075$ | 0.051 | 0.051 | 0.391 | $-0.377$ | 0.384 |
| $0.1 \leq \zeta$ | $-0.286$ | 1.152 | 1.409 | *0.152* | $-0.219$ | 0.220 |
| All | 0.259 | 0.107 | 0.107 | 0.681 | $-0.141$ | 0.142 |

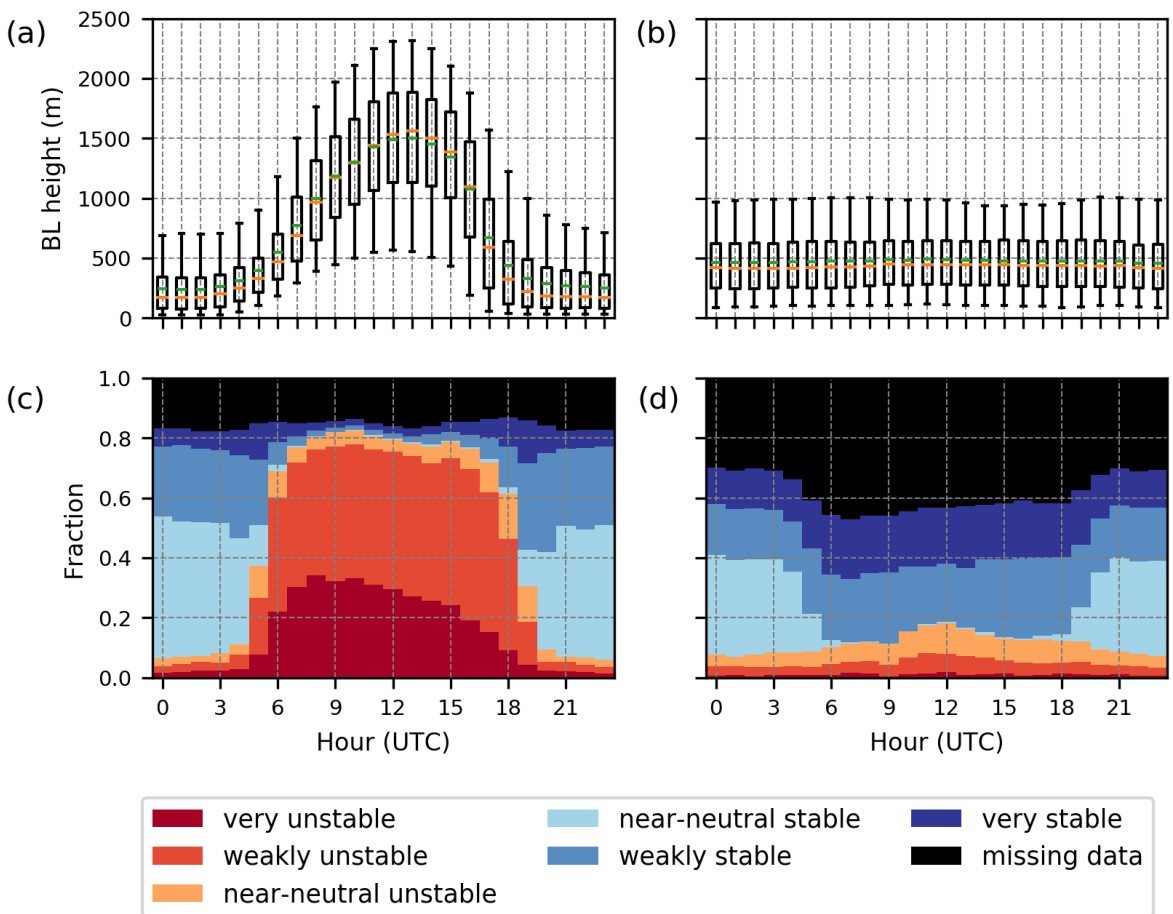

**Figure 11.** Variation in the (a,b) ERA5 BL height and occurrence of different stability classes dervied from the EC data (c,d) as a function of time of day in (a,c) May and (b,d) December. In (a) and (b) boxes represent the 25th to 75th percentile (interquartile range, IQR) and whiskers extend to the 5th and 95th percentiles. Outliers are not plotted.

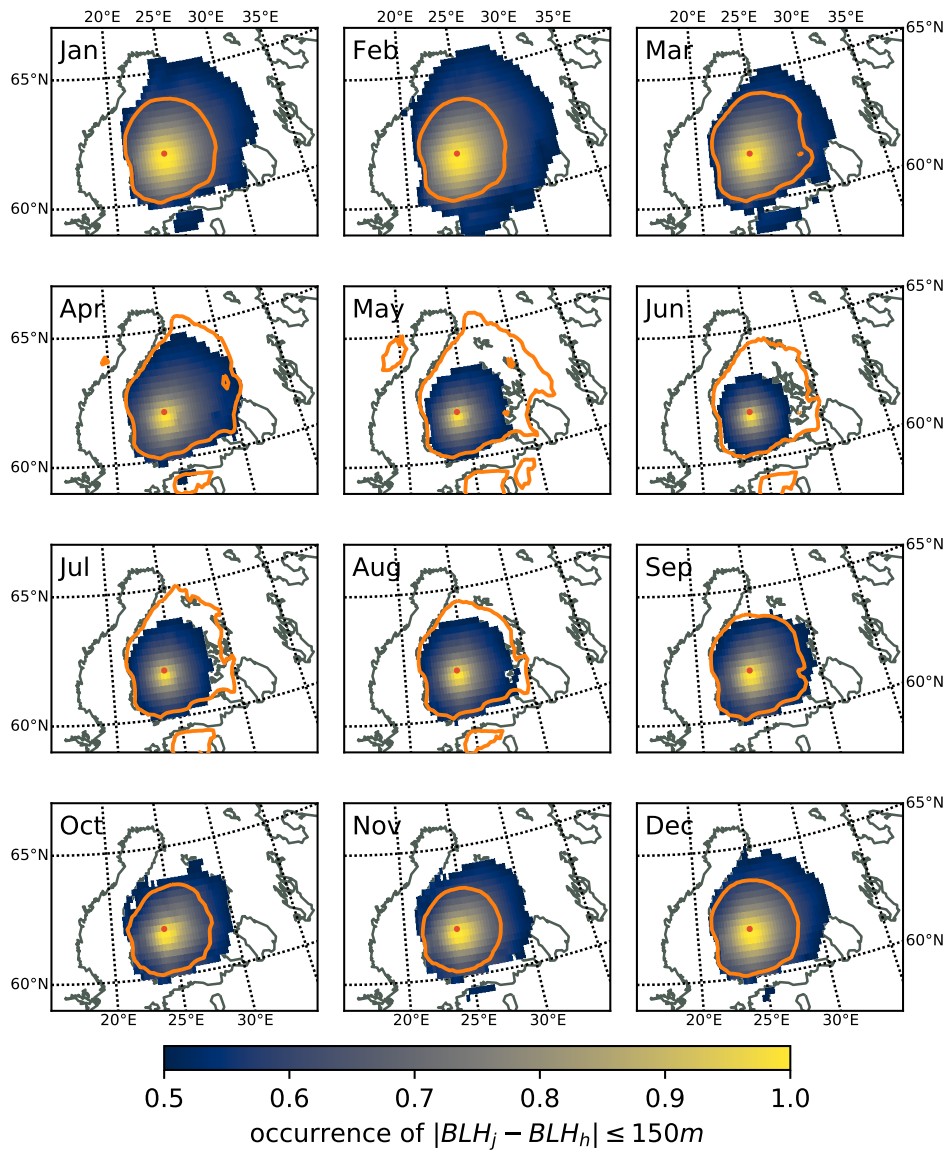

**Figure 12.** Spatial variability of the BL height in ERA5. Shading shows the fraction of all times (with 1 hour temporal resolution) between January 1979 and December 2019 that the BL height at each grid point is within 150 m of the BL height at the grid cell closest to Hyytiälä. Orange contours shows the 0.75 contour of the monthly correlations between ERA5 BL height series from the grid cell closest to Hyytiälä and the neighbouring ERA5 grid cells computed over the time period from January 1979 to December 2019. The position of the Hyytiälä station is marked with a red dot.