# Peer review of "Boundary-layer height and surface stability at Hyytiälä, Finland in ERA5 and observations"

_Atmospheric Measurement Techniques, 2021_

## Referee Comment (RC2)

Comments on the manuscript

**Boundary-Layer Height and Surface Stability at SMEAR-II, Hyytiälä, Finland in ERA-5 and Observations**

Submitted by

**VA Sinclair, J Ritvanen, G Urbancic, I Statnaia, Yu Batrak, D Moisseev, and M Kurppa**

for publication in **Atmos. Meas. Tech. (AMT)**

**General Assessment**

The paper deals with the estimation of the atmospheric boundary layer height (ABLH) and surface layer stability for the Station for Measuring Ecosystem-Atmosphere Relations (SMEAR) in Hyytiälä, Southern Finland. This station has been in operation for about 25 years with the aim to study land surface atmosphere interaction processes including trace gas and aerosol measurements. The station is part of the European Research Infrastructure Networks ICOS and ACTRIS, its comprehensive characterization with respect to atmospheric mixing potential is thus of high practical relevance. Before 2018, profile measurements of atmospheric variables have been performed at Hyytiälä during short-term field campaigns only using radiosondes, a microwave radiometer has been installed at the site in 2018. To create a long-term ABLH climatology for the site, the authors therefore use ERA-5 data which are validated against radiosonde and microwave radiometer (MWR) observations for two shorter time periods, unfortunately the reference radiosonde and MWR measurements do not cover the same time period.

The paper is well written and well structured. The approach chosen follows a sound methodology, and the results obtained are comprehensively presented in the manuscript. The paper is of limited originality with respect to the methods applied and the relevance for progressing measurement techniques. It relies on the application of methods adopted from literature, not all of them are considered as appropriate. Findings regarding the strengths and weaknesses of the different methods to derive the ABLH from radiosonde and MWR are in line with previous studies reported in the literature. Data Analysis of the ERA-5 data is, to my opinion, the stronger part of the paper. The observations part could be shortened, e.g., I am not sure whether the case studies section is really needed and whether all the experimental methods need to be discussed extensively considering their (known) limitations and methodical differences to the approach used in the ERA-5 analysis. In addition, I see a number of specific deficits and minor issues the authors should address, before the manuscript might be finally accepted. These are specified below.

**Specific comments**

1. A few more experimental and methodical details should be given, e.g.
   - Which type of radiosondes was used during the 2014 experiment? I am not aware of any radiosonde that measures dry and wet bulb temperature.
   - Was the 12 UTC radiosonde really released at 12 UTC? (in the operational praxis of Met Services the so-called 12 UTC sounding is often performed an hour earlier or so to ensure that a profile covering the whole troposphere is available at the reference time even in case of a necessary second ascent if the first one fails at low altitudes) – this would affect the discussion in line 625ff.
   - The authors mention two sonics at 23 m and at 46 m, but it remains unclear which of the sonics was used to derive stability?
   - Why did the authors use a temperature from within the canopy as a reference temperature?
   - The definition of the stability class ranges in lines 178-180 (and also on p. 10) differs from what is indicated in Figure 6, 7 and in Tables 3, 4.
   - The authors work with quite a number of different stability estimates (from the sonic, from the LL10 method, from the MWR output) – all these methods rely on different variables and represent different scales – have the results from these methods ever been compared and critically assessed?
   - For the temporal assignment it appears a bit irritating in several places that UTC is translated to local legal time instead of local solar time since ABL dynamics is related to the latter rather than to the former.

2. A critical discussion on the limitations of the measurements and methods is largely missing – just a few examples:
   - According to Figure 1, the sonic is positioned at the edge of an about 1:12 slope with a lake and some larger clearings just a few hundred meters to the west. Does this have any influence on the representativeness of the sonic data? How can it be justified that a local z/L value close to a forest canopy is appropriate to characterize atmospheric stability up to a height of several hundred meters. Did the authors perform any comparison of, e.g., the stability derived from the eddy covariance measurements compared to the one derived from the soundings in the frame of the LL10 method – see above?
   - Both the parcel and Ri number methods are quite sensitive to the surface temperature value used in the analysis, and it is always a matter of discussion which data to use here. A near-surface temperature measured independently in a shelter often does not fit to the radiosonde profile, on the other side the first temperature values from the radiosonde profile may suffer from the sonde being in the hands of the observer before being released and not experiencing the ventilation as during the free flight. In some studies, an excess temperature is introduced or a limit for a possibly superadiabatic near-surface lapse rate is set. These issues should be addressed in the methods description.
   - In lines 522-523, the authors bring in another ABLH estimate which is based on a completely different variable than all estimates considered in this paper. I suggest to omit this,

otherwise it would call for a critical discussion of ceilometer ABLH estimates which is a delicate task itself.

- Figure 9 indicates quite a substantial part of missing data, even in summer. For winter the authors argument with icing of the sensor, what are the possible reasons in summer? Are these missing data assumed to introduce a bias in the distribution of stability classes?

3. The authors in their analysis mix methods that are based on the consideration of different physical processes, they thus compare apples and pears naming all these derivatives ABLH without a critical discussion on that. This necessarily will result in some dis-agreement when comparing the results. It is obvious that Heffter's method is basically suited to obtain an estimate of the convective ABLH, and previous studies have already demonstrated the tendency of this method to overestimate ABLH. From the MWR, in stable conditons, they derive a kind of inversion or stable layer upper boundary, that is purely based on an analysis of the temperature profile which contains limited information on shear-induced turbulence, if at all. For the ERA-5 analysis, a Ri number method is considered as appropriate, it would thus be consistent to put emphasis on those experimental data which provide a comparative ABLH estimate.

4. The presentation of results in Sections 5 and 6 is widely descriptive, I would love to see some discussion of the findings already here, even if it is partly given in Section 8, e.g.,
- Why is ERA-5 in better agreement with Ri(0.5) from radiosoundings for the very unstable case, but with Ri(0.25) for the weakly stable case?
- Why are very shallow BLs less common than moderately shallow BLs in winter?
- Why is the ABL height variability so large in April and May?
- How to explain the annual cycle in the occurrence frequency of the weakly stable class (line 620ff)?

5. Deficits exist concerning the discussion of the authors work with respect to previous studies on, e.g., ABLH climatology at higher latitudes. How do the authors results behave when compared to the cited studies of Liu and Liang (2010) and Seidel et al. (2010, 2012)? I would also like to bring to the author's attention the work of Lotteraner and Piringer (Boundary-Layer Meteorol. 161 (2016), 265-287 – concerning the deficits of Heffter's method), as well as of Beyrich and Leps (Meteorol. Z. 21 (2012), 337-348 – concerning the analysis of radiosonde data: many of the authors results on methods comparison, annual variability and uncertainties have been reported there as well).

6. For the Figures, I noticed some issues concerning either the interpretation in the text or the methods chosen to display the data:
- Figure 2 (related to text line 295ff): It appears not really possible to reproduce the ABLH values given in the text from Figure 2. At 00 UTC, I would not see any of the values at 307 m – all the lines appear to be well below 1/4 of the first height axis tick label at 1000 m. The same is with the inversion at the 06 UTC plot which appears to be well below 500 m. And at

12 UTC, the lowest plotted estimate is definitely below 1000 m whereas the text gives the minimum value with 1073 m.

- Figures 5-7, 9, 11: I am not sure whether the choice to plot the whiskers at Q1 – 1.5 IQR and Q2 + 1,5 IQR is a clever one given the skewed distributions of the variables, because this does not consider the real distribution of data points in the lowermost and uppermost quartiles. It may thus happen, that – especially the lower – whiskers cover a range where no data points occur. This becomes obvious by the many whiskers for ABLH ending at z = 0 m. I probably would have chosen to represent the 10% / 90% percentiles by the whiskers.
- Figure 10: It does probably make not much sense to average the mean diurnal cycle of the stability classes occurrence frequencies over the whole year as the seasons behave very different.
- Figure 12: I am not sure whether the correlation coefficient is a suitable measure for this type of analysis, at least if the authors chose this as the only parameter considered. From the seasonal distributions we have learned that derived ABLH values vary over a wider range in summer than in winter, this automatically gives higher correlation coefficients if at least the general tendencies are the same in the two data sets considered. Wouldn't it be an idea to consider absolute or relative differences here as well, the more since for practical applications it might be of higher relevance to know the area over which ABLH does not differ by more than a certain absolute (say 100 m or 200 m) or relative value (say 20 %), rather than to know correlation coefficients. Similarly, I doubt whether just from the value of the correlation coefficient one may judge about the spatial variability (lines 680f).

**Minor Issues**

- The abbreviation SMEAR is introduced in the title already and it occurs in the abstract again before it is finally explained in line 73.
- The abstract could be shortened a bit. E.g. the first two sentences might be deleted, they would well fit in an introduction section, but are unnecessary in an abstract. I also consider it for unusual in an abstract to write "for example …" – either the result is so relevant that it should be stated or it might be omitted.
- Line 36-37: Here the authors state that vertical profiles of temperature and wind speed are required to determine the ABLH, then they write that "these profiles can be obtained from … ceilometers." I do not know any ceilometer that would provide wind and temperature profiles.
- Line 45-46: Another limitation of remote sensing instruments has to be seen in the fact that often more than one instrument is needed to measure the relevant variables, while tower measurements or sondes may provide the full set of essential thermodynamic variables.
- The authors do not seem to be friends of using commas to structure a sentence, this makes it a bit difficult to read in some places (e.g., line 74, 632, 709, 737, and others)
- Line 119 / line 129: If M is the "main SMEAR station location" (line 110) in Figure 1, I would see the location R to the Southwest of M, not to the Southeast.

- Line 207: "Stable boundary layer are not characterized by inversions." – This is a statement I simply do not understand.
- Line 246: Isn't it a bit dangerous to consider just the immediate neighboring levels below AND above a given height to diagnose a low-level-jet. Often the jet nose may be not that sharp that there is a significant change of wind speed both below AND above the height of the maximum?
- Line 269: The use of $z_0$ is a bit misleading here, $z_0$ is normally taken as the aerodynamic roughness length.
- Line 278-280: The Vogelezang and Holtslag method additionally considers a correction term to account for surface friction / shear.
- Line 312: It is not clear to me why – "due to the inclusion of the wind profile ... the Ri number methods diagnose shallower SBLs". Additional consideration of the wind profile would add shear-induced turbulence. Parcel methods will give ABLH where $\theta_h > \theta_0$ while Ri methods will lift this up before Ri exceeds a value of Ri = 0.25 or Ri = 0.5. In fact, parcel methods are equivalent to Ri methods with Ricrit = 0, and with critical Ri values above zero they should give higher ABLH values.
- Line 327-328: I cannot follow this argumentation. The wind speed differences with respect to the near surface values are larger in the measurements when compared to the ERA-5 profiles up to a height of about 2.5 km, consequently the denominator in the Ri definition gets larger, and hence Ri gets smaller. It thus appears not logical that the threshold of Ri = 0.25 should be exceeded at lower heights.
- Line 370: I would prefer to call this RMSD instead of RMSE. To use the word "error" would imply that the truth is known based on a well-calibrated method.
- L406-L408: How to explain the outliers in Figure 4 where Ri(0.5) < Ri(0.25)?
- L458: I would not a-priori expect that the Ri(0.25) methods radiosonde data must be in better agreement than the Ri(0.5) method with the ERA-5 ABLH estimate, even if the latter uses the same threshold: Note that the profiles have different vertical resolution and the reanalysis profiles are usually much smoother than any measured profile which will strongly influence the level of exceedance of fixed threshold values.
- Line 466-467: 23 % is a quite accurate number, I would not call that "approximately".
- Line 609-610: This seems to be a bit too general, the very stable cases do occur more frequently in summer.
- L643: "In winter cold conditions are required for shallow boundary layers to develop" – This statement calls for some further explanation.
- L669: It appears to be a subjective view whether Figure 12 really represents a "considerable part of Northern Europe".
- L724f: This conclusion does a bit contradict to the discussion 457-463 ("high vertical resolution ... sufficient" vs. "limited vertical resolution" explaining disagreement)
- L735: This is not a suggestion but a rather logical consequence of the stability classification with the LL10 method as described in Section 3.1.
- L742-743: Again, a minor inconsistency: Here the authors, to my opinion, correctly point at the difficulties to determine the ABLH during the evening transition both from the

measurements and in ERA-5, in Section 5.2 (line 496) they solely "blame" ERA-5 for the deficits.

- L767f: I also see the widely missing diurnal cycle (no pronounced daytime heating) as an additional reason for the reduced occurrence of very stable situations in winter, except of the more frequent occurrence of clouds.
- Figure 2: Relative humidity is not a good choice to plot vertical humidity profiles for illustrating the ABL structure, absolute or specific humidity would be better here. In essence, the humidity profiles are not even needed to be displayed here, at least they are not discussed in the text.
- Figure 4: The diagrams give the impression that some lower boundary values exist for the LL10 method (no stable values below 40 m, no unstable values below ca. 150 m) – nothing has been mentioned about that when describing the method.
- Figure 5: The black points are black crosses.
- Figure 8: It appears unnecessary to explain the legend again in the text of the Figure caption.
- Figure 9, caption: Just to make sure: panel (b) is stability from the sonic or from ERA-5? The data period represented by the graphs should probably be mentioned in the Figure caption, the more if it is not the same (ERA-5 40 years vs. sonic 19 years, as in Figure 10?)
- Table 3, caption: NMSE must be nRMSE, in the Table NRMSE should be nRMSE according to the definition given in the text (but see my remark on RMSE vs. RMSD)

And a few misprints:
- Line 34: assumption
- Line 104: years
- Line 185: delete "can"
- Line 294: suggests
- Line 326: value
- Line 437: in → is
- Line 451: methods
- Line 459: a systematic bias
- Line 582: delete "the" at the beginning, and delete "monthly" after "month"
- Line 659: BLs instead of BLS
- Line 735: This suggests …
- Line 767: clouds
- Caption Table 1, 2nd line: coefficients
- Figure 6, legend: Heffler must be Heffter

**Recommendation: Major Revision**

---

## Author Response (AR1)

**Response to Reviewers – "Boundary-layer height and surface stability at SMEAR II, Hyytiälä, Finland in ERA5 and observations"**

Victoria Sinclair, Jenna Ritvanen, Gabin Urbancic, Irina Statnaia, Yurii Batrak, Dmitri Moisseev and Mona Kurppa

February 2022

We thank the reviewer for their constructive comments on our submitted manuscript. We have copied the comments of the reviewers in black here and include our response to each individual comment in blue.

**Reviewer 1**

**Main comments:**

1. The parcel method is a commonly and widely used method in the CBL determination. It is applied by the automatic BL detection from the MWR but not on RS. A comparison of this method with the already applied on RS would have been worth. Similarly, the Heffler and part of LL10 methods applied on RS could also be used for MWR.
   First, we would like to note that the manuscript does not aim to compare BL heights obtained from the MWR to those obtained from radiosondes. Second, we would like to stress that our study does not aim to develop new methods to identify the BL height nor improve pre-existing methods. Third, we want to point out that our choice of methods to identify the BL height from the radiosondes was based solely on the options provided within the ARM Value-Added-Product (VAP), "Planetary Boundary Layer Height": https://www.arm.gov/capabilities/vaps/pblht. As we state in section 2.2, the radiosondes were made during an intense campaign when one of the ARM mobile facilities was in Hyytiälä. ARM routinely computes and makes available this VAP for all permanent sites and all campaigns. For example, this VAP is available from ARM's permanent sites since 2001 (Oklahoma), 2002 (Barrow), 2013 (Grasiosa) and from 2001 - 2014 (Tropical west Pacific). Therefore, to remain consistent with these other stations, and other ARM campaigns, we only used the methods already in this VAP. This is the main reason we do not apply the parcel method to the radiosonde data. We have added text to the beginning of section 3 to stress this more clearly.

   Similarly, the Heffler and part of LL10 methods applied on RS could also be used for MWR.
   The aim of this manuscript was to take pre-existing methods, that are used by others already, to determine the BL height. This is why we only use the manufacturer provided algorithm to

diagnose the BL height from the MWR. In theory, we could apply the Heffter method to the MWR data but, in our opinion, it does not make sense to apply a modified LL10 method to the MWR. This is because we would be introducing yet another method to identify the BL height which is not the aim of this paper. Furthermore, given that the radiosonde observations and the microwave radiometer observations do not overlap in time and that we cannot compare them, there is limited scientific value of attempting to apply the same methods to the MWR as are provided in the ARM VAP.

2. The MWR has data since June 2018 leading nowadays to a 3 years time series. A one-year time series is presented in Fig. 8. A climatology with these measurements (at least the year presented) would also be very valuable and allow a comparison with the ERA5 long-term time series. We have created figures of the monthly and diurnal climatology of the BL height diagnosed from the MWR based on the data period considered in the manuscript - September 2018 to August 2019 (Figure 1 in this response). This figure includes all of the MWR data. When the mean diurnal cycle is considered in the warm season, two peaks in BL height are evident; one around 11 UTC and a second one in the middle of the night - around 00 UTC. This strongly suggested to us that the MWR does not diagnose the BL correctly at night in the warm season. Based on the climatology of stability class from the eddy covariance measurements, we know that this is when very stable or weakly stable conditions are common. Hence we concluded that in some cases when the BL is stable the MWR does not identify a physically meaningful BL height - it over-estimates the BL height considerably potentially identifying a residual layer. This is confirmed by Figure 2 in this response which only includes data when the MWR has determined the BL to be convective. When the stable points are removed, the diurnal evolution of the BL height from the MWR is much more reasonable. Given this issue with the stable BLs, a fair comparison with the ERA5 climatology figures is not straightforward and therefore, we do not add these new figures to the manuscript.

In addition, we also investigated if our results would have changed if we included additional data by also creating the climatology for June 2018 - Dec 2021 (Figure 3). By comparing this figure to Figure 1, we see that there are only small differences. Therefore, we do not include any additional MWR data in the revised manuscript as this would require considerable work and would not change the main conclusions.

3. The radiosondes data are only available from February to mid-September. What would be the impact of the October-January (i.e. most of the fall and winter periods) on the comparison between the methods? On the validation of ERA5? If ERA5 cannot be considered as validated for Fall and Winter, should the comparison between ERA5 and MWR and the climatology be discussed differently ?

Firstly, we do compare ERA5 to the radiosondes BL height estimates for February which is the coldest month of the year in southern Finland and certainly is a winter month. Thus, we are of the opinion that it is somewhat unjustified to say that ERA5 is not validated for winter. Secondly, our results show that we have validated ERA5 BL heights for all classes of surface stability - it is not the case that we are completely missing certain types of the BLs in our 8 month validation period. This is shown in Figure 4 in this response which, for the 23 years of surface stability data from the eddy covariance measurements, shows that, climatologically,

[Figure]

Figure 1: The (a) annual and (b) diurnal cycle of the boundary layer height from microwave radiometer measurements for the Sep 2018 - Aug 2019 period included in the manuscript. The figures contain both the convective and stable BLH MWR measurements.

it is only the weakly stable and near-neutral stable BLs that are more common in October to January (the time period we did not perform a comparison for) than in February to September. Furthermore, Figure 4 shows that even in the time period we do consider, these types of BLs are sampled enough i.e. the orange bars are not tiny. However, we have added some text to section 5.2 and to the discussion to reminder a reader that the comparison with ERA5 does not cover a full year.

4. Discussion of Fig. 5, 6 and 7:(§ 5.1, 5.2 and 5.3): the comparison between the methods as a function of time and of stability is very interesting. The description is fine and could be published as it. It could however be largely improved if the structure of these § would rely on the differences between methods. E.g. (1) The largest differences are found between Heffler and the other methods in very unstable cases due to the required temperature difference of 2K. Is it possible to improve this Heffler detection by lowering the required T difference in case of unstable situation? (2) ERA5 has the large discrepancy in case of very stable situations - is it due to the lower vertical resolution, due bias in T profile or in wind profiles? (3) ERA5 agree better with $Ri_{0.5}$ for very unstable cases - due to the lower vertical resolution (L457-463). (4) at 00 UTC or in case of stability, LL10 leads to higher MLH - inherent to the used method since LL10 estimate  the top of the stable layer or the LLJ and the Ri a very shallow height that does not really correspond to a physical layer. (5) ERA5 << MWR in case of stability - MWR measures

[Figure]

Figure 2: The (a) annual and (b) diurnal cycle of the convective boundary layer height from microwave radiometer measurements for the Sep 2018 - Aug 2019 period included in the manuscript.

the top of the stable layer and Ri another usually shallower layer. (6) ERA5 << MWR in case of stability - MWR measures the top of the stable layer and Ri another usually shallower layer. I am aware that a reorganization of the section represents a lot of work but I think that it would improve the manuscript. An alternative would be to enhance these points in the discussion.

Overall, these comments have been very helpful. Although we have not hugely re-structured section 5 we have made some major changes to parts of the manuscript. Firstly, we have added details to section 3 to discuss and highlight the differences between the methods we use to diagnose the BL height. Here we now also stress that the different methods do not all quantify exactly the same physical thing and thus even a priori we should not expect the different methods to agree. We also agree that it is important to be aware of the sensitivity of the diagnosed BL height to the various thresholds applied in the different methods. As such we now include more critical analysis of this in sections 3, 5 and 8 and more references to previous studies which have considered these sensitivities. Based on the specific comments here, we have also included more more physical interpretation to the detected differences in section 5.

Some specific comments relating to the specific points raised: (1) As we want to evaluate pre-existing methods included within the ARM VAP, it is not in the scope of this current study to further develop / alter the Heffter method. (2) To determine the cause of the ERA5 discrepancies in stable conditions would require considerable additional analysis which is not in the scope of the current study. It would also very challenging to separate the difference causes of error as

[Figure]

Figure 3: The (a) annual and (b) diurnal cycle of the boundary layer height from microwave radiometer measurements for June 2018 - Dec 2021 period. The figures contain both the convective and stable BLH MWR measurements.

the limited resolution in ERA5 may limit the accuracy of the temperature and wind but also any inadequacies in the boundary-layer parameterization scheme could also cause errors. (3) ERA5 may agree better with $Ri_{0.5}$ because of the resolution but also a warm surface temperature bias in ERA5 could cause this (We now discuss the potential impact of the difference in resolution between the radiosondes and ERA5 in more depth in the revised manuscript). (4, 5) We now attempt to highlight the differences between the methods, and the likely impacts of these differences on the BL height, more in section 3 and section 5. (6) For the unfiltered data, yes, ERA5 has much shallower BL heights than the MWR for very stable conditions. As stated in our response to major point 2 above, this discrepancy may be due to the MWR incorrectly identifying the height of some very stable BLs.

**Minor comments**

1. L 17: please mention that the climatology rely only on ERA5 time series. Revised to now state that the climatology is based on ERA5.

2. L 220: what is the initial vertical grid of the radio-sounding? What are the reasons for sub-sampling RS vertical profiles and the potential consequences on each PBLH detection method? The radiosondes measure data every two seconds. Therefore the vertical grid in terms of pressure is not uniform. On average, below 700 hPa, there are measurements every 1.2 hPa but this can

[Figure]

Figure 4: The fraction of times that different stability classes are detected for the months with no radiosondes (October to January) and for the months for which ERA5 was validated (February to September).

vary (standard deviation of 0.12 hPa) due to different ascent rates. This information has been added to section 2.2 where we previously stated the resolution as 10 - 15 m. Therefore, one reason for the sub-sampling is to put all of the radiosonde data on a vertical grid with the same resolution. A second reason is to smooth the profiles which makes computing gradients less noisy.

3. L 222-229 H80: Is the PBLH very sensitive to the chosen potential temperature difference of 2K and to the 15 hPa vertical resolution? In general increasing the resolution means the vertical potential temperature gradients are larger and thus the BL height can be diagnosed lower. Also if the 2 K threshold was decreased, lower BL heights would be diagnosed. However, as stated above, it is out of the scope of this study to revise the pre-existing methods that are used to diagnose BL height but it is important to be aware of the limitations, and the sensitivities, of all the methods used to identify the BL height. Therefore, we have attempted to add a more in-depth critical discussion of this in section 3 and in the discussion section. We have added additional reference to previous studies that have considered these sensitivities.

4. L 230-240: For convective BL, LL10 has a large similarity with the parcel method. The differences between both methods are 1) the parcel method take the potential temperature at ground (2 m) and LL10 at 150 m, 2) the parcel method use an instability threshold of 0 (and 0,5 for LL10) and 3) LL10 takes the first level with the potential temperature gradient $> 4.0$ K km$^{-1}$ higher than the level with the potential temperature difference with the one at 150 m $< 0.5$. Differences 1) and 3) lead to lower and higher PBLH than the parcel method, respectively. Are the PBLH sensitive to the various thresholds? What is the difference between LL10 and the parcel method in case of unstable situations? This is important since the MWR use the parcel method. We do not estimate the BL height from the parcel method and thus we cannot quantitatively answer how it differs from the LL10 method. The BL heights are likely sensitive to the various thresholds and we now include more discussion about this in the revised manuscript and discuss previous studies which have analysed these sensitivities.

5. L249: -9999 corresponds to a missing value but it's real value is not important and differs as a function of the programming language. Yes, this is true. We have deleted this sentence as how a missing value is specified is not necessary information.

6. Figure 2: The RS profile corresponds to the 5 hPa or 15hPa vertical resolution? a plot of the bRi number could also help to understand the difference between the threshold and explain the failure of this method at 12 UTC. A smaller vertical extend of the y scale for the 00 UTC case could help to see the differences between the methods.The radiosonde data is the raw, high resolution data measured by the radiosonde every 2 seconds / approximately every 1.2 hPa. We have added this information to the caption. In addition we have also changed the scale on the y-axis for the top and bottom rows.

7. L 306: it would help to know what are the potential reasons leading to underestimation by the Ri methods. It has also to be mentioned that ERA5 (also a Richardson number-based method) is ok contrarily to Ri applied to RS. This case study has proved difficult to fully understand as we took the ARM provided values and did not compute them ourselves. We believe that the underestimation by the Ri methods maybe related to the choice of surface values in this case or is affected by the vertical potential temperature profile which has some weakly stable layers present (see Figure 5 in this response). We have added text to the manuscript about this. We have also added more information to section 3 regarding the comparison of ERA5 (Richardson number method with $Ri_c = 0.25$) BL heights to those from the radiosondes using the "same" method. This is not an exact, fair comparison as the vertical resolution and surface values differ. Therefore, it cannot be expected a priori that these two methods should agree perfectly.

8. L324-325: the subject of "is within the range. . . " is not clearly defined. This has now been revised to read "the BL height from ERA5 is within the range of values estimated from the radiosondes".

9. L348-349 and 353-355: the fact that the nocturnal (or stable) BL are different between MWR and ERA5 rely mostly on the applied method. ERA5 uses Ri that gives a MLH almost at ground in case of atmospheric stability; the layer given by Ri during the night is however physically/thermodynamically not well defined. MWR search for the vanishing potential temperature gradient corresponding to the top of the stable layer. Both methods cannot be directly compared

[Figure]

Figure 5: Vertical profiles of potential temperature (blue) and virtual potential temperature (orange) on the 5-hPa vertical grid at 12 UTC on 7th May 2014. Horizontal lines should the BL height diagnosed by the four different methods.

in such a case. There are two separate issues here. Firstly, as the reviewer correctly states, the method applied by the MWR to identify the BL height is very different and thus not directly comparable to the method that ERA5 uses. We have revised section 3 and the discussion to highlight the differences between the methods more and to more clearly state that we would not expect the methods to agree exactly. The second issue is similar to that discussed in our response to major point 2 above and highlighted by figures 1b and 3b. The MWR algorithm does appear to identify physically incorrect BL heights at night (when the BL is stable).

10. L377- 378: this is not obvious. The same data are used but the comparison is between BLH. This refers to the correlation coefficients between the different methods applied to the radiosonde data. Although the methods are all different and thus not expected to agree perfectly, we expected the correlations to be at least positive. However, we agree that it is not a given that the correlations would be statistically significant. We have therefore revised this sentence.

11. L391-393: grammatical problems. We have revised this sentence.

12. L396-399: Per definition, the parcel method always detect a lower BLH than the Ri. The cases LL10>Ri and LL10<Ri should then be discussed as a function of the differences (see comment L230-240) between LL10, the parcel method and the Ri one. We do not use the parcel method to identify the BL height from the radiosondes and therefore to discuss this would be speculative. However, as said above, we now include more critical discussion of the differences between the methods.

13. L 420-421: "The LL10 scheme and especially the H80 method produce the deepest BLs when the surface layer is very unstable." It seems then that in case of large unstability, the LL10 "over-shooting parcel" is responsible of the higher BLH than the Ri method. Is my assumption right? This is certainly one possibility. However, the LL10 scheme also searches for the level where the temperature exceeds the temperature at 150 m ($T_{150m}$) plus 0.5K. Therefore if $T_{150m}$ + 0.5 is greater than the near surface temperature used by the Richardson number method, this would also result in deeper BLs in the LL10 method.

14. L427-428 + Fig 6:" When weakly stable and near-neutral stable conditions are considered, the LL10 scheme has the shallowest BLs and the narrowest distributions": From which stability threshold the LL10 method apply the neutral and the stable detection method? This info can help interpreting Fig. 6. As stated in the caption of Figure 6, the stability classes plotted are from the eddy covariance. In the LL10 scheme, stability is determined based on the potential temperature difference $\theta_5 - \theta_2$ (as described in section 3): if $\theta_5 - \theta_2$ is less than -1K a convective BL is present, if it is greater than +1K a stable BL is present and if it is between -1K and +1K a neutral BL is present. We have also now compared the LL10 stability classes to the EC stability classes in section 3 and have added the related figure to the supporting information.

15. L504-510: For high stability the Ri (ERA5) leads per definition higher MLH than the parcel method (MWR) à this explain the results (see main comment 4) We are not sure we fully understand this comment as these lines refer to where we discuss unstable situations: *"for very unstable BLs the MWR diagnoses deeper BLs than ERA5 but for less unstable BLs the MWR*

*has shallower BLs than ERA5."* We do agree that the Ri method with a critical value greater than zero should give deeper BLs than the parcel method though.

16. L518-519: "These statistical results from the very stable case support our finding from the case study (Section 4.2) that the height of the BL diagnosed by the MWR under stable conditions can be significantly overestimated": I do not agree with this formulation. The method applied to MWR in stable cases is just different and try to measure another sublayer than ERA5. This sublayer corresponds also not to aerosol layers measured by ceilometers. Please see the response to major point 2 above and also Figures 1b and 3b in this response. We do believe that in some stable cases the MWR does diagnose unrealistically deep BLs and thus does over-estimate the BL height and that the differences between the MWR BL heights and the ERA5 values are not solely due to the difference in methods.

17. L542-544: "Note that the stable BL height values from the MWR have been filtered out, so most of the MWR BL heights during summer nights are not seen in Figure 8." Why ? I expect (due to the applied methods) that the removed MWR BL were higher than the ERA5. Is this right ? See the response to major point 2 above.

18. L563-564: do you have a tentative explanation for these kind of cases ? One hypothesis is that these weakly stable BLs are dominated by shear-driven turbulence and not buoyancy-driven turbulence. As the MWR only considers the potential temperature profile, it will not be able to estimate accuratly the BL depth in such conditions. ERA5 which uses a Richardson number method does account for shear driven turbulence and this could explain the deeper BLs in ERA5 in this case.

19. L566-572: it is however very important here to consider (and write a reminder for the reader) that most of the fall and winter periods ( 15 september-January) were not taken into account in the comparison between RS and ERA5. ERA5 was then considered as a good BLH retrieval method for spring and summer, when the atmospheric stability is very different than during fall and winter. MWR remains a measuring system and ERA5 a reanalysis, so that MWR results cannot be discarded without any clear reason. In fact, you wrote in the discussion (L728-729) "Thus, ERA5 still cannot capture the depth of very stable BLs accurately, which is likely due to deficiencies in the BL parameterization of lack of resolution". It is then very important to take this conclusion into account when describing and discussing the climatology. The comparison covers 1st Feb - 15 September 2014 and as we have written in response to major point 3 above, we do believe that we have verified ERA5 for all stability classes. As these lines refer to the 1-year (September 2018 - August 2019) comparison between ERA5 and the MWR we do not add information here concerning the time period of the radiosonde / ERA5 comparison as we feel it would be confusing for a reader. We do add elsewhere reminders that ERA5 was only compared against the radiosondes for 1st Feb - 13 Sept.

20. L574: Fig. 8 shows a complete year of MWR data. Why not at least provide the seasonal cycle of this first year of measurement? We have now created a new figure (Figure 2 in this response) which shows the average seasonal and diurnal cycle of convective BL heights derived from the MWR. For reasons discussed in our response to major point 2 above we excluded the stable cases as including them leads to clearly unphysical BLs during summertime nights (Figures 1b

and 3b in this response). Since we only include the convective BLs, this figure is not directly comparable with those from ERA5 and this is why we decided not to include it in the revised manuscript.

21. L575 as explained in a previous comment, ERA5 was validated only for spring and summer. In fall and winter (as seen in the climatology) weakly stable and near-neutral stable cases are much more frequent. In these cases, Ri can be largely influenced by the wind component (see results L584-586). This should be discussed anywhere. ERA5 was validated for February which is the coldest month in southern Finland. Figure 4 in this response does show that near neutral stable and weakly stable cases do occur more in the months not considered (October - January) than those we did considered. We have added some text based on previous studies about how Richardson number method behaves in weakly stable cases.

22. L638-645: same comment as for L575: the night seasonal cycle of BLH should also be discussed as a function of the wind compound in Ri method. We have analysed the wind climate using measurements made at 67.2 m and these show, as expected, that stronger winds occur in winter than in summer and that calm conditions are more common in summer than in autumn / early winter. While we appreciate that low-level wind speed is not the same as low-level wind shear, we do add some details concerning the seasonal wind variation to section 6.

23. L660-661: this conclusion is right only if Ri method is the right method to resolve BL height. We believe this conclusion is valid since there is good agreement between ERA5 and the Richardson number methods applied to the radiosondes. This at least means ERA5 represents the vertical structure of potential temperature and winds in a relatively ok manner.

24. L752-754: "A key outcome of our analysis is that the MWR does not reliably estimate the BL height under stable conditions, which at Hyytiälä, occur commonly at night between April and September." Once again, ERA5 and MWR just apply different methods estimating different sublayers in case of stable conditions. While we agree that ERA5 and the MWR use different methods, we still believe that our statement is valid (see response to major point number 2). We have revised the text in section 5.3 to make it clearer what the problem is with stable BLs and why we have removed them from Figure 8.

25. L760: once again, ERA5 was validated mostly in spring and summer. This should also be taken into account in the discussion. We have added text to the discussion to reminder a reader that we do not consider a full year, but we do sample all types of BL stability.

26. Fig 4: would it be possible to color dots of c), e) and f) with EC stability ? (or all plots with EC stability) We have made this figure and include it in the supporting material and as Figure 6 in this response. We did not not include this new figure in the manuscript for two reasons (1) we want to highlight how the stability from the LL10 scheme affects the diagnosed BL heights from the LL10 method and how they compare to BL heights from other methods and (2) some of the information about how the BL height from different methods relate to each other is already presented in Figure 6 of the manuscript, but in our opinion, in an easier to interpret manner. We also feel that including stability from the EC data on some panels and stability from the LL10 method on other panels would be confusing.

[Figure]

Figure 6: Scatter plots showing the relation between the BL height diagnosed from radiosondes taken during BAECC using 4 different methods. Data is from 1st February 2014 - 13th September 2014. Colours indicate the stability class from the EC measurements. Black solid lines shows the 1-to-1 line. Note the logarithmic scale.

27. Fig 7: since the MWR time series appears thereafter, I would indicate in the legend the period of comparison. The data included in Figure 7 is from the start of September 2018 to the end August 2019. We have added this information to the caption and also to section 2.3.

28. Fig 9: please also indicate the used period of time. We have now added the time periods to the caption.

29. Fig 11: mention that a) and b) correspond to ERA5 BL heights We have revised the caption to state that the BL heights are from ERA5.

30. Fig 12 (and similar figure in the supplement): it's not clear if all the plotted correlations (r>0.75) are statistically significant. Are correlations with r>0.75 sometimes also statistically significant ? All the values that are shown in Figure 12 are statistically significant and we have modified the caption to state this. This is likely due to the very large sample size (40 years). We assume the second question here was meant to be "are correlations with r<0.75 sometimes also statistically significant?". Yes, this does occur. However, in response to reviewer 2, we have modified Figure 12 to quantify how often the BL height at each grid point is within 150 m of the BL height at Hyytiälä.

31. Bigger font size in Figure S1 would be nice.We have increased the font size in this figure.

**Response to Reviewers – "Boundary-layer height and surface stability at SMEAR II, Hyytiälä, Finland in ERA5 and observations"**

Victoria Sinclair, Jenna Ritvanen, Gabin Urbancic, Irina Statnaia, Yurii Batrak, Dmitri Moisseev and Mona Kurppa

February 2022

We thank the reviewer for their constructive comments on our submitted manuscript. We have copied the comments of the reviewers in black here and include our response to each individual comment in blue.

**Reviewer 2**

The paper deals with the estimation of the atmospheric boundary layer height (ABLH) and surface layer stability for the Station for Measuring Ecosystem-Atmosphere Relations (SMEAR) in Hyytiälä, Southern Finland. This station has been in operation for about 25 years with the aim to study land surface atmosphere interaction processes including trace gas and aerosol measurements. The station is part of the European Research Infrastructure Networks ICOS and ACTRIS, its comprehensive characterization with respect to atmospheric mixing potential is thus of high practical relevance. Before 2018, profile measurements of atmospheric variables have been performed at Hyytiälä during short-term field campaigns only using radiosondes, a microwave radiometer has been installed at the site in 2018. To create a long-term ABLH climatology for the site, the authors therefore use ERA-5 data which are validated against radiosonde and microwave radiometer (MWR) observations for two shorter time periods, unfortunately the reference radiosonde and MWR measurements do not cover the same time period.

The paper is well written and well structured. The approach chosen follows a sound methodology, and the results obtained are comprehensively presented in the manuscript. The paper is of limited originality with respect to the methods applied and the relevance for progressing measurement techniques. It relies on the application of methods adopted from literature, not all of them are considered as appropriate. Findings regarding the strengths and weaknesses of the different methods to derive the ABLH from radiosonde and MWR are in line with previous studies reported in the literature. Data Analysis of the ERA-5 data is, to my opinion, the stronger part of the paper. The observations part could be shortened, e.g., I am not sure whether the case studies section is really needed and whether all the experimental methods need to be discussed extensively considering their (known) limitations and methodical differences to the approach used in the ERA-5 analysis. In addition, I see a number of specific deficits and minor issues the authors should address, before the manuscript might be finally accepted. These are specified below.

**Specific comments**

1. A few more experimental and methodical details should be given, e.g.

    - Which type of radiosondes was used during the 2014 experiment? I am not aware of any radiosonde that measures dry and wet bulb temperature. The sondes were standard Vaisala RS92. The text in line 120 has been clarified to make it clear that the radiosonde does not measure wet bulb temperature directly - we originally meant these variables are available in the data files downloaded from ARM. We have also added details about the vertical resolution of the soundings to this section.

    - Was the 12 UTC radiosonde really released at 12 UTC? (in the operational praxis of Met Services the so-called 12 UTC sounding is often performed an hour earlier or so to ensure that a profile covering the whole troposphere is available at the reference time even in case of a necessary second ascent if the first one fails at low altitudes) – this would affect the discussion in line 625ff. No, the radiosondes were not released at exactly 00, 06, 12 and 18 UTC, they were all released 9 - 45 minutes before these times which varied from day to day. The earliest 12 UTC sonde was released at 1115 UTC and the latest at 1151 UTC. We have added information about this to section 2.2.

    - The authors mention two sonics at 23 m and at 46 m, but it remains unclear which of the sonics was used to derive stability? We used the sonic at 23 m to derive the stability classes. We have revised the text in section 2.4 to make this clear.

    - Why did the authors use a temperature from within the canopy as a reference temperature? We used the 30-minute mean temperature measured at 16.8 m as it was the closest temperature to the surface. Furthermore, the 30-minute mean temperature was only available at 16.8 m and 33.6 m in the smartSMEAR portal. We have now investigate whether the choice of reference temperature strongly influenced the stability parameter by re-computing it using a constant reference temperature and found that it did not.

    - The definition of the stability class ranges in lines 178-180 (and also on p. 10) differs from what is indicated in Figure 6, 7 and in Tables 3, 4. The stability class ranges were correct in Figures 6 and 7 and in Tables 3 and 4. They were incorrect in the text but have now been updated. This error stemmed from originally having the boundary between the weakly and very cases at $\pm 1$ but the distribution made more sense to have it at $\pm$ 0.1 and thus we changed this during the research process. There were no errors in the statistics or conclusions presented in the manuscript.

    - The authors work with quite a number of different stability estimates (from the sonic, from the LL10 method, from the MWR output) – all these methods rely on different variables and represent different scales – have the results from these methods ever been compared and critically assessed? This is a valid point and we were not aware of any previous comparisons. We now have compared the stability classes derived from the sonic (eddy covariance data) to the LL10 stability classes (Figure 1 in this response). In general, there is reasonable, but not perfect, agreement. First, the LL10 method mainly (254 times) diagnoses neutral BLs when the EC diagnoses either very unstable or weakly stable BLs. However, there are 28 soundings for which the EC stability class indicates either very unstable or weakly

[Figure]

Figure 1: Comparison of the stability classes derived from the LL10 method (y-axis) and from the eddy covariance data (x-axis). Shading and numbers show the number of soundings in each bin. In the left hand panel, the EC stability classes have been group so that unstable includes very and weakly unstable, neutral includes near neutral unstable and near neutral stable, and stable includes weakly and very stable classes. NaNs are almost all caused by missing EC data.

unstable conditions but the LL10 stability is stable. There were 134 cases where the EC stability was weakly or very stable and the LL10 stability agreed, also predicting stable conditions. The biggest discrepancy is the 139 cases where the EC stability class is very or weakly stable but the LL10 stability class is neutral. It appears that the LL10 neutral class is too broad and contains many cases deemed unstable by the EC method and some stable cases. Other studies have also shown the convective BL type as diagnosed by the LL10 scheme to be less common than neutral BLs over land (Zhang *et al.*, 2021) . We now include these additional figures in supporting material and have added some discussion to section 3 where we describe the LL10 scheme.

- For the temporal assignment it appears a bit irritating in several places that UTC is translated to local legal time instead of local solar time since ABL dynamics is related to the latter rather than to the former. We now include both local legal time and local solar time in the text when we have translated UTC to local legal time. We retained local legal time as it may be useful to some readers who want to compare other studies to this one (many papers using aerosol data from Hyytiälä use local legal time).

2. A critical discussion on the limitations of the measurements and methods is largely missing – just a few examples:

- According to Figure 1, the sonic is positioned at the edge of an about 1:12 slope with a lake and some larger clearings just a few hundred meters to the west. Does this have any influence on the representativeness of the sonic data? How can it be justified that a local z/L value close to a forest canopy is appropriate to characterize atmospheric stability up to a height of several hundred meters. Did the authors perform any comparison of, e.g., the stability derived from the eddy covariance measurements compared to the one derived from the soundings in the frame of the LL10 method – see above? We agree that the station

site is not homogeneous and therefore does have some limitations when observations from slightly different locations are compared. We have added text to section 2.1 noting the non-homogeneous nature of the site and the potential limitations of the measurements. We also have added text to the methods section to highlight how the non-homogeneous nature of the site, and the different measurement locations, means that the EC measurements may not be completely representative of the surface layer where the radiosondes were released.

- Both the parcel and Ri number methods are quite sensitive to the surface temperature value used in the analysis, and it is always a matter of discussion which data to use here. A near-surface temperature measured independently in a shelter often does not fit to the radiosonde profile, on the other side the first temperature values from the radiosonde profile may suffer from the sonde being in the hands of the observer before being released and not experiencing the ventilation as during the free flight. In some studies, an excess temperature is introduced or a limit for a possibly superadiabatic near-surface lapse rate is set. These issues should be addressed in the methods description. Yes, we agree that these are important aspects and have added text about these issues in section 3

- In lines 522-523, the authors bring in another ABLH estimate which is based on a completely different variable than all estimates considered in this paper. I suggest to omit this, otherwise it would call for a critical discussion of ceilometer ABLH estimates which is a delicate task itself. We have removed the text about the Jiang et al (2021) study from this section as we agree that we do not want to include discussion of estimating the BL height from ceilomters.

- Figure 9 indicates quite a substantial part of missing data, even in summer. For winter the authors argument with icing of the sensor, what are the possible reasons in summer? Are these missing data assumed to introduce a bias in the distribution of stability classes? The main reasons for the missing data in summer is due to the thresholds and requirements of the methods used to compute the fluxes. First, the data must be stationary enough over the 30 minute averaging time and second the friction velocity must exceed a threshold otherwise these timesteps are removed. We do not think that the missing data introduce a bias into the distributions as although there is an annual cycle in missing data there a much weaker (or no) diurnal cycle in the amount of missing data (Figures 10b, 11c).

3. The authors in their analysis mix methods that are based on the consideration of different physical processes, they thus compare apples and pears naming all these derivatives ABLH without a critical discussion on that. This necessarily will result in some dis-agreement when comparing the results. It is obvious that Heffter's method is basically suited to obtain an estimate of the convective ABLH, and previous studies have already demonstrated the tendency of this method to overestimate ABLH. From the MWR, in stable conditions, they derive a kind of inversion or stable layer upper boundary, that is purely based on an analysis of the temperature profile which contains limited information on shear-induced turbulence, if at all. For the ERA-5 analysis, a Ri number method is considered as appropriate, it would thus be consistent to put emphasis on those experimental data which provide a comparative ABLH estimate.

This is a valid point. We fully agree that the different methods are estimating slightly different things and acknowledge that whether shear-driven turbulence is considered or not, will have a

critical impact on the values of BL height derived. We have added text to section 3 to highlight this limitation and attempt to relate some of the detected differences in section 5 and 6 to the differences in the methods. Furthermore, we agree that the fairest comparison with ERA5 is with the Richardson number methods applied to the radiosondes. We now attempt to make this clearer in section 6 and try to put more weight on this comparison.

4. The presentation of results in Sections 5 and 6 is widely descriptive, I would love to see some discussion of the findings already here, even if it is partly given in Section 8, e.g., We have attempted to add more physical explanation into section 5. However, for many of the points we cannot provide solid evidence or show causality. Therefore, much of the discussion is somewhat speculative.

   - Why is ERA-5 in better agreement with Ri(0.5) from radiosoundings for the very unstable case, but with Ri(0.25) for the weakly stable case? This is difficult to know for sure. ERA5 uses a critical Richardson number of 0.25 which should give shallower BLs than the sonde with $Ri_c$ =0.5. However, ERA5 has lower vertical resolution which means that in very unstable conditions the capping inversion may be weaker (more smoothed out) in ERA5 than in the sounding (which is the case at 12 and 18 UTC on the 7th of May in the case study example). This would cause a higher BL in ERA5 than in the sounding if the same $Ri_c$ value was used. This may explain why ERA5 has deeper BLs compared to the radiosondes with $Ri_{0.25}$ and hence why ERA5 agrees better with the radiosondes with $Ri_{0.5}$. For weakly stable cases, ERA5 and $Ri_{0.25}$ agree well whereas $Ri_{0.5}$ estimates deeper BLs than ERA5. This may be because the type of vertical profiles found in weakly stable cases are easier for ERA5 to resolve as they likely do not have strong surface-based inversions or strong capping inversion. However, this is somewhat speculative - a thorough analysis of how ERA5 represents vertical profiles of potential temperature and wind under different stability classes is not within the scope of this current study (although it would be very interesting!).

   - Why are very shallow BLs less common than moderately shallow BLs in winter? It is hard to know for sure but there are a few possible explanations. To get very shallow BLs, usually very stable conditions are required. This needs calm wind conditions and strong radiative cooling at night so clear skies. Figure 2 in this response shows that the mean, median, 10th and 90th percentiles of wind speed measured at the main SMEAR station at 67.2 m is higher in winter than in summer. In addition, Figure 2 also shows that the percentage of times that calm conditions (wind speed $< 2ms^{-1}$) occurs is less in winter than in summer. Therefore, it is windier in winter which promotes shear driven turbulence which likely prevents very shallow BLs developing.

     A recent study using ceilometer and pyranometer data at Hyytiälä (Ylivinkka *et al.*, 2020) shows that winter has more cloud than other seasons. This also helps explain why very shallow BLs are less common in winter as the presence of cloud will prevent strong radiative cooling at night. Ylivinkka *et al.* (2020) also showed that there was no diurnal cycle in cloud cover from October to April. This helps explain our results that there is no diurnal cycle in the depth of the BL from November to March. We have added this discussion about the seasonal variation of windspeed and cloud cover to section 6

[Figure]

Figure 2: Left: The monthly mean, median, 10th and 90th percentile of windspeed measured at 67.2 m at SMEARII estimated using data from January 1997 to December 2019. Right: The monthly mean value of the standard deviation in the wind speed and the monthly mean value in the difference between the 90th and 10th percentile of windspeed measured at 67.2 m

- Why is the ABL height variability so large in April and May? The BL height variability is largest in May and second largest in June and this should be explained. The variability in April is smaller so we do not focus on this. We analysed the long-term temperature climate and its variability over the 23 year period. The BL height is most likely very variable in May since the temperature is more variable in May than in most other warm season months (Figure 3 in this response) - both the standard deviation and the difference between the 90th and 10th percentiles show a local maximum in May. Large variations in cloud cover could also be a reason for more variability and Ylivinkka *et al.* (2020) show that there is a diurnal cycle in cloud cover at Hyytiälä in May with a peak in late afternoon. We have added more discussion about the high variability in BL height and its likely causes to section 6.

- How to explain the annual cycle in the occurrence frequency of the weakly stable class (line 620ff)? This is not easy to explain. The weakly stable case has two peaks, one in October and one in March. These peaks are primarily caused by the frequent occurrence of weakly stable cases at night - a feature which is not evident in winter or summer. Unlike in summer, it is too windy in March and October for very stable BLs to occur frequently at night so even in the case of some radiative cooling, weakly stable BLs form instead of very stable BLS.

5. Deficits exist concerning the discussion of the authors work with respect to previous studies on, e.g., ABLH climatology at higher latitudes. How do the authors results behave when compared to the cited studies of Liu and Liang (2010) and Seidel et al. (2010, 2012)? I would also like to bring to the author's attention the work of Lotteraner and Piringer (Boundary-Layer Meteorol. 161 (2016), 265-287 – concerning the deficits of Heffter's method), as well as of Beyrich and Leps (Meteorol. Z. 21 (2012), 337-348 – concerning the analysis of radiosonde data: many of the authors results on methods comparison, annual variability and uncertainties have been reported there as well). Liu and Liang (2010) combine multiple locations in many of their plots so a fair,

[Figure]

Figure 3: Left: The monthly mean, median, 10th and 90th percentile of temperature measured at 4.2 m at SMEARII estimated using data from January 1997 to December 2019. Right: The monthly mean value of the standard deviation in the temperature and the monthly mean value in the difference between the 90th and 10th percentile of temperature measured at 4.2 m

direct comparison is quite difficult. Liu and Liang (2010) find that for land stations, neutral BLs are more common than convective BLs during the day which is in agreement with our results. However, we find notably fewer convective BLs than Liu and Liang (2010). We have added a statement about this to section 3. One difference between our results and those of Liu and Liang (2010) is that while we find that the mean BL height is shallower in late spring and summer compared to winter, no such seasonal cycle is detected by Liu and Liang (2010). However, they only show this result for the Southern Great Plains site (36N) so this is not too surprising given that Hyytiälä has a much greater variation in day length and a very different climate in general. Seidel *et al.* (2012) exclude Arctic areas and only investigate European stations between 35 and 60N so a direct comparison is again slightly difficult. However, our results do agree quite well - summer daytime values of BL height averaged over all of Europe are deeper than what we found (potentially due to the lower latitudes) but the same seasonal cycle is evident. Seidel *et al.* (2012) also find that at night-time ERA-Interim gives consistently higher BL heights than obtained from the radiosonde soundings which is agreement with our results between ERA5 and the $Ri_{0.25}$ method. Furthermore, at night in Europe, both Seidel *et al.* (2012) and Beyrich and Leps (2012) find that the shallowest BLs occur in spring and summer which is in agreement with our results. We also find agreement with the results of Lotteraner and Piringer (2016) in that they also find that the Heffter method diagnoses deeper BL heights at 12 UTC compared to the Richardson number method and shallower BLs at 00 UTC. We have added more comparison to these studies in the discussion section of the revised manuscript in an attempt to better relate our results to these previous studies.

6. For the Figures, I noticed some issues concerning either the interpretation in the text or the methods chosen to display the data:

   - Figure 2 (related to text line 295ff): It appears not really possible to reproduce the ABLH values given in the text from Figure 2. At 00 UTC, I would not see any of the values at 307 m – all the lines appear to be well below 1/4 of the first height axis tick label at 1000

m. The same is with the inversion at the 06 UTC plot which appears to be well below 500 m. And at 12 UTC, the lowest plotted estimate is definitely below 1000 m whereas the text gives the minimum value with 1073 m. Thank you for highlighting this. The values reported in the text were the BL heights above sea level (which is how ARM reports them in the VAP files) whereas the values plotted are the height above ground level. Therefore the values in the text all had a positive bias of 179 m. The values in the text have now been corrected to be height above ground level and thus now agree with the horizontal lines in Figure 2.

- Figures 5-7, 9, 11: I am not sure whether the choice to plot the whiskers at Q1 – 1.5 IQR and Q2 + 1,5 IQR is a clever one given the skewed distributions of the variables, because this does not consider the real distribution of data points in the lowermost and uppermost quartiles. It may thus happen, that – especially the lower – whiskers cover a range where no data points occur. This becomes obvious by the many whiskers for ABLH ending at z = 0 m. I probably would have chosen to represent the 10% / 90% percentiles by the whiskers. This is a valid point and therefore we have re-made figures 5-7, 9 and 11. However, we have plotted the whiskers to represent the 5% and 95% percentiles as this reduced the number of outliers (and made the figures tidier) in Figure 5-7 compared to when we tried 10% to 90%.

- Figure 10: It does probably make not much sense to average the mean diurnal cycle of the stability classes occurrence frequencies over the whole year as the seasons behave very different.Yes this is true and it is one reason we also include figures 11c and 11d. However, to make figure 10 consistent with figure 9 we decided to keep panel 10b. We now note in section 6 / the discussion of Figure 10 the limitations of averaging the mean diurnal cycle of the stability classes occurrence frequencies now.

- Figure 12: I am not sure whether the correlation coefficient is a suitable measure for this type of analysis, at least if the authors chose this as the only parameter considered. From the seasonal distributions we have learned that derived ABLH values vary over a wider range in summer than in winter, this automatically gives higher correlation coefficients if at least the general tendencies are the same in the two data sets considered. Wouldn't it be an idea to consider absolute or relative differences here as well, the more since for practical applications it might be of higher relevance to know the area over which ABLH does not differ by more than a certain absolute (say 100 m or 200 m) or relative value (say 20 %), rather than to know correlation coefficients. Similarly, I doubt whether just from the value of the correlation coefficient one may judge about the spatial variability (lines 680f) Originally, we were more interested in the spatial patterns of the BLH 'evolution' and not as much in the BLH 'value' per se and this is why we used correlations. However, we find the idea of absolute differences to be very interesting as, as the reviewer states, this may be useful information in many practical applications. We have computed the percentage of times during each month and each synoptic time that the BL height at each grid point is within 150m of the BL height at the grid point closest to Hyytiälä. This information is included in a revised versions of Figures 12, S2 - S5 and is discussed in section 7. However, it is important to note that using an absolute value, rather than a relative value, does mean that this diagnostic is impacted by the diurnal and seasonal cycle of the BL height which

we also highlight in the revised text.

**Minor Issues**

1. The abbreviation SMEAR is introduced in the title already and it occurs in the abstract again before it is finally explained in line 73. We have removed the abbreviation SMEAR from the title and abstract.

2. The abstract could be shortened a bit. E.g. the first two sentences might be deleted, they would well fit in an introduction section, but are unnecessary in an abstract. I also consider it for unusual in an abstract to write "for example ..." – either the result is so relevant that it should be stated or it might be omitted. We have revised the abstract based on these suggestions and have also attempted to shorten it.

3. Line 36-37: Here the authors state that vertical profiles of temperature and wind speed are required to determine the ABLH, then they write that "these profiles can be obtained from ... ceilometers." I do not know any ceilometer that would provide wind and temperature profiles. Correct, ceilometers only provide backscatter. We have deleted ceilometer here and also added text to be clear that the MWR only provides profiles of temperature.

4. Line 45-46: Another limitation of remote sensing instruments has to be seen in the fact that often more than one instrument is needed to measure the relevant variables, while tower measurements or sondes may provide the full set of essential thermodynamic variables. Yes, this is very good point. We have added text about this to the introduction.

5. The authors do not seem to be friends of using commas to structure a sentence, this makes it a bit difficult to read in some places (e.g., line 74, 632, 709, 737, and others) We have revised these specific cases and have re-read and edited the manuscript paying attention to the punctuation.

6. Line 119 / line 129: If M is the "main SMEAR station location" (line 110) in Figure 1, I would see the location R to the Southwest of M, not to the Southeast.This was a mistake and it has now been revised to southwest.

7. Line 207: "Stable boundary layer are not characterized by inversions." – This is a statement I simply do not understand. We meant that, unlike convective, well-mixed BLs, stable boundary layers do not have a capping inversion above a well mixed layer. However, we agree this was a confusing sentence, as stable BLs are usually characterised by surface-based inversions. The start of section 3 has now been completely revised and this sentence is no longer included.

8. Line 246: Isn't it a bit dangerous to consider just the immediate neighboring levels below AND above a given height to diagnose a low-level-jet. Often the jet nose may be not that sharp that there is a significant change of wind speed both below AND above the height of the maximum? Yes, we agree that taking just one level below and above could be problematic and would only capture very sharp jets. In the ARM VAP documentation (Sivaraman *et al.*, 2013) the word immediately is used. However, in Liu and Liang (2010) they state that "the LLJ nose is identified at the level where wind speed reaches a maximum that is at least $2 \mathrm{ms}^{-1}$ stronger than the layers above and below". This suggests that it is not just the immediate levels above and below.

Hence we are not 100% sure which levels the ARM VAP considered. We have deleted the word immediately here as we think it is unlikely that only one level above and one below is considered.

9. Line 269: The use of z0 is a bit misleading here, z0 is normally taken as the aerodynamic roughness length. Yes, we agree that this is confusing and revised this. We have replaced z0 with z1.

10. Line 278-280: The Vogelezang and Holtslag method additionally considers a correction term to account for surface friction / shear. Yes, we were aware of this and it is discussed in the IFS documentation. The method used in the IFS (and thus ERA5) to compute the Richardson number does not include the friction velocity term as *"since the friction velocity is not known from radiosonde data, the surface frictional effects are ignored in the computation of the bulk shear"* (IFS documentation). Therefore, the reference to Vogelezang and Holtslag (1996) is potentially misleading and we have removed it from this specific sentence. Furthermore, the Richardson number is computed using virtual dry static energy in the IFS whereas the Richardson number computed from the soundings uses virtual potential temperature. We have added some more details to the manuscript in section 3.3. about how the BL height is computed in the IFS / ERA5 and note more clearly how it differs from the Richardson number methods applied to the radiosonde.

11. Line 312: It is not clear to me why – "due to the inclusion of the wind profile ... the Ri number methods diagnose shallower SBLs". Additional consideration of the wind profile would add shear-induced turbulence. Parcel methods will give ABLH where $\theta_h > \theta_0$ while Ri methods will lift this up before Ri exceeds a value of Ri = 0.25 or Ri = 0.5. In fact, parcel methods are equivalent to Ri methods with Ricrit = 0, and with critical Ri values above zero they should give higher ABLH values. We agree that include wind speed and hence accounting for shear-driven turbulence should results in a deeper BL. We have deleted statement.

12. Line 327-328: I cannot follow this argumentation. The wind speed differences with respect to the near surface values are larger in the measurements when compared to the ERA-5 profiles up to a height of about 2.5 km, consequently the denominator in the Ri definition gets larger, and hence Ri gets smaller. It thus appears not logical that the threshold of Ri = 0.25 should be exceeded at lower heights. We have done further investigation of the case study shown in Figure 2 of the manuscript. It proved difficult to fully understand as we took the ARM provided values of BL height from the Richardson number methods and did not compute them ourselves. However, we now believe that the underestimation by the Ri methods at 12 UTC maybe related to the choice of surface values in this case or is affected by the vertical potential temperature profile which has some weakly stable layers present. Consequently we have revised the text in section 4.1 and removed this sentence.

13. Line 370: I would prefer to call this RMSD instead of RMSE. To use the word "error" would imply that the truth is known based on a well-calibrated method. We agree with this suggestion as in this study we do not have a measurement or model values that are obviously the "truth". We are unsure if "D" stands for difference or deviation here (both appear in the literature) but we now now replace RMSE with RMSD (Root Mean Square Deviation).

14. L406-L408: How to explain the outliers in Figure 4 where Ri(0.5) < Ri(0.25)? We investigated this and found that there are only 7 cases out of the 827 soundings where the BL height diagnosed from the $Ri_{0.5}$ method is lower than from the $Ri_{0.25}$ method. The cases occur in the sondes released at the following times: 18 May at 0520 UTC (358.0 m), 17 June at 2316 UTC (13.5 m), 21 June at 1118 UTC (17.4 m), 26 June at 1117 UTC (36.2 m), 13 August at 0517 UTC (207.0 m), 15 August at 0519 UTC (13.5 m) and 22 August at 2332 UTC (124.3 m) where the numbers in brackets show the difference in the two diagnosed BL heights. These cases have $Ri > 0.25$ from the surface upwards to a low-level maximum in $Ri$ above which $Ri$ then decreases. Hence the profile crosses $Ri = 0.5$ for the first time at a lower level than it crosses $Ri = 0.25$. This situation arises when there is a very shallow stable layer at the surface with a well-mixed layer above it but below another stable layer. These cases may be affected by instrument error e.g. the surface initial values may be inaccurate (although they were not flagged by the ARM quality control flags.)

15. L458: I would not a-priori expect that the Ri(0.25) methods radiosonde data must be in better agreement than the Ri(0.5) method with the ERA-5 ABLH estimate, even if the latter uses the same threshold: Note that the profiles have different vertical resolution and the reanalysis profiles are usually much smoother than any measured profile which will strongly influence the level of exceedance of fixed threshold values. Yes, this is a good point and in section 3 we have added details to highlight the likely differences in the diagnosed BL height from ERA5 and from the Richardson number method applied to ERA5. We have also revised this sentence in section 5.2.

16. Line 466-467: 23% is a quite accurate number, I would not call that "approximately". We have deleted "approximately" here.

17. Line 609-610: This seems to be a bit too general, the very stable cases do occur more frequently in summer. Yes, this was too much of a generalisation here. This sentence has now been revised.

18. L643: "In winter cold conditions are required for shallow boundary layers to develop" – This statement calls for some further explanation. Sorry this was too generic and too vague a statement. We have now revised it to state that shallow BLs tend to develop under stable conditions, which usually occur under calm and clear conditions.

19. L669: It appears to be a subjective view whether Figure 12 really represents a "considerable part of Northern Europe". We have now deleted the word "considerable" here.

20. L724f: This conclusion does a bit contradict to the discussion 457-463 ("high vertical resolution ... sufficient" vs. "limited vertical resolution" explaining disagreement) This is a small language issue and potentially we were not explicit enough. In Line 724 we did add "in most situations" at the end of the sentence which to us suggests that there are some specific situations (i.e the shallow cases mentioned in lines 457-463 of the original manuscript) that the resolution is not adequate. We have revised this to be more explicit.

21. L735: This is not a suggestion but a rather logical consequence of the stability classification with the LL10 method as described in Section 3.1. We have replaced "This suggests" with "This means".

22. L742-743: Again, a minor inconsistency: Here the authors, to my opinion, correctly point at the difficulties to determine the ABLH during the evening transition both from the measurements and in ERA-5, in Section 5.2 (line 496) they solely "blame" ERA-5 for the deficits. We think that radiosondes would more accurately represent the evening transition than ERA5 but it is likely that one radiosonde may not be completely representative of the atmospheric state during the evening transition when the BL structure changes rapidly. We have modified the text in the discussion to be clearer that it may not just be ERA5 at fault here.

23. L767f: I also see the widely missing diurnal cycle (no pronounced daytime heating) as an additional reason for the reduced occurrence of very stable situations in winter, except of the more frequent occurrence of clouds. We have added text about the lack of radiative heating during the day and its likely impact on the frequency of very stable BLs to the discussion.

24. Figure 2: Relative humidity is not a good choice to plot vertical humidity profiles for illustrating the ABL structure, absolute or specific humidity would be better here. In essence, the humidity profiles are not even needed to be displayed here, at least they are not discussed in the text. We have removed the relative humidity profiles from Figure 2 as we did not discuss these profiles in the text. In addition, we have also changed the scale on the y-axes to better show the structure of the BL at 00 UTC.

25. Figure 4: The diagrams give the impression that some lower boundary values exist for the LL10 method (no stable values below 40 m, no unstable values below ca. 150 m) – nothing has been mentioned about that when describing the method. For neutral or convective cases, the two vertical upward scans (to check for the required thresholds in potential temperature difference and potential temperature gradient) start from the data level right above 150 m a.g.l. to avoid noisy readings near the surface. Therefore, in the LL10 method, convective or neutral BLs cannot be shallower than 150m. However, for stable cases there is no minimum boundary value. To confirm this, we plotted the lowest BL heights which are shown in Figure 4 in this response. This zoomed in figure illustrates that there is no sharp cut-off at 40 m. We have modified the text in section 3.1 where we discuss the LL10 method and also remind a reader of the minimum in convective and neutral cases when we discuss figure 4 in the manuscript.

26. Figure 5: The black points are black crosses. The caption has been revised accordingly.

27. Figure 8: It appears unnecessary to explain the legend again in the text of the Figure caption. We have kept this information in the caption as it contains more information (e.g. that hourly median values are plotted) than is possible to fit into the legend.

28. Figure 9, caption: Just to make sure: panel (b) is stability from the sonic or from ERA-5? The data period represented by the graphs should probably be mentioned in the Figure caption, the more if it is not the same (ERA-5 40 years vs. sonic 19 years, as in Figure 10?) The stability data shown in Figure 9b is from the sonic. We have revised the caption to state this and to now include the time periods of the two data sets.

29. Table 3, caption: NMSE must be nRMSE, in the Table NRMSE should be nRMSE according to the definition given in the text (but see my remark on RMSE vs. RMSD) This has now been revised.

[Figure]

Figure 4: Boundary layer heights less than 200 m diagnosed by the LL10 scheme. Blue points show BLs diagnosed as stable by the LL10 scheme, orange points neutral BLs and green points convective BLs.

**References**

Beyrich, F. and Leps, J.-P. (2012). An operational mixing height data set from routine radiosoundings at lindenberg: Methodology. *Meteorologische Zeitschrift*, pages 337–348.

Liu, S. and Liang, X.-Z. (2010). Observed diurnal cycle climatology of planetary boundary layer height. *J. Climate*, **23**(21), 5790–5809.

Lotteraner, C. and Piringer, M. (2016). Mixing-height time series from operational ceilometer aerosol-layer heights. *Boundary-Layer Meteorology*, **161**(2), 265–287.

Seidel, D. J., Zhang, Y., Beljaars, A., Golaz, J.-C., Jacobson, A. R., and Medeiros, B. (2012). Climatology of the planetary boundary layer over the continental united states and europe. *J. Geophys. Res.*, **117**(D17).

Sivaraman, C., McFarlane, S., Chapman, E., Jensen, M., Toto, T., Liu, S., and Fischer, M. (2013). Planetary Boundary Layer (PBL) Height Value Added Product (VAP): Radiosonde Retrievals. Technical report, Department of Energy Office of Science Atmospheric Radiation Measurement (ARM) Program. DOE/SC-ARM/TR-132.

Vogelezang, D. H. P. and Holtslag, A. A. M. (1996). Evaluation and model impacts of alternative boundary-layer height formulations. *Boundary-Layer Meteorology*, **81**(3-4), 245–269.

Ylivinkka, I., Kaupinmäki, S., Virman, M., Peltola, M., Taipale, D., Petäjä, T., Kerminen, V.-M., Kulmala, M., and Ezhova, E. (2020). Clouds over hyytiälä, finland: an algorithm to classify clouds based on solar radiation and cloud base height measurements. *Atmospheric Measurement Techniques*, **13**(10), 5595–5619.

Zhang, D., Comstock, J., and Morris, V. (2021). Comparisons of planetary boundary layer height from ceilometer with arm radiosonde data. *Atmospheric Measurement Techniques Discussions*, pages 1–28.

---

## Author Response (AR2)

**Response to Reviewers – "Boundary-layer height and surface stability at SMEAR II, Hyytiälä, Finland in ERA5 and observations"**

Victoria Sinclair, Jenna Ritvanen, Gabin Urbancic, Irina Statnaia, Yurii Batrak, Dmitri Moisseev and Mona Kurppa

April 2022

This study presents a very valuable comparison of several BLH detection methods as a function of atmospheric stability computed by several methods. This analysis and the climatology are the most interesting parts of the study and the discussion has been largely improved in the reviewed version of the manuscript.

I still have a main comment about the choice of the vocabulary used for some conclusions. The authors choose to strictly apply the BL height detection methods provided at ARM (e.g. even if several publications mentioned that a better threshold should be applied for the H80 method) and, as mentioned several time in their answers, they did not compute the values by themselves. A major consequence of this choice is that BLH estimated by Ri is taken as the true boundary layer height. The fact that the sole expression "boundary layer" is used regardless of method and time is characteristic of a simplified vision of a basic boundary layer with ONE characteristic height. It is however obvious that the authors have a large knowledge on the topic and that the words "fail" (lines 652, 661, 673, 907, 908) and "true" (lines 622, 842, 903) just tend to emphasize the results by dichotomous considerations. Surprisingly, the word "truth" is used three times to warn the readers "none of the BL height considered here are clearly the truth" (line 444). The Ri method allows a good BL characterization in most cases but it remains limited for some atmospheric conditions. I ask therefore the authors to really discuss the complexity of the BL height determination in the comparison between MWR and ERA5 results by nuancing the notions of "true" and "false".

We thank the reviewer for taking the time to provide these comments. We reply to the minor points below in blue. As a general response, we did not mean to give the impression that the BL height estimated by Ri is true boundary layer height. We are of the option that all methods applied to the radiosondes provide valid estimates of the broadest definition of the BL - the part of the atmosphere influenced by the surface. However, we do acknowledge that some methods are designed to identify only certain types of BLs that could be called a mixed layer or mixing layer.

Regarding the MWR diagnosed BL heights, we have carefully revised the text in this section to stress we are comparing ERA5 and the MWR and it is unclear which is correct. We have revised the manuscript to remove words such as fail and true. The exception is the text describing when the MWR retrieval really does "fail" which is occasionally the case due to for technical reasons.

**Minor comments**

1. Line 3: the use of the two "and" leads to confusion about the role of " the level of agreement". This sentence has been split into two sentences to make it clearer.

2. Line 230-231: The problem is not of "no truth", but of the complexity of BL, the definition of BL height and the fact that BL height is more a layer than a precise altitude. In that sense the use of the word "subjective" is completely inappropriate since your study relies to measured/modeled data. These sentences have now been revised. We delete "subjective" and now refer to the complexity of the BL.

3. Lines 288-289: What are the bias/problems bounded to the altitude differences used to define stability classes? Which method is the most trustful one? The stability determined from the eddy covariance system reflects the stability of the surface layer whereas from the sounding, the stability reflects more that of the lower boundary layer. It is hard / impossible to know which is most trustworthy. However, we hypothesis that the stability from the sounding may be more representative of the large-scale environment as it is not directly above the forest canopy. However, on the other hand, the radiosonde derived stability estimate is from one instantaneous measurement, whereas the stability from the eddy covariance system is a 30-minute average of very high resolution observations. Some of the discrepancies in Figure S1 may be due to the differences in altitude used / the two diagnostics measuring slightly different things but as we discuss in the conclusions, we believe that the stability threshold used by the LL10 scheme is not appropriate for this location.

4. Lines 338-339: please give the "surface height"used for the sounding instead of "much lower". It would be also nice to have a comparison with the "surface height" of the MWR. This sentence has been revised to now read *However, in ERA5 this is typically around 10 m a.g.l. whereas in the radiosonde soundings this is much lower - typically 2 m a.g.l.*. The potential temperature at the "surface" that the MWR uses is taken from the automatic weather station which incorporated with the MWR. This means that the "surface" potential temperature is at a height of 1.5 m above the surface. We have added this to the manuscript in section 4.2.

5. Fig. S2: the x labels of plot e and f are missing. Thank you for pointing this out. This has now been corrected.

6. Lines 473-479: please add a sentence about the results of Fig. S2: the Heffter method over-estimates BLH mostly for unstable EC conditions (similarly to Fig. 4). LL10 overestimate RI mostly for stable conditions (different to Fig. 4, the answer being probably given by Fig. S1). The apparent difference for how the LL10 BL heights compared to the $Ri_{0.25}$ BL heights for different stabilities is related to the different definition and bins of stability. In Figure 4b, the red points include neutral and unstable points identified from the LL10 scheme. The points where the LL10 method has BL heights between 200 - 800 m, which generally are larger than the BL heights, are red meaning either unstable or neutral stratification. In Figure S2, the points in the this area are mainly light blue (near neutral stable), middle blue (weakly unstable) and orange (near neutral unstable). Figure S1 tells us all of these cases are most likley to be diagnosed as neutral by the LL10 scheme and thus appear as red points in Figure 4. Therefore, we see not discrepancies in the results in Figures 4 and S2 that need to be explained.

7. Lines 661-673: In case of weak stability and very low wind speed, the Ri method applied by ERA5 leads to very high, constant BLH. The description is exact, but I do not think that you can qualify the MWR BLH estimate as a failure and ERA5 BLH estimate as the real value. The ERA5 BLH follows the aerosol top layer (Fig S4 c)), which is not always considered as BL top (e.g. the authors do not consider RL top as BLH in previous sections of the manuscript dealing with MWR BL detection during night). Moreover, as mentioned in your paper, MWR is an instrument and ERA5 a model and weak unstability is detected (Fig. S4 d) when MWR presents a CBL development. The only use of the expression "boundary layer" for the various methods and times is an easy shortcut of this manuscript. However it induces a too easy appreciation of the "true BLH", but BL, especially during night, is often complex. Then, to which boundary layer or sublayer correspond ERA5 BLH of the 28th of October 2018? We agree with the reviewer here in that a "true" BL height is not a good term as it implies an exact height whereas in reality, and depending on the application and the method used, a range of values / heights is potentially "correct" or at least appropriate. We have tried to stress this more in parts of the manuscript. In the introduction we added text to stress that our definition of the BL is very broad and that we acknowledge there are many different types of the BLs and that the BL structure is complex. We have also modified the text in lines 661 - 673 to remove terms such as "fail" or "true".

8. Line 673: see previous comment on success and failure. This sentence has now been removed.

9. Lines 795-796: As already commented in the first review, this conclusion is right only if Ri is the right method to resolve BL height. The authors answered that Ri applied by ERA5 leads to

similar results to Ri applied to RS. This proves that ERA5 temperature and wind profiles are similar to sounding profiles, but it does not demonstrate that Ri is the right method. We agree that we cannot conclude that the Richardson number is the "right" method to diagnose the BL, although previous studies have suggested that this is often a good method to use with model / reanalysis data. We have now revised this sentence to be clear that the results show that the temperature and wind profiles in ERA5 and the radiosondes are similar and that if the same method is applied to both to identify the BL height, good agreement can be expected.